# Self-Calibrated Consistency can Fight Back for Adversarial Robustness in Vision-Language Models

## Abstract

Pre-trained vision-language models (VLMs) such as CLIP have demonstrated strong zero-shot capabilities across diverse domains, yet remain highly vulnerable to adversarial perturbations that disrupt image-text alignment and compromise reliability. Existing defenses typically rely on adversarial fine-tuning with labeled data, limiting their applicability in zero-shot settings. In this work, we identify two key weaknesses of current CLIP adversarial attacks—lack of semantic guidance and vulnerability to view variations—collectively termed semantic and viewpoint fragility. To address these challenges, we propose SELF-CALIBRATED CONSISTENCY (SCC), an effective test-time defense. SCC consists of two complementary modules: *Semantic consistency*, which leverages soft pseudo-labels from counterattack warm-up and multi-view predictions to regularize cross-modal alignment and separate the target embedding from confusable negatives; and *Spatial consistency*, aligning perturbed visual predictions via augmented views to stabilize inference under adversarial perturbations. Together, these modules form a plug-and-play inference strategy. Extensive experiments on 22 benchmarks under diverse attack settings show that SCC consistently improves the zero-shot robustness of CLIP while maintaining accuracy, and can be seamlessly integrated with other VLMs for further gains. These findings highlight the great potential of establishing an adversarially robust paradigm from CLIP, with implications extending to broader vision-language domains such as BioMedCLIP.

## 1 Introduction

With the rapid proliferation of image-text data and advances in self-supervised learning, vision-language models (VLMs) have attracted increasing attention from both academia and industry (Radford et al., 2021; Chen et al., 2023; Liu et al., 2025; Wang et al., 2025a; Li et al., 2024b). Among them, CLIP has demonstrated impressive zero-shot capabilities, effectively aligning images with descriptive text and enabling strong transfer across classification, retrieval, and diverse downstream tasks (Zhou et al., 2022a; Shin et al., 2022; Liu et al., 2024; Zhao et al., 2022; Zhang et al., 2023). However, recent studies reveal that even subtle, imperceptible perturbations can cause CLIP to misclassify, exposing a fundamental vulnerability shared by many neural networks (Radford et al., 2021). As foundation models are increasingly deployed in real-world applications, ensuring their adversarial robustness has become critical (Xing et al., 2025). This work investigates the robustness of CLIP and its derivatives under such perturbations.

CLIP, unlike conventional models with well-studied adversarial robustness, is a foundation model pre-trained on massive image–text pairs. It encodes broad real-world knowledge yet requires careful handling to preserve generalization, particularly under adversarial attacks (Zhou et al., 2022c;b). Since its pretraining demands large-scale data and substantial computational resources, most practitioners rely on open-source variants from a limited pool of models (Zhang et al., 2025), leaving CLIP-based applications especially exposed to adversarial risks. Recent studies further reveal that VLMs are highly susceptible to such perturbations, undermining their reliability in open-world deployment (Li et al., 2024a; Schlarmann et al., 2024; Malik et al., 2025).

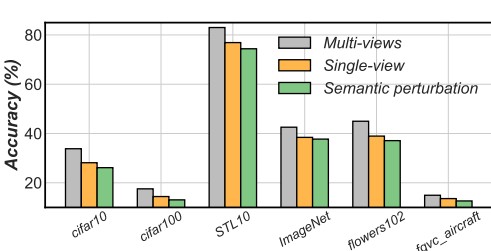

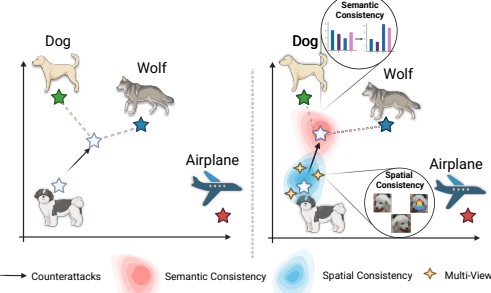

Figure 1: Analysis of Counterattack for Adversarial Robustness. Performance drops when reducing from two views to a single view, and degrades further under semantic perturbations.

Figure 2: During counterattack inference, embeddings tend to drift within the adversarial space and fall into hard-negative traps; SCC leverages cross-modal semantic and spatial consistency to push them away from hard samples and back toward the correct class space.

Research on CLIP's adversarial robustness is still nascent. A main line of work is training-based defenses, including adversarial fine-tuning (AFT) (Malik et al., 2025; Schlarmann et al., 2024) and adversarial prompt tuning (APT) (Shu et al., 2022; Zanella & Ben Ayed, 2024). AFT fine-tunes the visual encoder via a min–max game with dynamically generated adversarial images, yielding transferable zero-shot robustness but at high computational cost, reliance on labeled data, and a tendency to overfit the fine-tuning set, which degrades generalization on unseen distributions. APT instead adjusts learnable tokens in the text embedding space to align adversarial images, but similarly overfits to training data—boosting clean accuracy only on seen distributions while harming generalization (Yu et al., 2024). Another emerging line is test-time defense, which adapts models during inference without retraining. Recent works include R-TPT (Sheng et al., 2025), minimizing pointwise entropy with reliability-weighted ensembles, and Test-Time Counterattack (TTC) (Xing et al., 2025), leveraging CLIP's visual encoder to counter adversarial perturbations. While promising, both remain prone to semantic misalignment and unstable recovery under attacks.

Building on prior robustness studies, adversarial attacks often induce pseudo-stability, where perturbed images appear deceptively stable (Xing et al., 2025); thresholded counter-attacks mitigate this but still shift embeddings toward hard negatives and leave single-view corrections insufficient to suppress noise, as shown in Figure 1. Motivated by these observations, we propose Self-Calibrated Consistency (SCC), a simple yet effective test-time defense composed of two complementary components. *Semantic consistency*, which leverages soft pseudo-labels from counterattack warm-up and multi-view predictions to regularize cross-modal alignment and separate target embeddings from confusable negatives (Figure 2); and *Spatial consistency*, which enforces agreement among perturbed visual predictions and leverages augmented views to mitigate viewpoint fragility and stabilize feature calibration (Figure 2). Extensive experiments on 22 zero-shot benchmarks demonstrate that SCC consistently improves adversarial robustness while preserving clean accuracy, surpassing state-of-the-art test-time defenses. In summary, our main contributions are:

- This work identifies and provides analytical insights into three vulnerabilities in test-time defenses-semantic drift, view sensitivity, and hard-negative dominance-and proposes SCC, a framework that shifts the paradigm from unimodal defenses to cross-modal, multi-view self-corrective robustness.

- SCC unifies semantic and spatial consistency into a principled test-time defense: a cross-modal consistency constraint preserves alignment against hard negatives, while spatial consistency stabilizes perturbed views to mitigate viewpoint fragility, together forming a dual defense that delivers robust and generalizable zero-shot performance.

- SCC is a plug-and-play defense that boosts robustness without retraining, consistently outperforming prior test-time methods on 22 benchmarks and extending effectively to CLIP derivatives such as BioMedCLIP.

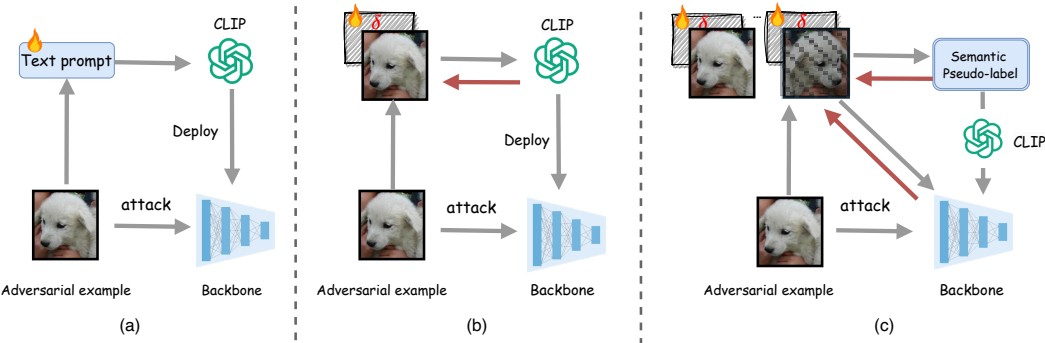

Figure 3: Test-time defense paradigms on CLIP. (a). R-TPT adapts text prompts online but still suffers from adversarial perturbations. (b). TTC repairs adversarial inputs via corrective perturbations, yet remains sensitive to view variance and hard negatives. (c). SCC enforces semantic and spatial consistency, yielding more stable recovery.

## 2 PRELIMINARIES AND RELATED WORK

Despite notable success, VLMs are highly vulnerable to adversarial perturbations: imperceptible changes crafted by PGD (Madry, 2018) or CW can flip predictions, and multimodal misalignment exacerbates this by shifting image embeddings toward hard negatives, causing semantic drift (Su et al., 2019; Moosavi-Dezfooli et al., 2017; Andriushchenko et al., 2020; Ilyas et al., 2018).

To address adversarial vulnerability in VLMs, several directions have been explored (Mao et al., 2023; Li et al., 2024a; Liang et al., 2024; Yu et al., 2023; Shu et al., 2022; Li et al., 2020; Zhang et al., 2022; Li et al., 2023; Huang et al.; Tong et al., 2025). AFT (Malik et al., 2025; Schlarmann et al., 2024) enhances robustness with adversarial examples but is costly, label-dependent, and overfits, hurting zero-shot generalization. APT (Yu et al., 2024; Wang et al., 2025b; Zhang et al., 2024) adjusts learnable tokens in the text space, yet also overfits, inflating clean accuracy only on seen data while degrading unseen performance. Test-time defenses, including R-TPT (Sheng et al., 2025) and TTC (Xing et al., 2025), adapt models without retraining but remain unstable and semantically misaligned under attacks (Shu et al., 2022; Zanella & Ben Ayed, 2024; Sui et al., 2025). Overall, existing methods either demand expensive retraining or fail to ensure semantic and stable predictions (Yu et al., 2024; Abdul Samadh et al., 2024), motivating our SCC (Figure 3).

**Problem formulation:** Given an image $x$ and a set of text prompts $\{t_k\}$, zero-shot classification in CLIP is performed by computing cosine similarities between the normalized image embedding $f_{\text{img}}(x)$ and text embeddings $g_{\text{text}}(t_k)$, followed by a softmax over classes: $p(y = k \mid x) = \frac{\exp(\tau \cdot \langle f_{\text{img}}(x), g_{\text{text}}(t_k) \rangle)}{\sum_j \exp(\tau \cdot \langle f_{\text{img}}(x), g_{\text{text}}(t_j) \rangle)}$, where $\tau$ denotes a learnable temperature parameter.

We consider CLIP, consisting of an image encoder $f_{\text{img}}(\cdot)$ and a text encoder $g_{\text{text}}(\cdot)$. Given an image $x$ and class prompts $\{t_k\}_{k=1}^K$, zero-shot prediction is

$$\hat{y} = \arg\max_k \langle f_{\text{img}}(x), g_{\text{text}}(t_k) \rangle, \tag{1}$$

In adversarial settings, an attacker perturbs $x$ within an $\ell_p$ ball of radius $\epsilon_a$ (perturbation budget), yielding $x^{\text{adv}} = x + \delta^{\text{atk}}$, $\|\delta^{\text{atk}}\|_p \le \epsilon_a$, To counteract this, our defense applies a corrective perturbation $\delta$ to recover alignment:

$$x^{\text{cnt}} = x^{\text{adv}} + \delta, \quad \|\delta\|_p \le \epsilon_d, \tag{2}$$

Here, $\delta$ is optimized at test time, and $\epsilon_d$ controls the maximum allowable perturbation magnitude.

## 3 METHODOLOGY

### 3.1 THE FINDINGS OF TEST-TIME COUNTERATTACK

To motivate our approach, we revisit TTC and identify three vulnerabilities. (1) *Semantic drift*: the repaired similarity $\cos(\hat{z}(x^{\text{cnt}}), \hat{t}_{y^\star})$, with $x^{\text{cnt}} = x^{\text{adv}} + \delta$, often fluctuates and can even shift toward

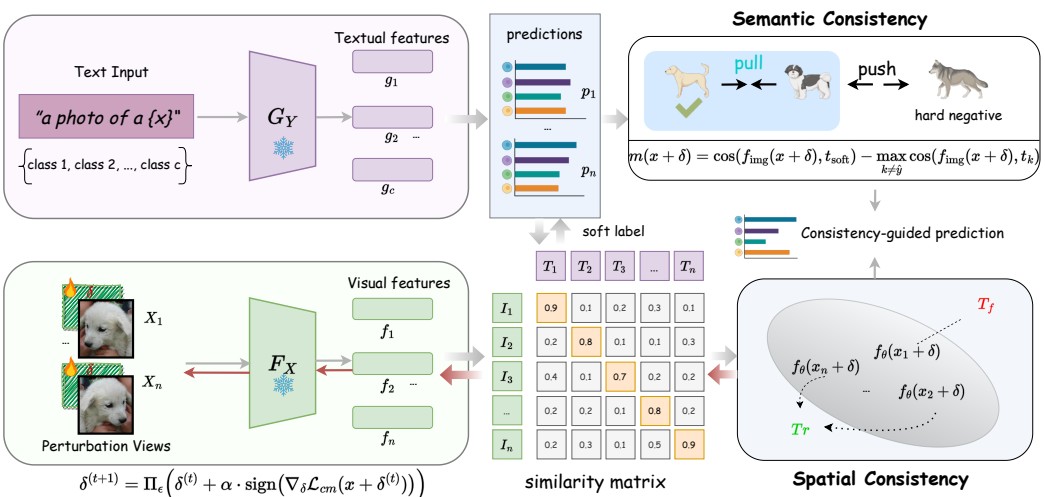

Figure 4: Pipeline of SCC: text and augmented views are encoded into features, multi-view embeddings are aggregated with averaging and combined with a short counterattack warm-up to yield stable soft pseudo-labels, which then guide cross-modal consistency optimization through the corrective perturbation $\delta$. $T_r$ denotes the correct class embedding (e.g., dog), while $T_f$ is an incorrect class embedding (e.g., wolf). Spatial consistency enforces perturbed views $f_\theta(x_i + \delta)$ to stay close to $T_r$ rather than drift toward $T_f$.

non-target texts under strong attacks ($\hat{z}$ denotes the cosine-similarity logit vector after counterattack, Figure 1). (2) *Hardest-competitor dominance*: misclassifications arise when the repaired embedding aligns closely with the strongest competitor $j^\star = \arg\max_{j \neq y^\star} \langle \hat{z}(x^{\text{cnt}}), \hat{t}_j \rangle$ (Figure 2). (3) *View sensitivity*: across semantics-preserving augmentations $\{v_i\}$ (horizontal flip or low-variance Gaussian noise), the repaired logit gaps exhibit high variance, e.g., $\text{Var}_i[\Delta^{(i)}]$ with $\Delta^{(i)} = z_{(1)}^{(i)} - z_{(2)}^{(i)}$, indicating inconsistent recovery (Figure 1). Together, these expose TTC's fragility in preserving cross-modal semantics and spatial stability, motivating a principled solution. We next formalize *semantic* (1-2) and *spatial fragility* (3), which underpin our SCC framework.

## 3.2 THE ANALYSIS OF SEMANTIC AND SPATIAL FRAGILITY

Let $\hat{f}(x) \in \mathbb{R}^d$ denote the $\ell_2$-normalized image embedding and $\{\hat{t}_k\}_{k=1}^K \subset \mathbb{R}^d$ the set of normalized text embeddings. The semantic margin of an image $x$ with ground-truth $y^\star$ is

$$m(x) = \langle \hat{f}(x), \hat{t}_{y^\star} \rangle - \max_{j \neq y^\star} \langle \hat{f}(x), \hat{t}_j \rangle. \tag{3}$$

Under adversarial perturbation $\delta$ with $\|\delta\|_p \leq \epsilon$, the margin becomes

$$m(x + \delta) = \langle \hat{f}(x + \delta), \hat{t}_{y^\star} \rangle - \max_{j \neq y^\star} \langle \hat{f}(x + \delta), \hat{t}_j \rangle, \tag{4}$$

which often collapses or even turns negative, indicating a shift toward hard negatives. This fragility manifests in three forms:

**Prediction noise.** For adversarial inputs $x^{\text{adv}}$, the single-view distribution $\tilde{q}(y \mid x^{\text{adv}})$ deviates from the ground-truth $p^\star$, introducing

$$\text{Bias} = \|\mathbb{E}[\tilde{q}] - p^\star\|_1, \quad \text{Var} = \sum_k \text{Var}[\tilde{q}_k],$$

which reduce expected alignment $\mathbb{E}[\langle \hat{f}(x^{\text{adv}}), t_{y^\star} \rangle]$.

**Hard-negative alignment.** Counterattacks often pull embeddings toward the hardest negative

$$j^\star = \arg\max_{j \neq y^\star} \langle \hat{f}(x^{\text{adv}}), t_j \rangle,$$

causing the margin $m(x^{\text{adv}}) = \langle \hat{f}(x^{\text{adv}}), t_{y^\star} \rangle - \langle \hat{f}(x^{\text{adv}}), t_{j^\star} \rangle$ to collapse.

**View sensitivity.** Let $A$ be a distribution of semantics-preserving augmentations. Across $N$ sampled views $\{v_i\}$, logits $z^{(i)} = \langle \hat{f}(v_i(x^{\text{adv}})), t_k \rangle$ exhibit high variance $\text{Var}_i[z^{(i)}]$, and the hardest negative $j^\star(i)$ may differ by view. Consequently, PGD updates guided by $\nabla_\delta z_{j^\star(i)}^{(i)}$ are inconsistent, yielding large gradient variance $\text{Var}_i[\nabla_\delta \mathcal{L}(z^{(i)})]$ and unstable recovery.

Together, these effects define *semantic and spatial fragility*, underscoring the difficulty of preserving cross-modal alignment under adversarial perturbations.

### 3.3 MITIGATING SEMANTIC FRAGILITY VIA SEMANTIC CONSISTENCY

**Cross-modal consistency.** Given an adversarial input $x^{\text{adv}}$, the defense applies a counter-perturbation $\delta$ by optimizing a margin objective that encourages alignment with a soft semantic anchor while repelling hard negatives, as shown in Figure 4:

$$\mathcal{L}_{cm}(x^{\text{adv}}, \delta) = \cos(f_{\text{img}}(x^{\text{adv}} + \delta), t_{\text{soft}}) - \max_{k \neq \hat{y}} \cos(f_{\text{img}}(x^{\text{adv}} + \delta), t_k), \tag{5}$$

where $\hat{y} = \arg\max_k \cos(f_{\text{img}}(x^w), t_k)$ is pseudo-label predicted from the warm-up embedding $x^w$.

**Soft prototype construction.** To stabilize $t_{\text{soft}}$, we perform a short TTC warm-up (A.1) on $x^{\text{adv}}$ to obtain $x^w$, then generate $N$ augmented views $\{v_i(x^w)\}_{i=1}^N$. The view-wise predictions $\{q^{(i)}\}$ are averaged and sharpened with temperature $T < 1$:

$$q_k^{\text{sharp}} = \frac{\left(\frac{1}{N} \sum_{i=1}^N q_k^{(i)}\right)^{1/T}}{\sum_j \left(\frac{1}{N} \sum_{i=1}^N q_j^{(i)}\right)^{1/T}}, \tag{6}$$

and the soft prototype is defined as

$$t_{\text{soft}} = \sum_k q_k^{\text{sharp}} t_k, \tag{7}$$

which acts as the semantic anchor in $\mathcal{L}_{cm}$. The detailed SCC procedure is provided in algorithm 1.

**Proposition 1** (Hard-negative repulsion). *Let $x + \delta$ denote the counter-perturbed input during optimization. Optimizing $\mathcal{L}_{cm}$ by PGD ascent increases the semantic margin*

$$m(x + \delta) = \cos(f_{img}(x + \delta), t_{soft}) - \max_{k \neq \hat{y}} \cos(f_{img}(x + \delta), t_k)$$

*monotonically (up to $\mathcal{O}(\alpha^2)$), thereby preventing drift toward confusable negatives. See proof in Appendix.*

**Iterative counter-attack.** Corrective perturbations are computed as

$$\delta^{(t+1)} = \Pi_\epsilon \left( \delta^{(t)} + \alpha \cdot \text{sign}\left( \nabla_\delta \mathcal{L}_{cm}(x + \delta^{(t)}) \right) \right), \tag{8}$$

where $\Pi_\epsilon$ projects onto the $\ell_p$ ball of radius $\epsilon$ and $\alpha$ is the step size. A step-weighted fusion is applied across PGD iterations, where intermediate perturbations $\delta^{(t)}$ are aggregated with weights proportional to their step index, yielding a smoother final correction.

### 3.4 MITIGATING SPATIAL FRAGILITY VIA SPATIAL CONSISTENCY

**Multi-view self-consistency.** To stabilize predictions, we aggregate $L$ *augmented views* (Sheng et al., 2025) of the same input. Let $z^{(i)}$ be the logits of view $i$, then

$$\bar{z} = \frac{1}{L} \sum_{i=1}^L z^{(i)}, \qquad \bar{q} = \text{softmax}(\bar{z}), \qquad t_{\text{soft}} = \sum_k \bar{q}_k t_k.$$

While each view may yield noisy predictions under adversarial perturbations, their aggregation reduces variance and yields a more reliable semantic anchor.

**Proposition 2** (Variance reduction). *If $\{q^{(i)}\}$ are i.i.d. with covariance $\Sigma$, then $Cov(\bar{q}) = \frac{1}{L}\Sigma$, showing variance shrinks as $1/L$ and $t_{soft}$ becomes more stable.*

**Remark 1.** *Temperature sharpening ($T < 1$) further amplifies dominant classes: $q_k(T) = \frac{\bar{q}_k^{1/T}}{\sum_j \bar{q}_j^{1/T}}$, which enlarges semantic margins by suppressing noisy tail classes.*

Table 1: Classification accuracy (%) on clean images (Acc.) and adversarial images (Rob.) under 10-step PGD attack ($\epsilon_a = 1/255$) across 16 datasets. The threat model assumes full access to model weights and gradients. We compare our paradigm against test-time defenses adapted from prior adversarial robustness studies, and include fine-tuned models as references. The last column shows the gains of SCC over the CLIP.

| Dataset | Metric | CLIP | Adversarial Finetuning | | | | Test-time Defence | | | | | Δ |
|---|---|---|---|---|---|---|---|---|---|---|---|---|
| | | | CLIP-FT | TeCoA | PMG-AFT | FARE | RN | Anti-adv | HD | TTC | SCC(ours) | |
| CIFAR10 | Rob. | 0.74 | 3.34 | 33.61 | 40.66 | 19.65 | 2.01 | 12.39 | 17.22 | 28.75 | **59.18** | +58.44 |
| | Acc. | 85.12 | 84.90 | 64.61 | 70.69 | 74.44 | 81.18 | 83.52 | 78.23 | 81.18 | 82.24 | -2.88 |
| CIFAR100 | Rob. | 0.26 | 0.90 | 18.95 | 22.52 | 11.40 | 0.67 | 5.73 | 3.86 | 14.31 | **32.09** | +31.83 |
| | Acc. | 57.14 | 59.51 | 35.96 | 40.32 | 46.67 | 56.34 | 53.95 | 52.86 | 56.34 | 55.21 | -1.93 |
| STL10 | Rob. | 11.00 | 12.73 | 70.08 | 73.08 | 59.06 | 16.23 | 37.42 | 39.02 | 76.70 | **90.50** | +79.50 |
| | Acc. | 96.40 | 94.49 | 87.40 | 88.56 | 91.72 | 95.85 | 95.45 | 89.50 | 95.85 | 95.62 | -0.78 |
| ImageNet | Rob. | 1.15 | 0.93 | 18.89 | 21.43 | 14.00 | 1.77 | 8.67 | 6.63 | 38.41 | **49.77** | +48.62 |
| | Acc. | 59.69 | 54.24 | 34.89 | 36.12 | 48.79 | 59.34 | 54.27 | 54.54 | 49.39 | 56.03 | -3.66 |
| Caltech101 | Rob. | 14.67 | 14.21 | 55.51 | 61.08 | 50.74 | 18.90 | 34.81 | 31.53 | 65.78 | **77.25** | +62.58 |
| | Acc. | 85.66 | 83.63 | 71.68 | 75.45 | 80.95 | 86.61 | 84.02 | 82.33 | 86.53 | 86.44 | +0.78 |
| Caltech256 | Rob. | 8.47 | 6.76 | 43.19 | 45.91 | 38.79 | 11.33 | 25.36 | 23.48 | 60.11 | **72.88** | +64.41 |
| | Acc. | 81.72 | 78.53 | 61.14 | 62.24 | 73.32 | 81.25 | 79.38 | 79.12 | 79.66 | 81.16 | -0.56 |
| OxfordPets | Rob. | 1.04 | 2.10 | 38.35 | 41.18 | 31.07 | 1.86 | 20.42 | 12.04 | 57.87 | **76.67** | +75.63 |
| | Acc. | 87.44 | 84.14 | 62.12 | 65.88 | 79.37 | 87.41 | 80.62 | 80.91 | 83.35 | 86.48 | -0.96 |
| Flowers102 | Rob. | 1.14 | 0.54 | 21.94 | 23.43 | 17.14 | 1.52 | 7.16 | 7.29 | 39.14 | **54.59** | +53.45 |
| | Acc. | 65.46 | 53.37 | 36.80 | 37.00 | 47.98 | 64.62 | 62.66 | 58.22 | 64.16 | 64.16 | -1.30 |
| FGVC-Aircraft | Rob. | 0.00 | 0.00 | 2.49 | 2.22 | 1.35 | 0.00 | 1.27 | 1.26 | 13.77 | **17.40** | +17.40 |
| | Acc. | 20.10 | 14.04 | 5.31 | 5.55 | 10.86 | 19.25 | 15.88 | 16.36 | 18.00 | 17.61 | -2.49 |
| StanfordCars | Rob. | 0.02 | 0.06 | 8.76 | 11.65 | 6.75 | 0.16 | 4.40 | 2.71 | 33.01 | **43.24** | +43.22 |
| | Acc. | 52.02 | 42.11 | 20.91 | 25.44 | 38.68 | 52.14 | 36.21 | 44.28 | 48.16 | 51.19 | -0.83 |
| SUN397 | Rob. | 1.14 | 0.94 | 19.39 | 22.58 | 14.91 | 1.72 | 8.05 | 6.40 | 41.52 | **53.27** | +52.13 |
| | Acc. | 58.50 | 55.73 | 36.69 | 37.98 | 52.42 | 59.69 | 56.00 | 53.17 | 55.13 | 58.25 | -0.25 |
| Country211 | Rob. | 0.04 | 0.03 | 1.78 | 2.12 | 0.85 | 0.06 | 0.67 | 0.47 | 7.09 | **9.41** | +9.37 |
| | Acc. | 15.25 | 12.07 | 4.75 | 4.64 | 9.26 | 14.80 | 11.58 | 11.72 | 13.08 | 13.36 | -1.89 |
| Food101 | Rob. | 0.70 | 0.42 | 13.90 | 18.57 | 11.65 | 1.20 | 13.12 | 8.03 | 57.84 | **65.39** | +64.69 |
| | Acc. | 83.88 | 64.86 | 29.98 | 36.61 | 55.31 | 83.44 | 75.81 | 80.30 | 82.18 | 82.13 | -1.75 |
| EuroSAT | Rob. | 0.03 | 0.04 | 11.96 | 12.60 | 10.67 | 0.15 | 2.15 | 4.57 | 12.19 | **20.64** | +20.61 |
| | Acc. | 42.59 | 27.64 | 16.58 | 18.53 | 21.88 | 53.24 | 36.78 | 39.08 | 53.24 | 41.69 | -0.90 |
| DTD | Rob. | 2.98 | 2.39 | 17.61 | 14.95 | 15.64 | 3.71 | 5.62 | 11.63 | 27.32 | **34.57** | +31.59 |
| | Acc. | 40.64 | 36.49 | 25.16 | 21.76 | 32.07 | 37.96 | 38.92 | 34.89 | 36.98 | 37.34 | -3.30 |
| PCAM | Rob. | 0.08 | 1.11 | 48.24 | 46.18 | 16.23 | 0.41 | 4.97 | 44.74 | 52.85 | **69.99** | +69.91 |
| | Acc. | 52.02 | 47.21 | 49.96 | 50.03 | 52.54 | 52.73 | 52.49 | 50.38 | 52.73 | 54.41 | +2.39 |
| Avg. | Rob. | 2.70 | 2.91 | 26.54 | 28.76 | 20.00 | 3.86 | 12.01 | 13.81 | 39.17 | **51.68** | +48.98 |
| | Acc. | 61.51 | 55.80 | 40.25 | 42.30 | 51.02 | 61.61 | 57.35 | 56.62 | 59.75 | 60.21 | -1.30 |

**Confidence weighting.** We assign each sample a confidence $w(x) \in [0, 1]$ (based on margin or entropy), so that high-confidence predictions dominate optimization, while noisy pseudo-labels are down-weighted. This weighting mitigates error propagation and stabilizes semantic alignment.

**Spatial counterattacks.** For each input $x$, we first obtain a single counter-perturbation $\delta$ by the TTC inner loop (e.g., PGD-like ascent) under $\|\delta\|_p \leq \epsilon$. We then form $L$ semantics-preserving views of the corrected image $x + \delta$ via horizontal flip and low-variance Gaussian pixel noise:

$$\mathcal{V}(x + \delta) = \{ v_i(x + \delta) \}_{i=1}^{L}, \quad v_i(\cdot) = \text{flip}_h^{\mathbf{1}_{i \text{ odd}}}(\cdot) + \eta_i, \ \eta_i \sim \mathcal{N}(0, (\sigma/255)^2 I).$$

Let $z^{(i)}$ be the logits of view $i$. We aggregate by averaging logits and then softmax:

$$\bar{z} = \frac{1}{L} \sum_{i=1}^{L} z^{(i)}, \qquad \hat{p} = \text{softmax}(\bar{z}), \qquad \hat{y} = \arg\max_k \bar{z}_k.$$

**Remark 2** (Optimization coupling). *Unlike pure test-time ensembling, all augmented views share a common corrective perturbation $\delta$, which is optimized jointly in the TTC loop. This coupling enforces spatial consistency while repairing adversarial effects.*

**Proposition 3** (Suppression of spurious negatives). *For averaged logits,*

$$\max_{j \neq y^\star} \bar{z}_j \ \leq \ \frac{1}{L} \sum_i \max_{j \neq y^\star} z_j^{(i)},$$

*so aggregation suppresses view-dependent hardest negatives and stabilizes TTC updates. See proof in Appendix.*

Table 2: Adversarial (Rob.) and clean (Acc.) accuracy (%) on 16 datasets under PGD-10 ($\epsilon_a = 4/255$). Superscripts denote fine-tuning budgets. The last row shows gains over CLIP.

| (%) | CLIP | CLIP-FT | TeCoA[1] | TeCoA[4] | PMG-AFT[1] | PMG-AFT[4] | FARE[1] | FARE[4] | RN | Anti-adv | HD | TTC | SCC(ours) | $\Delta$ |
|------|------|---------|----------|----------|------------|------------|---------|---------|------|----------|-------|-------|-----------|--------|
| Rob. | 0.09 | 0.96 | 6.51 | 10.03 | 7.03 | 10.70 | 1.50 | 3.67 | 0.06 | 0.53 | 1.19 | 20.63 | **27.88** | +27.79 |
| Acc. | 61.51 | 55.80 | 40.25 | 35.57 | 42.30 | 37.58 | 51.02 | 46.17 | 61.61 | 57.32 | 56.62 | 55.99 | **60.42** | -1.09 |

**Objective.** The SCC optimization couples cross-modal semantics with spatial stability:

$$\max_{\|\delta\| \leq \epsilon} \quad \lambda_{cm} \mathcal{L}_{cm}(x, \delta) + \|f_{\text{img}}(x + \delta) - f_{\text{img}}(x)\|_2^2, \tag{9}$$

where the second term follows TTC in promoting feature deviation to escape pseudo-stability.

**Remark 3.** *This unifies semantic alignment and spatial consistency into a defense objective.*

## 4 EXPERIMENTS

### 4.1 EXPERIMENTAL SETUP

**Datasets and Baselines:** Building on prior studies of CLIP's adversarial robustness (Mao et al., 2023; Xing et al., 2025), we evaluate on 16 public datasets spanning diverse visual domains: generic object recognition (CIFAR10 (Krizhevsky et al., 2012), CIFAR100 (Krizhevsky et al., 2012), STL10 (Coates et al., 2011), ImageNet (Deng et al., 2009), Caltech101 (Fei-Fei et al., 2006), Caltech256 (Griffin & Perona, 2008)), fine-grained recognition (OxfordPets (Parkhi et al., 2012), Flowers102 (Nilsback & Zisserman, 2008), Food101 (Bossard et al., 2014), StanfordCars (Krause et al., 2013)), scene recognition (SUN397 (Xiao et al., 2010), Country211 (Radford et al., 2021)), and specialized domains (FGVCAircraft (Maji et al., 2013), EuroSAT (Helber et al., 2019), DTD (Cimpoi et al., 2014), PCAM (Bejnordi et al., 2017)). Comprehensive evaluation further includes experiments on 6 medical datasets such as BUSI (Al-Dhabyani et al., 2020), BTMRI (Koleilat et al., 2025), CHMNIST (Kather et al., 2016), COVID-19 (Tahir et al., 2021), DermaMNIST (Codella et al., 2019), and KneeXray (Chen, 2018).

We implemented several baselines for comparison. Test-time defenses include Test-time Counterattack (TTC) (Xing et al., 2025), following the original setup, Anti-Adversarial (Alfarra et al., 2022) (adapted to CLIP by maximizing image–text similarity), Hedging Defense (HD) (Wu et al., 2021) (minimizing cross-entropy across all classes), and RN, which perturbs inputs with random noise of the same strength as $\epsilon$ (Xing et al., 2025). As reference, we evaluated adversarial fine-tuning methods—TeCoA (Mao et al., 2023), PMG-AFT (Wang et al., 2024), FARE (Schlarmann et al., 2024)—and a clean fine-tuned CLIP (CLIP-FT) on TinyImageNet, using 2-step PGD ($\alpha = 1/255$, $\epsilon_a = 1/255$) and learning rate $5 \times 10^{-5}$, then transferring the models to 16 downstream datasets.

**Implementation:** We adopt CLIP ViT-B/32 as the backbone (Radford et al., 2021) and BioMed-CLIP (Zhang et al., 2025) for medical tasks, using the handcrafted prompt templates from CLIP. Counterattack budget are set to $\epsilon = 4/255$, and 2 steps (Xing et al., 2025). For the semantic consistency, $\lambda_{cm} = 4$ and temperature $T = 0.5$ (selected via grid search). For the spatial consistency, we use $L = 2$ augmented views with noise $\sigma = 6$ (tuned by search). We evaluate against white-box and adaptive attacks, including $PGD\text{-}\ell_\infty$ and CW (Xing et al., 2025). By default, we report top-1 accuracy on both clean and adversarial examples. Counterattack parameters follow (Xing et al., 2025). The batch size is set to 256. We conducted all experiments on NVIDIA H20 GPUs.

### 4.2 MAIN RESULTS

We evaluate robustness under an attack budget of $\epsilon_a = 1/255$, following prior CLIP robustness studies (Xing et al., 2025). All baselines are tested on 16 datasets with 10-step PGD attacks, assuming full access to model weights and gradients but no access to test-time operations. As shown in Table 1, adversarially fine-tuned models (TeCoA, PMG-AFT, FARE, CLIP-FT) suffer from severe overfitting: while robust accuracy improves on training-like datasets, clean accuracy drops significantly across downstream tasks. Among test-time defenses, Anti-Adversarial and HD yield only marginal gains, while RN fails to provide robustness even with perturbations much larger than $\epsilon_a$. TTC delivers noticeable gains but falls significantly short of SCC. In contrast, our SCC achieves

Table 3: CLIP and BioMedCLIP Robustness on Medical Benchmarks ($\epsilon_a = 1/255$).

| Backbone | | BUSI | | BTMRI | | CHMNIST | | COVID_19 | | DermaMNIST | | KneeXray | | Avg. | |
|---|---|---|---|---|---|---|---|---|---|---|---|---|---|---|---|---|
| | | Rob. | Acc. | Rob. | Acc. | Rob. | Acc. | Rob. | Acc. | Rob. | Acc. | Rob. | Acc. | Rob. | Acc. |
| CLIP | CLIP | 0.00 | 47.18 | 0.00 | 25.60 | 0.00 | 21.32 | 0.13 | 6.39 | 0.02 | 19.76 | 0.00 | 13.74 | 0.02 | 22.33 |
| | TTC | 11.67 | 42.05 | 8.93 | 27.84 | 2.20 | 19.64 | **7.51** | 9.11 | 6.28 | 20.68 | 7.68 | 14.32 | 7.38 | 22.27 |
| | SCC | **23.85** | 42.31 | **16.19** | 27.78 | **9.12** | 17.26 | 7.30 | 7.05 | **12.40** | 20.48 | **11.08** | 13.12 | **13.32** | 21.33 |
| BioMedCLIP | BioMedCLIP | 0.00 | 40.38 | 0.49 | 60.33 | 0.00 | 32.62 | 0.02 | 72.53 | 0.00 | 35.62 | 0.00 | 27.92 | 0.08 | 44.90 |
| | TTC | 7.95 | 37.05 | 22.20 | 53.08 | 2.80 | 29.62 | 18.36 | 57.20 | 4.91 | 24.58 | 7.51 | 34.10 | 10.62 | 39.27 |
| | SCC | **31.92** | 40.26 | **48.93** | 59.72 | **16.56** | 31.24 | **57.58** | 68.95 | **20.64** | 32.51 | **28.35** | 35.07 | **34.00** | 44.63 |

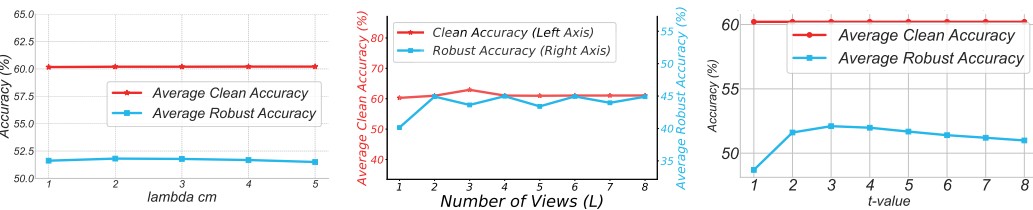

Figure 5: Sensitivity of SCC to $\lambda_{cm}$, number of views $L$, and effect of the temperature $t$ in soft-label sharpening (The $t$-axis in the plot is scaled by $\times 10$). A moderate $t$ yields the best trade-off.

consistent improvements: the average robust accuracy rises from 2.70% (CLIP) and 39.17% (TTC) to 51.68%, a substantial gain of +48.98% over vanilla CLIP and +12.51% over TTC, with clean accuracy only slightly reduced ($-1.30\%$). These results highlight SCC as an test-time defense that delivers strong and stable adversarial robustness without sacrificing clean performance.

We further evaluate robustness under a stronger attack budget $\epsilon_a = 4/255$. For the stronger-budget setting, we increase counterattack iterations to 5 while keeping all other hyperparameters fixed; adversarial fine-tuning baselines are trained with the same perturbation budget. As shown in Table 2 and A.4, robust accuracy of all models drops significantly under stronger attacks. Anti-Adversarial and HD almost lose robustness in this setting, while TTC provides moderate protection but suffers from high variance across datasets. In contrast, our SCC achieves stable improvements: average robust accuracy rises to 27.88%, outperforming TTC by +7.25% and vanilla CLIP by +27.79%, with only a negligible clean accuracy drop (–1.09%). These results demonstrate that SCC remains effective even under high-budget adversarial perturbations, highlighting its robustness and generalization. Per-dataset results are provided in the Appendix. We further evaluate SCC under CW attacks (Carlini & Wagner, 2017), with results deferred to the Appendix due to space limits (A.3).

Adversarial robustness in the medical domain is particularly challenging: as shown in Table 3, BioMedCLIP nearly collapses under $\epsilon_a = 1/255$ attacks, with average adversarial accuracy close to 0% (Koleilat et al., 2025). TTC alleviates this issue by introducing counterattacks, improving robustness to 10.62% on average. Our SCC further restores robustness substantially, reaching 34.00% on BioMedCLIP (a +23.38% improvement over TTC) while maintaining clean accuracy (44.63%). On CLIP, SCC also consistently outperforms TTC across six medical datasets, improving robustness by +5.94% on average. These results demonstrate that SCC not only generalizes to domain-specific models like BioMedCLIP but also provides a plug-and-play defense that stabilizes zero-shot medical prediction under adversarial perturbations.

### 4.3 ABLATION STUDIES

**Effect of self-calibrated consistency:** Figure 6 and Table 5 ablate SCC's two modules on individual datasets and averaged over 16 datasets. As shown in Figure 6, retaining both semantic (Sec) and spatial (Spa) consistency yields the best performance across all datasets, while removing either leads to sharp drops, and removing both results in the lowest accuracy. Concretely, with only semantic consistency, robustness is 39.76% (clean 60.28%); with only spatial consistency, 48.01% (clean 59.76%). Combining both raises robustness to 51.68% (clean 60.21%), and while removing both modules drops the performance to -12.5% robustness and -0.48% clean accuracy. These results

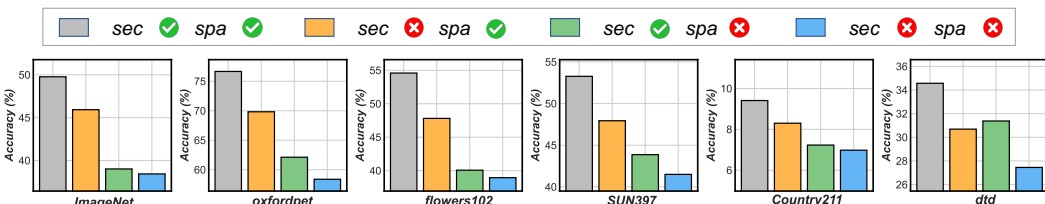

Figure 6: Ablation results of semantic consistency (sec) and spatial consistency (spa) across datasets. Removing either component degrades performance, while combining both yields the best robustness and accuracy.

confirm the complementarity of the two modules: each provides modest gains alone, but together they deliver substantial robustness improvements while maintaining clean accuracy.

**Analysis of hyperparameter sensitivity.** We conducted grid searches over the key hyperparameters of SCC. As shown in Figure 5, the cross-modal regularization weight $\lambda_{cm} \in [1, 5]$ has little effect on clean accuracy and only mild impact on robustness, with a small peak around the mid-range; we adopt $\lambda_{cm} = 4$ for stability. The number of views $L$ strongly influences robustness, which increases sharply from $L = 1$ to $L = 2$–$4$ before saturating; we set $L = 2$ for a balance of accuracy and efficiency. The temperature $T$ used in soft-label sharpening (Figure 5) also affects robustness, with $T = 0.5$ yielding the best trade-off. Finally, the noise scale $\sigma$ (Figure 9) steadily boosts robustness until saturation, at the cost of a slight clean accuracy drop; we adopt $\sigma = 6$. Overall, SCC is not overly sensitive to hyperparameter choices, and the selected defaults yield strong robustness gains with minimal accuracy loss.

### 4.4 VISUALIZATION AND EFFICIENCY ANALYSIS.

Figure 7 illustrates the effect of SCC on CIFAR-10 under adversarial attacks. In panel (a), the distribution of maximum soft-label probabilities shows that, compared to the adversarial case (red), SCC (blue) shifts the distribution closer to clean samples (green), indicating better calibration and reduced over-confidence. Panels (b) and (c) compare confusion matrices: without SCC (b), adversarial perturbations induce widespread misclassifications, whereas with SCC (c), diagonal dominance is largely restored, confirming improved accuracy and stability across categories. In terms of efficiency, Table 4 shows that, unlike R-TPT which requires many view transformations, both TTC and SCC achieve much lower inference overhead. Notably, SCC incurs only an additional 0.0005s per image compared to TTC, yet delivers a +7.2% gain in robustness. This demonstrates SCC's clear superiority in achieving a favorable trade-off between robustness and efficiency.

| Method | Stage | Time | Rob. |
|---|---|---|---|
| R-TPT (64 views) | Test time | 0.37s/img | 32.8 |
| TTC | Test time | 0.012s/img | 27.4 |
| SCC (ours) | Test time | 0.0125s/img | **34.6** |

Table 4: Running time and adversarial accuracies (%) of methods against adversarial attack on DTD dataset.

| Semantic Consistency | Spatial Consistency | Rob. | Acc. |
|---|---|---|---|
| | | 39.18 | 59.73 |
| ✓ | | 39.76 | 60.28 |
| | ✓ | 48.01 | 59.76 |
| ✓ | ✓ | **51.68** | 60.21 |

Table 5: Ablation of semantic and spatial consistency across 16 datasets.

## 5 CONCLUSION

In this paper, we presented SCC, a test-time defense that strengthens the adversarial robustness of vision–language models in the zero-shot setting. SCC unifies two complementary components: semantic consistency, which resists cross-modal drift by repelling hard negatives, and spatial consistency, which stabilizes predictions through multi-view augmentation and correction. Extensive experiments across 22 benchmarks, including the domain-specific BioMedCLIP model, show that SCC yields consistent gains in robustness with minimal loss of clean accuracy. Our results demonstrate that SCC offers a simple and effective way to enhance the reliability of VLMs across both general-purpose and safety-critical domains.

ETHICS STATEMENT

This work uses only publicly available datasets without personal or sensitive information. By improving adversarial robustness of vision–language models, SCC aims to enhance reliability in both general and medical applications.

REPRODUCIBILITY STATEMENT

We provide implementation details in Experments and Appendix, including algorithmic descriptions, and hyperparameters. All experiments are conducted on publicly available datasets, and our code with scripts for reproducing results will be released upon publication.

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

# A APPENDIX

## A.1 IMPLEMENTATION DETAIL

For the counterattack analysis (Figure 1), we compare three settings. In the *multi-view* case, two augmented views (horizontal flip) are used to construct predictions. The *single-view* case reduces this to one view, removing variance reduction. For the *semantic perturbation* case, we randomly insert additional words into the text prompts, which distorts cross-modal alignment. Results in Figure 1 show that reducing to a single view significantly decreases robustness, and adding random semantic perturbations further degrades performance.

We perform a short TTC warm-up on each adversarial input $x^{\text{adv}}$ using PGD-like steps ($\epsilon = 4/255$, $\alpha = 1/255$), optimizing only the feature-deviation term to avoid label bias. Instead of early stopping, perturbations from all steps are fused by a $\tau$-threshold weighting scheme, yielding a stabilized

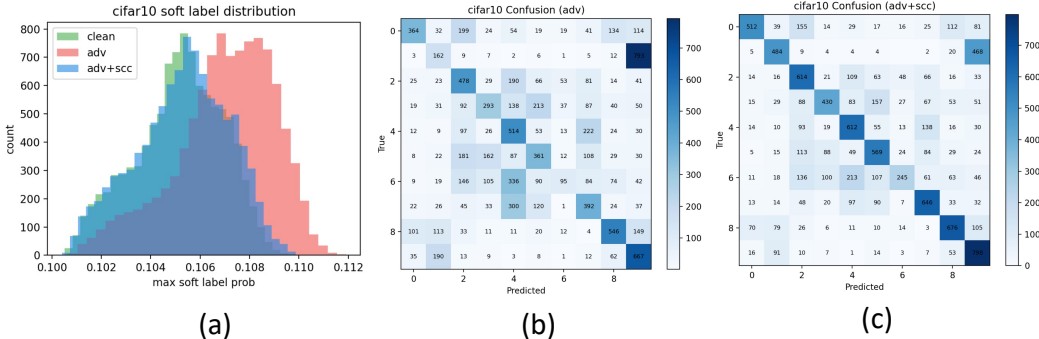

Figure 7: (a) Distribution of maximum soft label probabilities for clean, adversarial, and adversarial+SCC samples on CIFAR-10. SCC shifts the distribution toward clean probability. (b) Confusion matrix for adversarial samples, showing increased misclassification. (c) Confusion matrix for adversarial samples with SCC, demonstrating improved classification accuracy and reduced confusion

initialization $x^w$. On $x^w$, $N$ lightweight augmented views Sheng et al. (2025) (flip + Gaussian noise with $\sigma = 6/255$) are generated, their logits averaged before softmax, and the sharpened distribution ($T = 0.5$) used to construct the soft prototype $t_{\text{soft}}$, which serves as the semantic anchor in subsequent optimization.

## A.2 PERFORMANCE OF SCC GUIDED BY CLEAN IMAGE PREDICTIONS

Table 15 reports results when SCC is guided by predictions on clean images. Under this setting, SCC achieves an average robust accuracy of 53.62% with clean accuracy of 61.78%, showing a +50.92% improvement over vanilla CLIP. However, this requires access to clean-image predictions at inference, which is impractical in real-world deployment. By contrast, our pseudo-labeling strategy achieves 51.68% robust accuracy, closely approaching the clean-prediction upper bound while remaining label-free and deployable.

## A.3 ROBUSTNESS UNDER CW ATTACKS

Following prior CLIP robustness studies, we evaluate under a 10-step CW attack (Carlini & Wagner, 2017) with budget $\epsilon_a = 1/255$ across 16 datasets (white-box access to weights/gradients). As shown in Table 16, SCC attains the highest average robust accuracy, 49.42%, improving over vanilla CLIP by +45.88% and over the strongest test-time baseline (TTC) by large margins, while keeping clean accuracy essentially unchanged (60.21%, -1.30%). RN and TTE preserve clean accuracy (they do not counter-perturb inputs) but offer limited or unstable robustness. Anti-Adversarial and HD, which optimize targeted perturbations, yield low robust accuracy and further reduce clean performance. Adversarially fine-tuned models increase robustness on some datasets but at a substantial clean-accuracy cost. Overall, SCC consistently delivers the best robustness–accuracy trade-off under CW, indicating that inference-time self-calibrated consistency generalizes beyond PGD to stronger optimization-based attacks.

## A.4 ANALYSIS OF ROBUSTNESS (UNDER $\epsilon_a = 4/255$)

Table 17 summarizes robustness under a stronger 10-step PGD attack with budget $\epsilon_a = 4/255$ across 16 datasets. We observe that Anti-Adversarial and HD almost collapse under this setting, offering negligible robustness. RN maintain high clean accuracy, as they do not introduce counter-perturbations, but RN provides no robustness and TTE exhibits highly unstable gains, as reflected by large standard deviations across runs. By contrast, SCC consistently improves robustness across all datasets, achieving an average robust accuracy of 27.88%, a gain of +27.79% over vanilla CLIP, while keeping clean accuracy largely intact (60.42%, $-1.09\%$). To further strengthen counterattacks under this high-budget regime, we increase the iteration number to $N = 5$ for TTC. Although this slightly reduces clean accuracy by 5.52 points compared to CLIP, the substantial robustness

Table 6: Robust accuracy (%) under $\epsilon = 6/255$ PGD attacks.

| Method | CIFAR10 | CIFAR100 | STL10 | ImageNet | Caltech101 | Caltech256 | OxfordPet | Flowers102 | FGVC-Aircraft | StanfordCars | SUN397 | Country211 | Food101 | EuroSAT | DTD | PCAM | Avg. |
|--------|---------|----------|-------|----------|------------|------------|-----------|------------|---------------|--------------|--------|------------|---------|---------|------|------|------|
| SCC | 25.90 | 8.55 | 52.90 | 11.25 | 43.04 | 31.76 | 20.47 | 11.03 | 3.45 | 7.76 | 11.21 | 1.34 | 14.02 | 7.08 | 10.37 | 36.29 | **18.53** |
| TTC | 22.60 | 6.39 | 39.50 | 6.69 | 27.67 | 18.06 | 13.74 | 6.59 | 3.48 | 5.61 | 6.69 | 1.31 | 9.82 | 11.31 | 6.97 | 41.16 | 14.22 |

Table 7: Robust accuracy (%) under $\epsilon = 8/255$ PGD attacks.

| Method | CIFAR10 | CIFAR100 | STL10 | ImageNet | Caltech101 | Caltech256 | OxfordPet | Flowers102 | FGVC-Aircraft | StanfordCars | SUN397 | Country211 | Food101 | EuroSAT | DTD | Avg. |
|--------|---------|----------|-------|----------|------------|------------|-----------|------------|---------------|--------------|--------|------------|---------|---------|------|------|
| SCC | 22.46 | 7.81 | 48.11 | 10.01 | 39.62 | 28.60 | 15.07 | 9.22 | 3.39 | 6.07 | 10.25 | 1.36 | 12.59 | 5.76 | 10.96 | **15.42** |
| TTC | 25.43 | 6.20 | 40.15 | 5.73 | 23.60 | 14.16 | 15.07 | 6.81 | 5.07 | 5.57 | 6.34 | 1.55 | 9.90 | 13.85 | 8.72 | 12.54 |

gains justify the trade-off. Overall, these results confirm that SCC maintains stable and significant robustness improvements even under stronger adversarial budgets.

## A.5 EVALUATION UNDER LARGER PERTURBATION RADII.

To further expand the attack coverage, we additionally evaluate SCC under substantially stronger adversarial budgets, including $\epsilon = 6/255$ and $\epsilon = 8/255$. As shown in Table 6 and Table 7, TTC and related test-time defenses nearly collapse under these large-radius attacks, whereas SCC consistently retains a non-trivial robustness margin across all datasets. These results demonstrate that SCC scales more gracefully to stronger adversarial perturbations, highlighting its stability under challenging threat models.

## A.6 EXPANDED ATTACK COVERAGE.

Beyond PGD-10 and the moderate CW budget used in prior test-time defense studies (Xing et al., 2025), we further incorporate AutoAttack (Croce & Hein, 2020), a parameter-free ensemble attack consisting of APGD-CE and APGD-DLR. Across all 16 datasets, SCC consistently delivers substantial robustness gains over TTC, demonstrating that our method remains effective under a broad and strong family of adversarial attacks, including PGD-100, large-radius perturbations, and AutoAttack (Table 8).

## A.7 EXPANDED EVALUATION UNDER STRONGER ATTACKS (PGD-100)

Regarding attack coverage, our main paper originally focused on PGD-10 and a moderate CW budget for consistency with prior work (Xing et al., 2025). To provide a more comprehensive robustness evaluation, we substantially expand the attack setting by incorporating a much stronger adversary: PGD-100. As shown in Table 9, SCC continues to significantly outperform TTC across all 16 datasets under this stronger attack. For example, on CIFAR-10, SCC achieves 60.72% robust accuracy compared to TTC's 22.36%, demonstrating that SCC remains highly effective even under long-trajectory iterative attacks.

## A.8 RETRIEVAL ROBUSTNESS EVALUATION ON THE FLICKR30K DATASET

While our work focuses on zero-shot classification robustness—the standard setting in recent test-time defense studies (Sheng et al., 2025)—the design of SCC is inherently task-agnostic, as it improves cross-modal margin consistency, a mechanism shared across many VLM tasks such as retrieval. To verify this generality, we further evaluate retrieval robustness on the Flickr30k dataset under PGD attacks. CLIP collapses almost entirely (I→T@1: 60.60%→0.12%, T→I@1: 68.92%→0.16%), and TTC recovers only marginally (I→T@1: 8.16%, T→I@1: 7.02%). In contrast, SCC restores a substantial portion of retrieval accuracy (I→T@1: 42.62%, T→I@1: 43.06%). These results demonstrate that SCC naturally extends beyond classification and remains effective in cross-modal retrieval settings (Table 10).

## A.9 ADAPTIVE PGD ATTACKS AGAINST SCC

To further examine SCC under truly adaptive threats, we implement an attacker that explicitly incorporates the SCC correction mechanism into its optimization objective. Concretely, the adversary back-propagates through the SCC update process and performs gradient ascent on the SCC-corrected

Table 8: Robust accuracy under AutoAttack across 16 datasets. SCC consistently outperforms TTC and recovers a substantial portion of performance lost under adversarial perturbations.

| Method | CIFAR10 | CIFAR100 | STL10 | ImageNet | Caltech101 | Caltech256 | OxfordPet | Flowers102 | FGVC-Aircraft | StanfordCars | SUN397 | Country211 | Food101 | EuroSAT | DTD | PCAM |
|---|---|---|---|---|---|---|---|---|---|---|---|---|---|---|---|---|
| Clean | 85.08 | 57.17 | 96.41 | 59.75 | 85.69 | 81.74 | 87.33 | 65.47 | 20.25 | 51.97 | 58.50 | 15.27 | 83.89 | 42.57 | 40.48 | 53.09 |
| Adv | 0.00 | 0.00 | 0.00 | 0.01 | 0.41 | 0.00 | 0.00 | 0.00 | 0.00 | 0.00 | 0.00 | 0.01 | 0.02 | 0.00 | 0.05 | 0.00 |
| SCC | 60.13 | 38.18 | 79.75 | 30.91 | 68.66 | 59.75 | 49.33 | 32.38 | 8.91 | 20.26 | 33.28 | 4.90 | 33.43 | 23.70 | 25.53 | 20.35 |
| TTC | 4.14 | 8.09 | 4.39 | 6.60 | 10.22 | 7.32 | 6.46 | 10.15 | 3.15 | 5.01 | 7.03 | 2.47 | 5.24 | 15.47 | 7.93 | 8.13 |

Table 9: Robust accuracy (%) under PGD-100 across 16 datasets.

| Method | CIFAR10 | CIFAR100 | STL10 | ImageNet | Cal101 | Cal256 | Pet | Flowers | Aircraft | Cars | SUN | Country | Food | EuroSAT | DTD | PCAM |
|---|---|---|---|---|---|---|---|---|---|---|---|---|---|---|---|---|
| SCC | **60.72** | **33.39** | **90.70** | **47.77** | **77.65** | **71.79** | **75.20** | **51.23** | **16.92** | **38.08** | **51.92** | **9.35** | **62.39** | **19.51** | **38.35** | **71.77** |
| TTC | 22.36 | 11.35 | 66.35 | 28.95 | 52.28 | 45.95 | 48.60 | 29.79 | 9.21 | 16.02 | 33.72 | 5.58 | 39.07 | 8.74 | 29.63 | 62.73 |

logits, corresponding to a worst-case white-box setting in which the attacker fully observes and differentiates through the defense (pgd-adaptive). As shown in Table 12, SCC consistently surpasses TTC across both perturbation budgets, demonstrating that SCC remains robust even when the adversary directly targets its correction dynamics.

## A.10 STABILITY ANALYSIS UNDER MULTI-SEED EVALUATION

Since SCC introduces stochastic multi-view averaging, we evaluate its stability across three random seeds (1–3) on all 16 datasets. The results indicate that SCC exhibits small variance, comparable to TTC, showing that the noise-based multi-view mechanism does not introduce instability. On the contrary, averaging multiple views smooths prediction fluctuations and leads to stable robustness across seeds. Representative statistics (mean $\pm$ var) are shown in Table 11.

## A.11 ROBUSTNESS UNDER STRONGER GEOMETRIC AND PHOTOMETRIC PERTURBATIONS

The multi-view mechanism in SCC intentionally adopts lightweight, semantics-preserving augmentations, following the standard practice in prior test-time robustness methods. Strong geometric or photometric transformations (e.g., large rotations, aggressive cropping, heavy color jitter) often distort CLIP's semantic alignment and cause zero-shot predictions to shift away from the intended semantic content, which contradicts the goal of maintaining stable inference at test time. Therefore, lightweight augmentations have been the default choice for preserving semantics while stabilizing model predictions.

To further examine SCC under more challenging conditions, we additionally evaluate robustness using stronger geometric and photometric perturbations, including random resized-crop, color shifting, and moderate rotation. As shown in Table 13, SCC continues to outperform TTC by large margins across all 16 datasets, demonstrating that SCC remains stable even when the view distribution undergoes substantial visual changes. Notably, several datasets in our benchmark (e.g., ImageNet, Caltech101/256, OxfordPet, Flowers102, FGVC-Aircraft, StanfordCars, SUN397) already contain significant real-world geometric and photometric variations, and SCC maintains strong robustness on these datasets even under lightweight semantics-preserving views.

## A.12 VISUALIZATION OF COUNTERATTACK PERTURBATIONS.

As shown in Figure 8, the counterattack perturbations produced by SCC are small but not strictly imperceptible at the pixel level. Instead, SCC applies localized, low-magnitude corrections that minimally alter the visual appearance while effectively restoring the correct semantic prediction. The zoom-in regions illustrate that SCC selectively adjusts only the most adversarially corrupted areas, whereas the global structure of the image remains visually unchanged. These representative examples are included in the appendix to illustrate the qualitative behavior of SCC's perturbations

## A.13 RESULTS ON OPENCLIP.

To examine whether SCC generalizes beyond the original CLIP architecture, we further evaluate it on OpenCLIP, a widely used CLIP variant trained with different corpora and implementation details. As shown in Table 14, SCC yields substantial robustness gains over TTC across all 16 datasets,

Table 10: Retrieval robustness on Flickr30k under PGD attack. I→T: image-to-text retrieval; T→I: text-to-image retrieval. Top-$k$ accuracy (%).

| Method | I→T | | | T→I | | |
|---|---|---|---|---|---|---|
| | @1 | @5 | @10 | @1 | @5 | @10 |
| CLIP (Clean) | 60.60 | 93.66 | 97.18 | 68.92 | 91.86 | 95.52 |
| Adv (PGD) | 0.12 | 0.86 | 1.70 | 0.16 | 0.98 | 1.80 |
| TTC (Xing et al., 2025) | 8.16 | 28.74 | 39.52 | 7.02 | 22.90 | 31.60 |
| **SCC (Ours)** | **42.62** | **83.10** | **90.78** | **43.06** | **76.52** | **84.88** |

Table 11: Three-seed robustness evaluation (mean ± variance). SCC shows consistently low variance comparable to TTC.

| Dataset | SCC (Mean ± Var) | TTC (Mean ± Var) |
|---|---|---|
| CIFAR10 | 59.51 ± 0.3911 | 28.43 ± 0.2914 |
| CIFAR100 | 31.82 ± 0.2610 | 14.36 ± 0.0503 |
| STL10 | 90.32 ± 0.1960 | 76.83 ± 0.1300 |
| ImageNet | 49.60 ± 0.2155 | 38.47 ± 0.1400 |
| Caltech101 | 77.80 ± 0.5052 | 65.48 ± 0.3395 |
| Caltech256 | 72.87 ± 0.1301 | 60.10 ± 0.0458 |
| OxfordPet | 76.33 ± 0.6806 | 57.93 ± 0.0513 |
| Flowers102 | 54.03 ± 0.4915 | 39.27 ± 0.1893 |
| FGVC-Aircraft | 17.24 ± 0.2516 | 14.00 ± 0.2252 |
| StanfordCars | 43.26 ± 0.1212 | 33.04 ± 0.4306 |
| SUN397 | 53.30 ± 0.0300 | 41.47 ± 0.0416 |
| Country211 | 9.44 ± 0.1026 | 7.11 ± 0.0529 |
| Food101 | 65.46 ± 0.1212 | 57.93 ± 0.0757 |
| EuroSAT | 20.79 ± 0.1890 | 12.33 ± 0.1518 |
| DTD | 34.77 ± 0.1721 | 27.12 ± 0.2358 |
| PCAM | 70.24 ± 0.2194 | 52.96 ± 0.1212 |

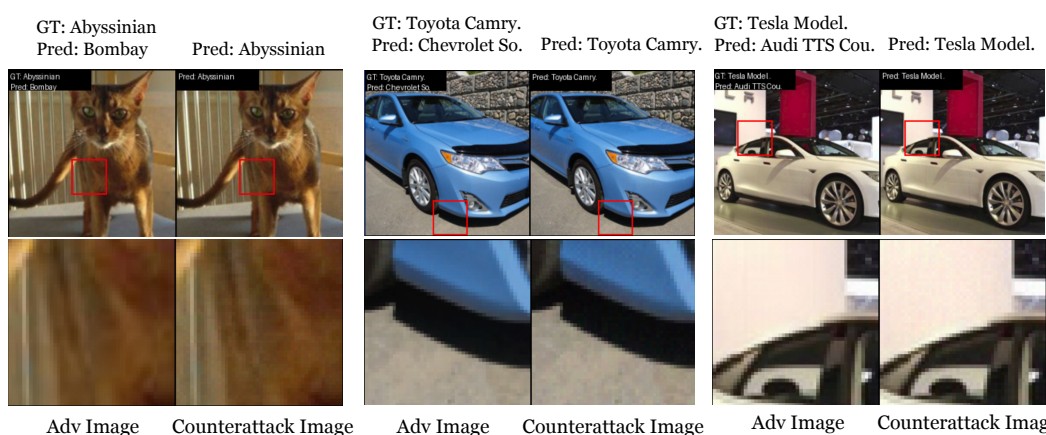

GT: Abyssinian
Pred: Bombay    Pred: Abyssinian

GT: Toyota Camry.
Pred: Chevrolet So.    Pred: Toyota Camry.

GT: Tesla Model.
Pred: Audi TTS Cou.    Pred: Tesla Model.

Adv Image    Counterattack Image        Adv Image    Counterattack Image        Adv Image    Counterattack Image

Figure 8: SCC counterattacks effectively reverse adversarial misclassification with only imperceptible corrections.

demonstrating that its consistency-based mechanism transfers effectively to alternative CLIP backbones.

Table 12: Adaptive PGD attacks that differentiate through SCC. SCC retains clear robustness advantages over TTC across both perturbation budgets.

| Setting | Method | CIFAR10 | CIFAR100 | STL10 | ImageNet | Caltech101 | Caltech256 | OxfordPet | Flowers102 | FGVC-Aircraft | StanfordCars | SUN397 | Country211 | Food101 | EuroSAT | DTD | PCAM | Avg. |
|---|---|---|---|---|---|---|---|---|---|---|---|---|---|---|---|---|---|---|
| pgd_adaptive ($\epsilon$=1) | SCC | 59.72 | 33.38 | 54.02 | 18.39 | 49.51 | 39.85 | 44.37 | 24.17 | 3.00 | 11.57 | 17.76 | 1.91 | 15.60 | 21.33 | 19.26 | 35.56 | 28.09 |
| | TTC | 48.81 | 27.44 | 46.05 | 17.01 | 38.05 | 30.42 | 40.64 | 23.86 | 2.28 | 7.67 | 16.10 | 1.94 | 17.51 | 21.87 | 17.34 | 29.38 | 24.15 |
| pgd_adaptive ($\epsilon$=4) | SCC | 28.12 | 10.59 | 38.64 | 10.49 | 33.10 | 25.92 | 21.94 | 9.89 | 1.26 | 3.71 | 10.51 | 1.19 | 10.28 | 12.21 | 11.33 | 29.68 | 16.18 |
| | TTC | 19.37 | 6.68 | 29.86 | 7.58 | 24.32 | 18.93 | 12.13 | 5.55 | 0.90 | 2.26 | 7.93 | 0.91 | 9.03 | 8.51 | 8.56 | 14.20 | 11.05 |

Table 13: Robustness under stronger geometric and photometric perturbations. SCC remains consistently stronger than TTC across all datasets.

| Method | CIFAR10 | CIFAR100 | STL10 | ImageNet | Caltech101 | Caltech256 | OxfordPet | Flowers102 | FGVC-Aircraft | StanfordCars | SUN397 | Country211 | Food101 | EuroSAT | DTD | PCAM |
|---|---|---|---|---|---|---|---|---|---|---|---|---|---|---|---|---|
| TTC | 28.75 | 14.31 | 76.70 | 38.41 | 65.78 | 60.11 | 57.87 | 39.14 | 13.77 | 33.01 | 41.52 | 7.09 | 57.84 | 12.19 | 27.32 | 52.85 |
| SCC | 59.18 | 32.09 | 90.50 | 49.77 | 77.25 | 72.88 | 76.67 | 54.59 | 17.40 | 43.24 | 53.27 | 9.41 | 65.39 | 20.64 | 34.57 | 69.99 |
| Stronger Aug. | 53.75 | 27.11 | 89.79 | 47.26 | 76.87 | 70.82 | 78.99 | 54.53 | 14.04 | 37.56 | 47.08 | 8.22 | 64.33 | 20.45 | 30.85 | 56.85 |

## A.14 EFFECTS OF OTHER HYPERPARAMETERS

We further analyze the impact of additional hyperparameters on SCC. As shown in Figure 9, increasing the warm-up steps $w$ used for generating pseudo-labels leads to stable clean accuracy but only marginal gains in robustness, which peaks around $w = 5$ before declining. This indicates that a small number of warm-up iterations is sufficient to stabilize pseudo-label quality without introducing excessive counter-perturbations. Therefore, we set $w = 5$. On the other hand, the noise scale $\sigma$ for multi-view augmentation plays a more critical role. Larger $\sigma$ significantly boosts robustness by enhancing view diversity, while clean accuracy decreases gradually as perturbations grow stronger. Overall, SCC exhibits stable behavior across a wide range of hyperparameters, with robustness consistently improving under larger $\sigma$ and modest warm-up steps providing the best trade-off.

## A.15 PROOF SKETCH OF PROPOSITION (HARD-NEGATIVE REPULSION)

Let $m(\delta) = \cos\big(f_{\text{img}}(x + \delta), t_{\text{soft}}\big) - \max_{k \neq \hat{y}} \cos\big(f_{\text{img}}(x + \delta), t_k\big)$ and define $\mathcal{L}_{cm}(\delta) = m(\delta)$. We analyze one PGD-ascent step

$$\delta^+ = \Pi_{\|\cdot\|_p \leq \epsilon}\Big(\delta + \alpha \operatorname{sign}\big(\nabla_\delta \mathcal{L}_{cm}(\delta)\big)\Big), \qquad \alpha > 0.$$

**Assumptions.** (i) In a small neighborhood of $\delta$, the maximizer in the second term is unique and fixed, i.e., there is an active index $j^\star(\delta)$ so the max is smooth; (ii) $f_{\text{img}}$ is differentiable and its Jacobian is bounded; (iii) either the projection is inactive (interior step) or its effect is $O(\alpha^2)$.

**Step 1 (First–order increase).** With the active competitor fixed, $m$ is differentiable. By Taylor's theorem,

$$m(\delta^+) = m(\delta) + \alpha \big\langle \nabla_\delta m(\delta), \operatorname{sign}\big(\nabla_\delta m(\delta)\big) \big\rangle + O(\alpha^2).$$

Since $\langle g, \operatorname{sign}(g) \rangle = \|g\|_1 \geq 0$, it follows that

$$m(\delta^+) \geq m(\delta) + \alpha \big\| \nabla_\delta m(\delta) \big\|_1 + O(\alpha^2).$$

**Step 2 (Relation to $\mathcal{L}_{cm}$).** By definition $\mathcal{L}_{cm} = m$, hence the PGD-ascent direction aligns with $\nabla_\delta m$. Therefore, for sufficiently small $\alpha$,

$$m(\delta^+) \geq m(\delta) + O(\alpha^2),$$

i.e., the semantic margin is monotonically non-decreasing up to second-order terms.

**Step 3 (Active-index changes & projection).** If the active negative $j^\star(\delta)$ switches, $m$ remains subdifferentiable; PGD uses a subgradient and the above inequality holds with $\nabla_\delta m$ replaced by a subgradient. When projection onto the $\ell_p$-ball is active, the component removed is orthogonal to the feasible set's tangent cone, contributing at most $O(\alpha^2)$.

**Conclusion.** Under these mild regularity assumptions, one PGD-ascent step on $\mathcal{L}_{cm}$ increases the margin $m(x + \delta)$ monotonically up to $O(\alpha^2)$. Iteration therefore *repels the hard negative* and prevents drift toward confusable classes, which proves the proposition.

Table 14: Robust accuracy (%) on OpenCLIP under PGD attack. SCC consistently outperforms TTC across all datasets.

| Method | CIFAR10 | CIFAR100 | STL10 | ImageNet | Caltech101 | Caltech256 | OxfordPet | Flowers102 | FGVC-Aircraft | StanfordCars | SUN397 | Country211 | Food101 | EuroSAT | DTD | PCAM | Avg. |
|--------|---------|----------|-------|----------|------------|------------|-----------|------------|---------------|--------------|--------|------------|---------|---------|-----|------|------|
| SCC | **49.38** | **27.63** | **82.92** | **50.39** | **80.05** | **77.39** | **77.84** | **61.31** | **18.84** | **62.70** | **59.42** | **10.12** | **60.03** | **13.56** | **45.48** | **52.32** | **51.84** |
| TTC | 9.64 | 4.51 | 59.08 | 41.86 | 69.67 | 66.19 | 70.92 | 54.94 | 15.18 | 52.94 | 49.34 | 8.01 | 55.60 | 2.80 | 37.07 | 28.16 | 39.12 |

Table 15: Comparison of robust accuracy (Rob.) and clean accuracy (Acc.) across datasets. ($\epsilon_a = 1/255$)

| Dataset | Metric | CLIP | Adversarial Finetuning | | | | Test-time Defence | | | | | | $\Delta$ |
|---------|--------|------|--------|-------|---------|------|------|---------|------|------|-----------|-----------|----------|
| | | | CLIP-FT | TeCoA | PMG-AFT | FARE | RN | Anti-adv | HD | TTC | SCC(ours) | SCC*(ours) | |
| CIFAR10 | Rob. | 0.74 | 3.34 | 33.61 | 40.66 | 19.65 | 2.01 | 12.39 | 17.22 | 28.75 | **59.18** | 48.63 | +47.89 |
| | Acc. | 85.12 | 84.90 | 64.61 | 70.69 | 74.44 | 81.18 | 83.52 | 78.23 | 81.18 | 82.24 | 81.29 | -3.83 |
| CIFAR100 | Rob. | 0.26 | 0.90 | 18.95 | 22.52 | 11.40 | 0.67 | 5.73 | 3.86 | 14.31 | **32.09** | 29.38 | +29.12 |
| | Acc. | 57.14 | 59.51 | 35.96 | 40.32 | 46.67 | 56.34 | 54.27 | 54.54 | 55.21 | 56.73 | -0.41 |
| STL10 | Rob. | 11.00 | 12.73 | 70.08 | 73.08 | 59.06 | 16.23 | 37.42 | 39.02 | 76.70 | **90.50** | 90.12 | +79.12 |
| | Acc. | 96.40 | 94.49 | 87.40 | 88.56 | 91.72 | 95.85 | 95.45 | 89.50 | 95.85 | 95.62 | 95.85 | -0.55 |
| ImageNet | Rob. | 1.15 | 0.93 | 18.89 | 21.43 | 14.00 | 1.77 | 8.67 | 6.63 | 38.41 | 49.77 | **56.14** | +54.99 |
| | Acc. | 59.69 | 54.24 | 34.89 | 36.12 | 48.79 | 59.34 | 54.27 | 54.54 | 49.39 | 56.03 | 59.66 | -0.03 |
| Caltech101 | Rob. | 14.67 | 14.21 | 55.51 | 61.08 | 50.74 | 18.90 | 34.81 | 31.53 | 65.78 | 77.25 | **82.04** | +67.37 |
| | Acc. | 85.66 | 83.63 | 71.68 | 75.45 | 80.95 | 86.61 | 84.02 | 82.33 | 86.53 | 86.44 | 86.56 | +0.90 |
| Caltech256 | Rob. | 8.47 | 6.76 | 43.19 | 45.91 | 38.79 | 11.33 | 25.36 | 23.48 | 60.11 | 72.88 | **76.85** | +68.38 |
| | Acc. | 81.72 | 78.53 | 61.14 | 62.24 | 73.32 | 81.25 | 79.38 | 79.12 | 79.66 | 81.16 | 81.64 | -0.08 |
| OxfordPets | Rob. | 1.04 | 2.10 | 38.35 | 41.18 | 31.07 | 1.86 | 20.42 | 12.04 | 57.87 | 76.67 | **85.69** | +84.65 |
| | Acc. | 87.44 | 84.14 | 62.12 | 65.88 | 79.37 | 87.41 | 80.62 | 80.91 | 83.35 | 86.48 | 87.79 | +0.35 |
| Flowers102 | Rob. | 1.14 | 0.54 | 21.94 | 23.43 | 17.14 | 1.52 | 7.16 | 7.29 | 39.14 | 54.59 | **63.38** | +62.24 |
| | Acc. | 65.46 | 53.37 | 36.80 | 37.00 | 47.98 | 64.62 | 62.66 | 58.22 | 64.16 | 64.16 | 64.43 | -1.03 |
| FGVC-Aircraft | Rob. | 0.00 | 0.00 | 2.49 | 2.22 | 1.35 | 0.00 | 1.27 | 1.26 | 13.77 | **17.40** | 16.98 | +16.98 |
| | Acc. | 20.10 | 14.04 | 5.31 | 5.55 | 10.86 | 19.25 | 15.88 | 16.36 | 17.61 | 17.61 | 18.63 | -1.47 |
| StanfordCars | Rob. | 0.02 | 0.06 | 8.76 | 11.65 | 6.75 | 0.16 | 4.40 | 2.71 | 33.01 | 43.24 | **50.95** | +50.93 |
| | Acc. | 52.02 | 42.11 | 20.91 | 25.44 | 38.68 | 52.14 | 36.21 | 44.28 | 48.16 | 51.19 | 52.64 | +0.62 |
| SUN397 | Rob. | 1.14 | 0.94 | 19.39 | 22.58 | 14.91 | 1.72 | 8.05 | 6.40 | 41.52 | 53.27 | **56.00** | +54.86 |
| | Acc. | 58.50 | 55.73 | 36.69 | 37.98 | 52.42 | 59.69 | 56.00 | 53.17 | 55.13 | 58.25 | 59.98 | +1.48 |
| Country211 | Rob. | 0.04 | 0.03 | 1.78 | 2.12 | 0.85 | 0.06 | 0.67 | 0.47 | 7.09 | 9.41 | **12.55** | +12.51 |
| | Acc. | 15.25 | 12.07 | 4.75 | 4.64 | 9.26 | 14.80 | 11.58 | 13.08 | 13.36 | 14.69 | -0.56 |
| Food101 | Rob. | 0.70 | 0.42 | 13.90 | 18.57 | 11.65 | 1.20 | 13.12 | 8.03 | 57.84 | 65.39 | **81.57** | +80.87 |
| | Acc. | 83.88 | 64.86 | 29.98 | 36.61 | 55.31 | 83.44 | 75.81 | 80.30 | 82.18 | 82.13 | 83.71 | -0.17 |
| EuroSAT | Rob. | 0.03 | 0.04 | 11.96 | 12.60 | 10.67 | 0.15 | 2.15 | 4.57 | 12.19 | 20.64 | **24.00** | +23.97 |
| | Acc. | 42.59 | 27.64 | 16.58 | 18.53 | 21.88 | 53.24 | 36.78 | 39.08 | 53.24 | 41.69 | 52.60 | +10.01 |
| DTD | Rob. | 2.98 | 2.39 | 17.61 | 14.95 | 15.64 | 3.71 | 5.62 | 11.63 | 27.32 | 34.57 | **36.06** | +33.08 |
| | Acc. | 40.64 | 36.49 | 25.16 | 21.76 | 32.07 | 37.96 | 38.92 | 34.89 | 36.98 | 37.34 | 38.09 | -2.55 |
| PCAM | Rob. | 0.08 | 1.11 | 48.24 | 46.18 | 16.23 | 0.41 | 4.97 | 44.74 | 52.85 | **69.99** | 47.54 | +47.46 |
| | Acc. | 52.02 | 47.21 | 49.96 | 50.03 | 52.54 | 52.73 | 52.49 | 50.38 | 52.73 | 54.41 | 54.24 | +2.22 |
| Avg. | Rob. | 2.70 | 2.91 | 26.54 | 28.76 | 20.00 | 3.86 | 12.01 | 13.81 | 39.17 | 51.68 | **53.62** | +50.92 |
| | Acc. | 61.51 | 55.80 | 40.25 | 42.30 | 51.02 | 61.61 | 57.35 | 56.62 | 59.75 | 60.21 | 61.78 | +0.27 |

## A.16 PROOF SKETCH OF PROPOSITION ( SUPPRESSION OF SPURIOUS NEGATIVES)

Let $\bar{z}_j \triangleq \frac{1}{L}\sum_{i=1}^{L} z_j^{(i)}$ and let $j^\dagger \in \arg\max_{j\neq y^\star} \bar{z}_j$ be an index achieving the maximum of the averaged logits (excluding $y^\star$). Then

$$\max_{j\neq y^\star} \bar{z}_j = \bar{z}_{j^\dagger} = \frac{1}{L}\sum_{i=1}^{L} z_{j^\dagger}^{(i)} \leq \frac{1}{L}\sum_{i=1}^{L} \max_{j\neq y^\star} z_j^{(i)},$$

since for each $i$, $z_{j^\dagger}^{(i)} \leq \max_{j\neq y^\star} z_j^{(i)}$. This proves $\max_{j\neq y^\star} \bar{z}_j \leq \frac{1}{L}\sum_{i=1}^{L}\max_{j\neq y^\star} z_j^{(i)}$.

## A.17 THE USE OF LARGE LANGUAGE MODELS

Large language models were used to improve the clarity and presentation of writing. All methodological design, experiments, and analysis were conducted by the authors.

Table 16: Classification accuracy (%) on adversarial images (Rob.) under 10-step CW attack ($\epsilon_a = 1/255$) (Carlini & Wagner, 2017) and on clean images (Acc.) across 16 datasets. We assume the threat model has full access to model weights and gradients. We compare with test-time defenses adapted from prior work and include fine-tuning methods as references. The last column reports gains over vanilla CLIP.

| Dataset | Metric | CLIP | Adversarial Finetuning | | | | Test-time Defence | | | | | | Δ |
|---|---|---|---|---|---|---|---|---|---|---|---|---|---|
| | | | CLIP-FT | TeCoA | PMG-AFT | FARE | RN | TTE | Anti-adv | HD | TTC | SCC(ours) | |
| CIFAR10 | Rob. | 0.87 | 0.94 | 33.27 | 39.50 | 20.60 | 2.05 | 40.01 | 12.53 | 14.79 | 29.04 | **58.42** | +57.55 |
| | Acc. | 85.12 | 84.90 | 64.61 | 70.69 | 74.44 | 81.18 | 84.74 | 83.52 | 78.64 | 81.18 | 82.24 | -2.88 |
| CIFAR100 | Rob. | 0.29 | 0.39 | 18.27 | 20.83 | 11.67 | 0.63 | 18.73 | 6.56 | 3.04 | 14.38 | **30.89** | +30.60 |
| | Acc. | 57.14 | 59.51 | 35.96 | 40.32 | 46.67 | 56.34 | 58.61 | 53.95 | 53.50 | 56.34 | 55.21 | -1.93 |
| STL10 | Rob. | 12.23 | 9.95 | 69.73 | 72.39 | 59.60 | 17.20 | 78.64 | 38.66 | 37.73 | 76.40 | **89.99** | +77.76 |
| | Acc. | 96.40 | 94.49 | 87.40 | 88.56 | 91.72 | 95.85 | 96.26 | 95.45 | 89.54 | 95.85 | 95.62 | -0.78 |
| ImageNet | Rob. | 1.46 | 1.27 | 18.28 | 19.42 | 27.71 | 2.21 | 29.77 | 9.37 | 7.46 | 36.01 | **45.75** | +44.29 |
| | Acc. | 59.69 | 54.24 | 34.89 | 36.12 | 48.79 | 59.34 | 60.02 | 54.27 | 55.06 | 49.39 | 56.03 | -3.66 |
| Caltech101 | Rob. | 20.88 | 15.95 | 56.23 | 61.58 | 54.86 | 25.89 | 69.44 | 41.47 | 36.26 | 66.17 | **76.59** | +55.71 |
| | Acc. | 85.66 | 83.63 | 71.68 | 75.45 | 80.95 | 86.61 | 85.84 | 84.02 | 83.00 | 86.53 | 86.44 | +0.78 |
| Caltech256 | Rob. | 9.69 | 7.24 | 42.63 | 44.55 | 39.58 | 13.11 | 59.81 | 27.17 | 24.54 | 58.79 | **70.55** | +60.86 |
| | Acc. | 81.72 | 78.53 | 61.14 | 62.24 | 73.32 | 81.25 | 82.48 | 79.38 | 79.38 | 79.66 | 81.16 | -0.56 |
| OxfordPets | Rob. | 1.64 | 1.14 | 37.91 | 39.28 | 33.85 | 3.11 | 51.12 | 22.99 | 13.84 | 57.15 | **75.06** | +73.42 |
| | Acc. | 87.44 | 84.14 | 62.12 | 65.88 | 79.37 | 87.41 | 88.13 | 80.62 | 80.64 | 83.35 | 86.48 | -0.96 |
| Flowers102 | Rob. | 1.35 | 0.80 | 21.13 | 21.34 | 17.25 | 2.13 | 34.97 | 8.06 | 8.51 | 36.84 | **49.76** | +48.41 |
| | Acc. | 65.46 | 53.37 | 36.80 | 37.00 | 47.98 | 64.62 | 65.20 | 62.66 | 57.79 | 64.16 | 64.16 | -1.30 |
| FGVCAircraft | Rob. | 0.00 | 0.00 | 2.25 | 1.86 | 1.35 | 0.00 | 5.15 | 0.83 | 0.97 | 12.41 | **15.18** | +15.18 |
| | Acc. | 20.10 | 14.04 | 5.31 | 5.55 | 10.86 | 19.25 | 20.18 | 15.88 | 16.18 | 18.00 | 17.61 | -2.49 |
| StanfordCars | Rob. | 2.38 | 2.04 | 8.74 | 10.53 | 9.14 | 2.44 | 21.19 | 4.76 | 5.11 | 30.38 | **37.96** | +35.58 |
| | Acc. | 52.02 | 42.11 | 20.91 | 25.44 | 38.68 | 52.14 | 52.73 | 36.21 | 48.16 | 51.19 | -0.83 |
| SUN397 | Rob. | 1.75 | 1.48 | 18.36 | 20.39 | 15.73 | 2.48 | 29.37 | 8.85 | 7.90 | 39.44 | **48.99** | +47.24 |
| | Acc. | 58.50 | 55.73 | 36.69 | 37.98 | 52.42 | 59.69 | 59.12 | 56.00 | 54.07 | 55.13 | 58.25 | -0.25 |
| Country211 | Rob. | 0.08 | 0.05 | 1.46 | 1.74 | 0.92 | 0.15 | 3.00 | 0.72 | 0.75 | 6.17 | **7.61** | +7.53 |
| | Acc. | 15.25 | 12.07 | 4.75 | 4.64 | 9.26 | 14.80 | 14.66 | 11.58 | 13.08 | 13.36 | -1.89 |
| Food101 | Rob. | 1.09 | 0.55 | 12.87 | 16.57 | 12.93 | 1.92 | 44.61 | 15.03 | 9.77 | 54.65 | **59.73** | +58.64 |
| | Acc. | 83.88 | 64.86 | 29.98 | 36.61 | 55.31 | 83.44 | 83.96 | 75.81 | 81.02 | 82.18 | 82.13 | -1.75 |
| EuroSAT | Rob. | 0.03 | 0.03 | 11.66 | 11.94 | 10.66 | 0.16 | 6.44 | 2.57 | 3.47 | 12.69 | **20.52** | +20.49 |
| | Acc. | 42.59 | 27.64 | 16.58 | 18.53 | 21.88 | 53.24 | 44.38 | 36.78 | 40.12 | 53.24 | 41.69 | -0.90 |
| DTD | Rob. | 2.87 | 2.77 | 16.28 | 13.72 | 14.36 | 3.46 | 22.62 | 6.06 | 10.11 | 27.39 | **33.35** | +30.48 |
| | Acc. | 40.64 | 36.49 | 25.16 | 21.76 | 32.07 | 37.96 | 38.92 | 35.25 | 36.98 | 37.34 | -3.30 |
| PCAM | Rob. | 0.10 | 1.10 | 48.29 | 46.36 | 16.41 | 0.44 | 10.70 | 5.07 | 46.92 | 52.86 | **70.36** | +70.26 |
| | Acc. | 52.02 | 47.21 | 49.96 | 50.03 | 52.54 | 52.73 | 50.92 | 52.49 | 50.35 | 52.73 | 54.41 | +2.39 |
| Avg. | Rob. | 3.54 | 2.86 | 26.09 | 27.62 | 20.86 | 4.84 | 32.85 | 13.17 | 14.45 | 38.17 | **49.42** | +45.88 |
| | Acc. | 61.51 | 55.80 | 40.25 | 42.30 | 51.02 | 61.61 | 61.79 | 57.35 | 56.88 | 59.75 | 60.21 | -1.30 |

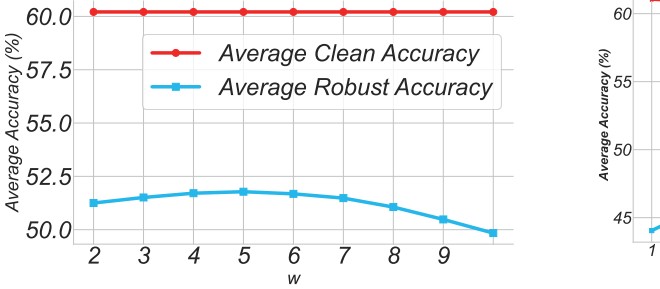
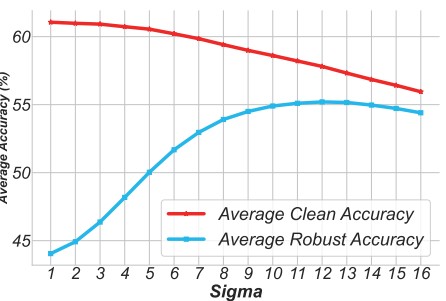

Figure 9: Effect of the warm-up step number $w$ in short TTC: a moderate number yields the best robustness, while clean accuracy is unaffected. Effect of the Gaussian noise scale $\sigma$ (Sigma). Robustness improves with more views and larger $\sigma$, while clean accuracy drops.

Table 17: Classification accuracy (%) on clean images (Acc.) and adversarial images (Rob.) under 10-step PGD attack ($\epsilon_a = 4/255$) across 16 datasets. The threat model assumes full access to model weights and gradients. We compare our paradigm against test-time defenses adapted from prior adversarial robustness studies, and include fine-tuned models as references. The last column shows the gains of SCC over the original CLIP.

| Dataset | Metric | CLIP | Adversarial Finetuning | | | | Test-time Defence | | | | | Δ |
|---|---|---|---|---|---|---|---|---|---|---|---|---|
| | | | CLIP-FT | TeCoA[1] | TeCoA[4] | PMG-AFT[1] | RN | Anti-adv | HD | TTC | SCC(ours) | |
| CIFAR10 | Rob. | 0.43 | 2.75 | 7.69 | 11.70 | 10.20 | 0.00 | 0.32 | 1.67 | 28.51 | **36.30** | +35.87 |
| | Acc. | 85.12 | 84.90 | 64.61 | 65.15 | 70.69 | 81.18 | 83.44 | 78.23 | 81.18 | 82.24 | -2.88 |
| CIFAR100 | Rob. | 0.05 | 0.67 | 6.54 | 9.25 | 7.60 | 0.00 | 0.22 | 0.00 | 14.46 | **14.46** | +14.41 |
| | Acc. | 57.14 | 59.51 | 35.96 | 36.30 | 40.32 | 56.34 | 53.96 | 52.86 | 56.34 | 55.21 | -1.93 |
| STL10 | Rob. | 0.16 | 3.75 | 24.80 | 31.83 | 28.49 | 0.06 | 2.25 | 3.39 | 52.40 | **67.66** | +67.50 |
| | Acc. | 96.40 | 94.49 | 87.40 | 81.69 | 88.56 | 95.85 | 95.47 | 89.50 | 95.83 | 95.62 | -0.78 |
| ImageNet | Rob. | 0.00 | 0.07 | 1.65 | 3.00 | 2.07 | 0.00 | 0.15 | 0.01 | 12.68 | **20.57** | +20.57 |
| | Acc. | 59.69 | 54.24 | 34.89 | 27.76 | 36.12 | 59.34 | 54.29 | 54.54 | 34.00 | 57.34 | -2.35 |
| Caltech101 | Rob. | 0.59 | 4.81 | 15.75 | 21.00 | 19.48 | 0.68 | 3.14 | 1.27 | 36.66 | **54.44** | +53.85 |
| | Acc. | 85.66 | 83.63 | 71.68 | 64.41 | 75.45 | 86.61 | 83.99 | 82.33 | 86.15 | 86.46 | +0.80 |
| Caltech256 | Rob. | 0.12 | 1.41 | 8.29 | 11.76 | 10.65 | 0.16 | 1.44 | 0.34 | 27.25 | **44.06** | +43.94 |
| | Acc. | 81.72 | 78.53 | 61.14 | 52.05 | 62.24 | 81.25 | 79.40 | 79.12 | 76.59 | 81.32 | -0.40 |
| OxfordPets | Rob. | 0.00 | 1.66 | 0.90 | 3.71 | 1.74 | 0.00 | 0.10 | 0.00 | 24.64 | **37.69** | +37.69 |
| | Acc. | 87.44 | 84.14 | 62.12 | 53.94 | 65.88 | 87.41 | 80.53 | 80.91 | 64.70 | 86.62 | -0.82 |
| Flowers102 | Rob. | 0.00 | 0.13 | 1.87 | 3.81 | 2.57 | 0.00 | 0.05 | 0.00 | 13.60 | **21.97** | +21.97 |
| | Acc. | 65.46 | 53.37 | 36.80 | 27.78 | 37.00 | 64.62 | 62.80 | 58.22 | 63.24 | 64.19 | -1.27 |
| FGVCAircraft | Rob. | 0.00 | 0.00 | 0.03 | 0.12 | 0.03 | 0.00 | 0.00 | 0.00 | 6.40 | **7.20** | +7.20 |
| | Acc. | 20.10 | 14.04 | 5.31 | 3.51 | 5.55 | 19.25 | 15.64 | 16.36 | 15.99 | 17.79 | -2.31 |
| StanfordCars | Rob. | 0.00 | 0.00 | 0.15 | 0.41 | 0.15 | 0.00 | 0.00 | 0.00 | 12.84 | **19.40** | +19.40 |
| | Acc. | 52.02 | 42.11 | 20.91 | 15.18 | 25.44 | 52.14 | 36.14 | 44.28 | 41.52 | 51.61 | -0.41 |
| SUN397 | Rob. | 0.00 | 0.02 | 1.30 | 2.31 | 1.90 | 0.00 | 0.11 | 0.00 | 13.43 | **21.77** | +21.77 |
| | Acc. | 58.50 | 55.73 | 36.69 | 28.16 | 37.98 | 59.69 | 55.99 | 53.17 | 46.68 | 58.68 | +0.18 |
| Country211 | Rob. | 0.00 | 0.00 | 0.05 | 0.19 | 0.12 | 0.00 | 0.00 | 0.00 | 2.44 | **2.85** | +2.85 |
| | Acc. | 15.25 | 12.07 | 4.75 | 3.66 | 4.64 | 14.80 | 11.60 | 11.72 | 11.99 | 13.55 | -1.70 |
| Food101 | Rob. | 0.00 | 0.04 | 0.56 | 1.35 | 1.03 | 0.00 | 0.07 | 0.01 | 17.89 | **26.58** | +26.58 |
| | Acc. | 83.88 | 64.86 | 29.98 | 21.90 | 36.61 | 83.44 | 75.95 | 80.30 | 80.00 | 82.36 | -1.52 |
| EuroSAT | Rob. | 0.00 | 0.00 | 9.77 | 10.71 | 9.61 | 0.00 | 0.03 | 0.20 | 13.57 | **10.61** | +10.61 |
| | Acc. | 42.59 | 27.64 | 16.58 | 17.53 | 18.53 | 53.24 | 36.81 | 39.08 | 53.24 | 41.69 | -0.90 |
| DTD | Rob. | 0.11 | 0.00 | 4.20 | 5.16 | 4.31 | 0.11 | 0.37 | 0.16 | 11.40 | **16.33** | +16.22 |
| | Acc. | 40.64 | 36.49 | 25.16 | 20.11 | 21.76 | 37.96 | 38.55 | 34.89 | 35.69 | 37.66 | -2.98 |
| PCAM | Rob. | 0.00 | 0.00 | 20.54 | 44.13 | 12.59 | 0.00 | 0.25 | 12.04 | 47.39 | **44.19** | +44.19 |
| | Acc. | 52.02 | 47.21 | 49.96 | 49.98 | 50.03 | 52.73 | 52.61 | 50.38 | 52.73 | 54.41 | +2.39 |
| Avg. | Rob. | 0.09 | 0.96 | 6.51 | 10.03 | 7.03 | 0.06 | 0.53 | 1.19 | 20.63 | **27.88** | +27.79 |
| | Acc. | 61.51 | 55.80 | 40.25 | 35.57 | 42.30 | 61.61 | 57.32 | 56.62 | 55.99 | 60.42 | -1.09 |

---

**Algorithm 1:** SCC: Self-Calibrated Consistency

---

**Input:** image $x$, text embeddings $\{t_k\}$, budget $\epsilon$, steps $S$, views $V$, temp $T$
**Output:** predicted label $\hat{y}$
/* Short warm-up (TTC) to stabilize predictions                     */
Initialize $\delta_{\text{warm}} = 0$; run $S_{\text{warm}}$ PGD-ascent steps on TTC to obtain $x + \delta_{\text{warm}}$.
/* Multi-view pseudo-label on the warmed input                      */
Sample $V$ augmented views $\{v_i(x + \delta_{\text{warm}})\}$;
$\bar{z} = \frac{1}{V} \sum_i f_{\text{img}}(v_i(x + \delta_{\text{warm}}))^\top [t_k]$;
$p = \text{softmax}(\bar{z}/T)$, $\hat{y} = \arg\max_k p_k$, $t_{\text{soft}} = \sum_k p_k t_k$.
/* Counterattack optimization (sign-PGD ascent; shared $\delta$)      */
Initialize $\delta = 0$;;
**for** $s = 1$ **to** $S$ **do**
    $f = f_{\text{img}}(x + \delta)$;
    $L_{\text{cm}} = \langle f, t_{\text{soft}} \rangle - \max_{j \neq \hat{y}} \langle f, t_j \rangle$;
    $L_{\text{drift}} = \|f - f_{\text{img}}(x)\|_2^2$;
    $\delta \leftarrow \Pi_{\|\delta\|_\infty \leq \epsilon}\big(\delta + \alpha \, \text{sign}(\nabla_\delta(\lambda_{cm} L_{\text{cm}} + L_{\text{drift}}))\big)$.

$\delta \leftarrow \text{StepWeightedFuse}(\{\delta^{(s)}\}_{s=0}^S; \tau, \beta)$.
/* Final prediction (logit averaging on shared-$\delta$ views)        */
Form views $\{v_i(x + \delta)\}$; ;
$\bar{z} = \frac{1}{V} \sum_i f_{\text{img}}(v_i(x + \delta))^\top [t_k]$; ;
$\hat{y} = \arg\max \text{softmax}(\bar{z})$.

