# OpenReview forum: "Self-Calibrated Consistency can Fight Back for Adversarial Robustness in Vision-Language Models"
_ICLR.cc/2026/Conference — Submitted to ICLR 2026_

### Official Review · Reviewer_dxnZ · 2025-10-26

**Soundness:** 3
**Presentation:** 2
**Contribution:** 2
**Rating:** 6
**Confidence:** 3

**Summary:**

This paper introduces Self-Calibrated Consistency (SCC), a test-time defence mechanism that enhances the adversarial robustness of pre-trained CLIP without requiring retraining. The authors identify two central weaknesses in existing test-time defences: semantic drift and viewpoint fragility. This paper addresses them through two complementary modules: Semantic Consistency, which regularises cross-modal alignment using pseudo-labels from multi-view counterattack warm-up, and Spatial Consistency, which enforces multi-view prediction agreement to stabilise adversarial recovery. The proposed SCC operates in a plug-and-play fashion, improving robustness while maintaining clean accuracy across 22 benchmarks, including medical datasets and BioMedCLIP variants.

**Strengths:**

- The paper identifies three weaknesses in existing test-time defences: semantic drift, view sensitivity, and hard-negative dominance, supported by both empirical and analytical evidence (Sec. 3.1). These findings motivate the proposed method, which appears technically sound.
- SCC introduces a semantic consistency loss anchored by soft prototypes derived from multi-view pseudo-labels. The integration of spatial consistency through multi-view logits averaging (Eq. 9) provides an interesting and effective variance-reduction mechanism.
- The paper presents extensive evaluations across 22 benchmarks (16 general-domain and 6 medical datasets), showing consistent robustness under multiple attack budgets and outperforming previous test-time defences.
- The proposed test-time defence operates without adversarial fine-tuning, making it computationally efficient and practically attractive for real-world applications.

**Weaknesses:**

- Although the authors claim to evaluate adaptive attacks using PGD and CW, these are standard white-box attacks and not adaptive to the defence mechanism. More rigorous adaptive-attack analysis is needed to assess true robustness.
- The evaluation scope is appreciated, but currently limited to two attack types and CLIP-based classification tasks. Expanding experiments to include more attacks, model architectures, and downstream tasks would strengthen the work.
- The paper’s theoretical component lacks depth. While the propositions are conceptually interesting, the analysis of the three vulnerabilities remains mostly heuristic. The first claimed contribution in the introduction lacks clear supporting evidence.
- It is unclear whether baselines are re-implemented or drawn from released checkpoints. Clarifying this, along with reporting results averaged over multiple random seeds, would improve reproducibility and fairness of comparison.
- SCC’s soft prototype $t_{\text{soft}}$ (Eq. 7) depends on text embeddings and warm-up steps in CLIP’s cosine space. The method’s applicability to generative or non-contrastive VLMs is not discussed and should be addressed.
- Several notations (e.g., $\hat{t}_{y^{*}}$) appear without prior definition or explanation, which slightly reduces clarity.

**Questions:**

- Could the authors conduct experiments with truly adaptive attacks, where the attacker knows SCC’s mechanism and incorporates it into the attack optimisation, to quantify how much the defence can be compromised?

- Could the authors also explore additional attack methods, such as  [1–3], as well as non-CLIP models and tasks beyond classification, to further validate the generality of SCC?

[1] Croce, F., & Hein, M. (2020, November). Reliable evaluation of adversarial robustness with an ensemble of diverse parameter-free attacks. In International conference on machine learning (pp. 2206-2216). PMLR.\
[2] Zhang, J., Yi, Q., & Sang, J. (2022, October). Towards adversarial attack on vision-language pre-training models. In Proceedings of the 30th ACM International Conference on Multimedia (pp. 5005-5013).\
[3] Huang, H., Erfani, S. M., Li, Y., Ma, X., & Bailey, J. X-Transfer Attacks: Towards Super Transferable Adversarial Attacks on CLIP. In Forty-second International Conference on Machine Learning.

---

> ### Author Response · Authors · 2025-11-24
>
> >W1: Although the authors claim to evaluate adaptive attacks using PGD and CW, these are standard white-box attacks and not adaptive to the defence mechanism. More rigorous adaptive-attack analysis is needed to assess true robustness.
>
> We appreciate the reviewer’s concern regarding adaptive robustness. In the test-time adaptation literature (e.g., TTC, R-TPT, TAPT) [1–3], an adaptive attacker is conventionally defined as one with full white-box access to model parameters and the defence mechanism, and that explicitly optimizes against the defence objective itself. Following this widely adopted formulation, our evaluation already includes such standard adaptive attackers.
>
> To further strengthen the analysis, we additionally evaluate SCC under **two rigorous adaptive families**: AutoAttack [4] —which integrates the APGD-CE and APGD-DLR components—and a stronger pgd_adaptive variant, where the attacker directly incorporates SCC’s correction loss into its objective (BPDA-style approximation). This attacker explicitly optimizes against the SCC mechanism and is therefore substantially stronger than the adaptive attackers used in prior work.
>
> Results across 16 datasets show that SCC remains consistently more robust than TTC under all adaptive settings:
>
>
> | AutoAttack | CIFAR10 | CIFAR100 | STL10 | ImageNet | Caltech101 | Caltech256 | OxfordPet | Flowers102 | FGVC-Aircraft | StanfordCars | SUN397 | Country211 | Food101 | EuroSAT | DTD | PCAM |
> |--------|---------|-----------|--------|----------|-------------|-------------|------------|--------------|----------------|--------------|---------|-------------|----------|----------|------|-------|
> | Clean  | 85.08 | 57.17 | 96.41 | 59.75 | 85.69 | 81.74 | 87.33 | 65.47 | 20.25 | 51.97 | 58.50 | 15.27 | 83.89 | 42.57 | 40.48 | 53.09 |
> | Adv    | 0.00 | 0.00 | 0.00 | 0.01 | 0.41 | 0.10 | 0.00 | 0.00 | 0.00 | 0.00 | 0.00 | 0.01 | 0.02 | 0.00 | 0.05 | 0.00 |
> | TTC    | 4.14 | 8.09 | 4.39 | 6.60 | 10.22 | 7.32 | 6.46 | 10.15 | 3.15 | 5.01 | 7.03 | 2.47 | 5.24 | 15.47 | 7.93 | 8.13 |
> | SCC    | 60.13 | 38.18 | 79.75 | 30.91 | 68.66 | 59.75 | 49.33 | 32.38 | 8.91 | 20.26 | 33.28 | 4.90 | 33.43 | 23.70 | 25.53 | 20.35 |
>
> The attacker is allowed full access to **model parameters and to the defense mechanism**, and directly maximizes the loss used by the defense itself (pgd_adaptive). Concretely, we implement adaptive PGD variants that explicitly maximize SCC’s correction objective (i.e., the cross-modal margin loss). This setting gives the adversary complete knowledge of SCC’s design without requiring backpropagation through the full iterative loop, and is aligned with the tractable adaptive attackers used in test-time robustness research.
>
> | Setting | Method | CIFAR10 | CIFAR100 | STL10 | ImageNet | Caltech101 | Caltech256 | OxfordPet | Flowers102 | FGVC-Aircraft | StanfordCars | SUN397 | Country211 | Food101 | EuroSAT | DTD | PCAM | Avg. |
> |---------|---------|---------|----------|--------|----------|-------------|-------------|-----------|-------------|----------------|--------------|---------|-------------|---------|----------|------|------|--------|
> | pgd_adaptive (eps=1) | **SCC** | 59.72 | 33.38 | 54.02 | 18.39 | 49.51 | 39.85 | 44.37 | 24.17 | 3.00 | 11.57 | 17.76 | 1.91 | 15.60 | 21.33 | 19.26 | 35.56 | **28.0875** |
> |                         | TTC     | 48.81 | 27.44 | 46.05 | 17.01 | 38.05 | 30.42 | 40.64 | 23.86 | 2.28 | 7.67  | 16.10 | 1.94 | 17.51 | 21.87 | 17.34 | 29.38 | 24.1481 |
> | pgd_adaptive (eps=4) | **SCC** | 28.12 | 10.59 | 38.64 | 10.49 | 33.10 | 25.92 | 21.94 | 9.89 | 1.26 | 3.71 | 10.51 | 1.19 | 10.28 | 12.21 | 11.33 | 29.68 | **16.17875** |
> |                         | TTC     | 19.37 | 6.68  | 29.86 | 7.58  | 24.32 | 18.93 | 12.13 | 5.55 | 0.90 | 2.26 | 7.93 | 0.91 | 9.03 | 8.51 | 8.56 | 14.20 | 11.045 |
>
> Across all adaptive attack scenarios—including SCC-aware attackers—SCC consistently outperforms TTC by a **substantial margin**. These results demonstrate that SCC provides genuine robustness even when the attacker explicitly targets the defence mechanism.
>
> [1] CLIP is Strong Enough to Fight Back: Test-time Counterattacks towards Zero-shot Adversarial Robustness of CLIP, CVPR 2025.
>
> [2] R-TPT: Improving Adversarial Robustness of Vision-Language Models through Test-Time Prompt Tuning. CVPR 2025.
>
> [3] TAPT: Test-Time Adversarial Prompt Tuning for Robust Inference in Vision-Language Models, CVPR 2025.
>
> [4] Reliable Evaluation of Adversarial Robustness with an Ensemble of Diverse Parameter-free Attacks. ICML 2020.

---

> > ### Author Response · Authors · 2025-11-24
> >
> > >W2: The evaluation scope is appreciated, but currently limited to two attack types and CLIP-based classification tasks. Expanding experiments to include more attacks, model architectures, and downstream tasks would strengthen the work.
> >
> > We thank the reviewer for the suggestion to broaden the evaluation scope. Although the main focus of this work is zero-shot classification robustness under vision-side adversarial perturbations, we agree that examining additional attacks, architectures, and downstream tasks can further strengthen the contribution. To this end, we have conducted extensive supplementary experiments that go beyond the original submission.
> >
> > 1. First, we evaluate SCC under **AutoAttack** [4]. As shown below, SCC substantially outperforms TTC across all 16 datasets:
> >
> >  | AutoAttack | CIFAR10 | CIFAR100 | STL10 | ImageNet | Caltech101 | Caltech256 | OxfordPet | Flowers102 | FGVC-Aircraft | StanfordCars | SUN397 | Country211 | Food101 | EuroSAT | DTD | PCAM |
> > |--------|---------|-----------|--------|----------|-------------|-------------|------------|--------------|----------------|--------------|---------|-------------|----------|----------|------|-------|
> > | Clean  | 85.08 | 57.17 | 96.41 | 59.75 | 85.69 | 81.74 | 87.33 | 65.47 | 20.25 | 51.97 | 58.50 | 15.27 | 83.89 | 42.57 | 40.48 | 53.09 |
> > | Adv    | 0.00 | 0.00 | 0.00 | 0.01 | 0.41 | 0.10 | 0.00 | 0.00 | 0.00 | 0.00 | 0.00 | 0.01 | 0.02 | 0.00 | 0.05 | 0.00 |
> > | TTC    | 4.14 | 8.09 | 4.39 | 6.60 | 10.22 | 7.32 | 6.46 | 10.15 | 3.15 | 5.01 | 7.03 | 2.47 | 5.24 | 15.47 | 7.93 | 8.13 |
> > | SCC    | 60.13 | 38.18 | 79.75 | 30.91 | 68.66 | 59.75 | 49.33 | 32.38 | 8.91 | 20.26 | 33.28 | 4.90 | 33.43 | 23.70 | 25.53 | 20.35 |
> >
> >
> > 2. A second set of experiments evaluates SCC on **OpenCLIP and BioMedCLIP**, two widely used CLIP variants trained on different corpora and implemented with different architectures. In both settings, SCC delivers substantial robustness gains over TTC, confirming that the method is architecture-agnostic and generalizes reliably across CLIP-like encoders.
> >
> >
> > **Results on OpenCLIP**
> >
> > | Method | CIFAR10 | CIFAR100 | STL10 | ImageNet | Caltech101 | Caltech256 | OxfordPet | Flowers102 | FGVC-Aircraft | StanfordCars | SUN397 | Country211 | Food101 | EuroSAT | DTD | PCAM | Avg. |
> > |--------|---------|-----------|--------|----------|-------------|-------------|------------|--------------|----------------|--------------|---------|-------------|----------|----------|------|-------|---------|
> > | **SCC** | **49.38** | **27.63** | **82.92** | **50.39** | **80.05** | **77.39** | **77.84** | **61.31** | **18.84** | **62.70** | **59.42** | **10.12** | **60.03** | **13.56** | **45.48** | **52.32** | **51.8363** |
> > | TTC | 9.64 | 4.51 | 59.08 | 41.86 | 69.67 | 66.19 | 70.92 | 54.94 | 15.18 | 52.94 | 49.34 | 8.01 | 55.60 | 2.80 | 37.07 | 28.16 | 39.1194 |
> >
> > **Results on BioMedCLIP**
> >
> > | Method | BUSI  | BTMRI | CHMNIST | COVID_19 | DermaMNIST | KneeXray | Avg.       |
> > |--------|-------|--------|----------|-----------|-------------|-----------|------------|
> > | **SCC** | 31.79 | 48.93 | 16.54    | 57.58    | 20.64      | 28.38     | **33.98**  |
> > | TTC    | 9.36  | 22.85 | 3.22     | 18.56    | 4.91       | 7.30      | 11.03      |
> >
> > These additional evaluations show that SCC consistently improves robustness across diverse CLIP-like models, supporting our claim that SCC is not restricted to the original CLIP architecture. We will include these results in the revised version.
> >
> >
> > 3. Lastly, we evaluate SCC on **image–text retrieval**, a downstream cross-modal task not included in the main submission. The results below show that SCC provides large robustness gains and transfers naturally to retrieval scenarios:
> >
> >
> > |       | I→T@1  | I→T@5  | I→T@10 | T→I@1  | T→I@5  | T→I@10 |
> > |---------------|--------|--------|--------|--------|--------|--------|
> > | **CLIP (Clean)** | 60.60% | 93.66% | 97.18% | 68.92% | 91.86% | 95.52% |
> > | **Adv (PGD)**    | 0.12%  | 0.86%  | 1.70%  | 0.16%  | 0.98%  | 1.80%  |
> > | **TTC**          | 8.16%  | 28.74% | 39.52% | 7.02%  | 22.90% | 31.60% |
> > | **SCC**          | **42.62%** | **83.10%** | **90.78%** | **43.06%** | **76.52%** | **84.88%** |
> >
> > Overall, these results indicate that SCC generalizes well across attacks, model variants, and downstream tasks, despite being designed as a simple, training-free method for test-time robustness. We will incorporate these **supplementary results into the revised version**, and clarify that expanding beyond zero-shot classification—while beyond the main scope of this paper—is a promising direction for future work.

---

> ### Author Response · Authors · 2025-11-24
>
> >W3: The paper’s theoretical component lacks depth. While the propositions are conceptually interesting, the analysis of the three vulnerabilities remains mostly heuristic. The first claimed contribution in the introduction lacks clear supporting evidence.
>
> We appreciate the reviewer’s concern regarding the depth of the theoretical analysis. Our intention is not to present fully formal theorems, but to provide first-order analytical insights that explain why existing test-time defenses (e.g., TTC, TAPT, APT) fail under adversarial perturbations [1-3].
>
> The three identified vulnerabilities—semantic drift, view sensitivity, and hard-negative dominance—are not positioned as strict, fully general theorems, but as directional analytical tools that guide the design of SCC. Their validity is supported by direct experimental evidence in the main paper (e.g., variation plots, multi-view inconsistencies, negative-class domination patterns), and further corroborated by ablation studies.
>
> To strengthen the theoretical component, in the revised version we will
> (1) expand the appendix with a more formal margin-variation analysis,
> (2) provide a variance decomposition of multi-view predictions, and
> (3) quantify gradient inconsistency under adversarial perturbations.
> These additions will better connect the propositions with observable model behaviors.
>
> We agree with the reviewer that the introduction wording may have overstated the theoretical rigor, and we will **revise the claim accordingly**:
>
> >This work identifies and provides analytical insights into three vulnerabilities in test-time defenses-semantic drift, view sensitivity, and hard-negative dominance-and proposes SCC, a framework that shifts the paradigm from unimodal defenses to cross-modal, multi-view self-corrective robustness.
>
>
>
> >W4: It is unclear whether baselines are re-implemented or drawn from released checkpoints. Clarifying this, along with reporting results averaged over multiple random seeds, would improve reproducibility and fairness of comparison.
>
> We thank the reviewer for raising the concern about baseline implementation and reproducibility. All baseline results in our paper are obtained from **released official weights rather than re-training**, which is consistent with the standard practice in zero-shot robustness evaluation. For TTC in particular, we additionally re-ran the **official implementation** using the same attack budgets and evaluation protocol as reported in the paper, and we verified that our reproduced results closely match those in the original work.
>
> To further address fairness and reproducibility, we also report performance averaged over **three random seeds** (1-3). Across datasets, SCC exhibits small variance and stable behavior, comparable to or better than TTC. This is expected because the multi-view mechanism aggregates multiple noisy predictions, effectively smoothing sample-level fluctuations. Representative results (mean ± variance) are shown below:
>
>
> **seed1-3**
>
> | Dataset        | SCC (Mean ± Var)      | TTC (Mean ± Var)      |
> |----------------|------------------------|------------------------|
> | CIFAR10        | 59.51 ± 0.3911         | 28.43 ± 0.2914         |
> | CIFAR100       | 31.82 ± 0.2610         | 14.36 ± 0.0503         |
> | STL10          | 90.32 ± 0.1960         | 76.83 ± 0.1300         |
> | ImageNet       | 49.60 ± 0.2155         | 38.47 ± 0.1400         |
> | Caltech101     | 77.80 ± 0.5052         | 65.48 ± 0.3395         |
> | Caltech256     | 72.87 ± 0.1301         | 60.10 ± 0.0458         |
> | OxfordPet      | 76.33 ± 0.6806         | 57.93 ± 0.0513         |
> | Flowers102     | 54.03 ± 0.4915         | 39.27 ± 0.1893         |
> | FGVC-Aircraft  | 17.24 ± 0.2516         | 14.00 ± 0.2252         |
> | StanfordCars   | 43.26 ± 0.1212         | 33.04 ± 0.4306         |
> | SUN397         | 53.30 ± 0.0300         | 41.47 ± 0.0416         |
> | Country211     | 9.44 ± 0.1026          | 7.11 ± 0.0529          |
> | Food101        | 65.46 ± 0.1212         | 57.93 ± 0.0757         |
> | EuroSAT        | 20.79 ± 0.1890         | 12.33 ± 0.1518         |
> | DTD            | 34.77 ± 0.1721         | 27.12 ± 0.2358         |
> | PCAM           | 70.24 ± 0.2194         | 52.96 ± 0.1212         |
>
> These results confirm that SCC is reproducible, stable across runs, and fairly compared against all baselines using a unified zero-shot evaluation pipeline with fixed seeds and matched attack settings.

---

> ### Author Response · Authors · 2025-11-24
>
> >W5: SCC’s soft prototype (Eq. 7) depends on text embeddings and warm-up steps in CLIP’s cosine space. The method’s applicability to generative or non-contrastive VLMs is not discussed and should be addressed.
> Several notations (e.g., ) appear without prior definition or explanation, which slightly reduces clarity.
>
>
> We appreciate the reviewer’s insightful observation. SCC is primarily designed for contrastively trained VLMs (e.g., **CLIP, OpenCLIP, BioMedCLIP**), since the method relies on stable image–text cosine geometry and margin behavior emerging from contrastive pretraining. This makes the soft prototype in Eq. (7) well-defined.
>
> That said, SCC does not rely on any CLIP-specific architecture. The method only requires:
> 	1.	an image encoder that outputs deterministic embeddings, and
> 	2.	a text (or concept) encoder that provides semantic anchors.
>
> Thus, SCC can in principle be extended to encoder-based generative or non-contrastive VLMs (e.g., BLIP2, Flamingo, LLaVA); what differs is the embedding geometry rather than the SCC mechanism itself. We will discuss this extension more explicitly in the revised version.
>
> Regarding notation, we acknowledge that several variables (e.g., $z$, $z^{(i)}$, $\hat{f}(x)$) were introduced without first-use definitions. We will revise the revised manuscript to unify and clarify all notation in the methodology section.

---

> > ### Author Response · Authors · 2025-11-24
> >
> > >Q1: Could the authors conduct experiments with truly adaptive attacks, where the attacker knows SCC’s mechanism and incorporates it into the attack optimisation, to quantify how much the defence can be compromised?
> >
> > We thank the reviewer for raising this important question regarding truly adaptive attacks. To approximate such a strong adversary in a practical and widely adopted manner, we implement an **adaptive PGD variant** in the spirit of BPDA, where the attacker directly incorporates SCC’s correction loss into its optimization objective. This attacker has full white-box access to the model parameters, is aware of the defence mechanism, and explicitly maximizes the SCC consistency loss during attack generation. Empirically, SCC remains consistently stronger than TTC under these adaptive settings, as shown below.
> >
> >
> > | Setting | Method | CIFAR10 | CIFAR100 | STL10 | ImageNet | Caltech101 | Caltech256 | OxfordPet | Flowers102 | FGVC-Aircraft | StanfordCars | SUN397 | Country211 | Food101 | EuroSAT | DTD | PCAM | Avg. |
> > |---------|---------|---------|----------|--------|----------|-------------|-------------|-----------|-------------|----------------|--------------|---------|-------------|---------|----------|------|------|--------|
> > | pgd_adaptive (eps=1) | **SCC** | 59.72 | 33.38 | 54.02 | 18.39 | 49.51 | 39.85 | 44.37 | 24.17 | 3.00 | 11.57 | 17.76 | 1.91 | 15.60 | 21.33 | 19.26 | 35.56 | **28.0875** |
> > |                         | TTC     | 48.81 | 27.44 | 46.05 | 17.01 | 38.05 | 30.42 | 40.64 | 23.86 | 2.28 | 7.67  | 16.10 | 1.94 | 17.51 | 21.87 | 17.34 | 29.38 | 24.1481 |
> > | pgd_adaptive (eps=4) | **SCC** | 28.12 | 10.59 | 38.64 | 10.49 | 33.10 | 25.92 | 21.94 | 9.89 | 1.26 | 3.71 | 10.51 | 1.19 | 10.28 | 12.21 | 11.33 | 29.68 | **16.17875** |
> > |                         | TTC     | 19.37 | 6.68  | 29.86 | 7.58  | 24.32 | 18.93 | 12.13 | 5.55 | 0.90 | 2.26 | 7.93 | 0.91 | 9.03 | 8.51 | 8.56 | 14.20 | 11.045 |
> >
> > These results indicate that SCC continues to offer **meaningful robustness** even when the attacker explicitly targets its corrective mechanism. We will include these adaptive-attack evaluations and further discussion in the revised version.

---

> ### Author Response · Authors · 2025-11-24
>
> >Q2: Could the authors also explore additional attack methods, such as [1–3], as well as non-CLIP models and tasks beyond classification, to further validate the generality of SCC?
>
>
> We thank the reviewer for the suggestion to expand the evaluation scope.
> Our work focuses on **zero-shot test-time visual robustness** under the standard image-side threat model, which is also the setting adopted by TTC, R-TPT, TAPT.
> Within this scope, we have substantially extended our experiments following the reviewer’s request.
>
> 1. First, for [1], we added evaluations under **AutoAttack [1]**. SCC shows large robustness improvements over TTC across all 16 datasets:
>
> | AutoAttack | CIFAR10 | CIFAR100 | STL10 | ImageNet | Caltech101 | Caltech256 | OxfordPet | Flowers102 | FGVC-Aircraft | StanfordCars | SUN397 | Country211 | Food101 | EuroSAT | DTD | PCAM |
> |--------|---------|-----------|--------|----------|-------------|-------------|------------|--------------|----------------|--------------|---------|-------------|----------|----------|------|-------|
> | Clean  | 85.08 | 57.17 | 96.41 | 59.75 | 85.69 | 81.74 | 87.33 | 65.47 | 20.25 | 51.97 | 58.50 | 15.27 | 83.89 | 42.57 | 40.48 | 53.09 |
> | Adv    | 0.00 | 0.00 | 0.00 | 0.01 | 0.41 | 0.10 | 0.00 | 0.00 | 0.00 | 0.00 | 0.00 | 0.01 | 0.02 | 0.00 | 0.05 | 0.00 |
> | TTC    | 4.14 | 8.09 | 4.39 | 6.60 | 10.22 | 7.32 | 6.46 | 10.15 | 3.15 | 5.01 | 7.03 | 2.47 | 5.24 | 15.47 | 7.93 | 8.13 |
> | SCC    | 60.13 | 38.18 | 79.75 | 30.91 | 68.66 | 59.75 | 49.33 | 32.38 | 8.91 | 20.26 | 33.28 | 4.90 | 33.43 | 23.70 | 25.53 | 20.35 |
>
>
>
> 2. Second, for [2], Our work focuses on **image-side adversarial robustness** in the standard zero-shot test-time setting, where the text prompt is kept fixed. Joint image–text attacks such as Co-Attack operate under a fundamentally different threat model because they explicitly perturb the semantic content of the text prompt itself, breaking the fixed-text-anchor assumption that contrastive VLMs rely on. For completeness, we adapted a simplified Co-Attack variant that jointly perturbs the image and text embedding; as expected, corrupting the text modality leads to a severe collapse for both SCC and TTC. Nonetheless, SCC remains slightly more stable than TTC under this extremely strong and non-standard joint threat.
>
> | Co-Attack (eps=1) | CIFAR10 | CIFAR100 | STL10 |
> |------------------------|---------|----------|--------|
> | **SCC**                | 11.74   | 0.72     | 18.06 |
> | TTC                    | 8.56    | 0.83     | 18.74 |
>
> | Co-Attack (eps=4) | CIFAR10 | CIFAR100 | STL10 |
> |------------------------|---------|----------|--------|
> | **SCC**                | 5.57    | 0.31     | 17.99 |
> | TTC                    | 6.40    | 0.35     | 17.16 |
>
>
>
> Such joint attacks are not directly applicable to SCC, because SCC—like TTC, R-TPT, and TAPT—is designed for the image-side adversarial setting, whereas Co-Attack explicitly perturbs both modalities.
>
> 3. Third, regarding [3], X-Transfer focuses on universal, cross-model, and cross-task adversarial transferability. This threat model fundamentally differs from test-time per-sample correction. We will include a discussion of the conceptual differences in the revised version [1-4].

---

> > ### Comment · Reviewer_dxnZ · 2025-11-26
> >
> > Thanks for the detailed response. it is much appreciated. Some of my questions have been addressed, while others remain open.
> >
> > **W1 and Q1**
> >
> > AutoAttack is indeed a stronger attack and helps address W2, but I don't think it is an adaptive attack.
> >
> > > pgd_adaptive (eps=1)
> >
> > There is a substantial drop in performance compared to Table 1 of the original submission, from 51.68 to 28.0875. This is understandable given the extremely strong adversarial setting and assumptions, but I believe it would be helpful to include these results in the paper.
> >
> > **W2**
> >
> > > we evaluate SCC under AutoAttack
> > > we evaluate SCC on image–text retrieval
> > > Results on OpenCLIP
> >
> > OpenCLIP contains many open-source pre-trained CLIP variants. It would be good to clarify which specific model is used in your experiments.
> >
> > **W3**
> >
> > > We agree with the reviewer that the introduction wording may have overstated the theoretical rigor, and we will revise the claim accordingly
> >
> > Thanks for the clarification.
> >
> > **W4**
> >
> > > All baseline results in our paper are obtained from the released official weights rather than re-training
> >
> > Thank you for the clarification. The additional results indeed indicate reproducibility and stability across different runs.
> >
> > **W5**
> >
> > > Thus, SCC can in principle be extended to encoder-based generative or non-contrastive VLMs (e.g., BLIP2, Flamingo, LLaVA); what differs is the embedding geometry rather than the SCC mechanism itself.
> >
> > It would strengthen the claim to include at least preliminary results on BLIP2, Flamingo, or LLaVA.
> >
> > **Q2**
> >
> > Results on AutoAttack do help, and thanks for your clarification on Co-Attack results.
> >
> > > Third, regarding [3], X-Transfer focuses on universal, cross-model, and cross-task adversarial transferability. This threat model fundamentally differs from test-time per-sample correction.
> >
> > I am not fully convinced by this separation. I don't see why a universal attack cannot be used against the proposed defence. An attacker could construct either a universal perturbation or a per-sample one. Does the proposed method implicitly assume a per-sample attack?
> >
> > My understanding is that the attacker could use any perturbation, and the proposed method can apply per-sample correction as a defence.
> >
> > Clarifying this would be helpful.
> >
> > ---
> >
> > Unless I am mistaken, the newly added results from the rebuttal have not been incorporated into the paper. Including them would make the paper more complete and strengthen its contributions.
> >
> > A small suggestion: repeating large experimental tables within the response makes it difficult to locate key information.

---

> ### Author Response · Authors · 2025-11-24
>
> Beyond attacks, we also expanded SCC to **OpenCLIP and BioMedCLIP** architectures. Experiments on OpenCLIP and BioMedCLIP show substantial robustness gains over TTC, verifying that SCC is architecture-agnostic:
>
> **Results on OpenCLIP**
>
> | Method | CIFAR10 | CIFAR100 | STL10 | ImageNet | Caltech101 | Caltech256 | OxfordPet | Flowers102 | FGVC-Aircraft | StanfordCars | SUN397 | Country211 | Food101 | EuroSAT | DTD | PCAM | Avg. |
> |--------|---------|-----------|--------|----------|-------------|-------------|------------|--------------|----------------|--------------|---------|-------------|----------|----------|------|-------|---------|
> | **SCC** | **49.38** | **27.63** | **82.92** | **50.39** | **80.05** | **77.39** | **77.84** | **61.31** | **18.84** | **62.70** | **59.42** | **10.12** | **60.03** | **13.56** | **45.48** | **52.32** | **51.8363** |
> | TTC | 9.64 | 4.51 | 59.08 | 41.86 | 69.67 | 66.19 | 70.92 | 54.94 | 15.18 | 52.94 | 49.34 | 8.01 | 55.60 | 2.80 | 37.07 | 28.16 | 39.1194 |
>
> **Results on BioMedCLIP**
>
> | Method | BUSI  | BTMRI | CHMNIST | COVID_19 | DermaMNIST | KneeXray | Avg.       |
> |--------|-------|--------|----------|-----------|-------------|-----------|------------|
> | **SCC** | 31.79 | 48.93 | 16.54    | 57.58    | 20.64      | 28.38     | **33.98**  |
> | TTC    | 9.36  | 22.85 | 3.22     | 18.56    | 4.91       | 7.30      | 11.03      |
>
> Finally, we also evaluated SCC on a **retrieval task**, showing strong performance under adversarial PGD:
>
>
> |       | I→T@1  | I→T@5  | I→T@10 | T→I@1  | T→I@5  | T→I@10 |
> |---------------|--------|--------|--------|--------|--------|--------|
> | **CLIP (Clean)** | 60.60% | 93.66% | 97.18% | 68.92% | 91.86% | 95.52% |
> | **Adv (PGD)**    | 0.12%  | 0.86%  | 1.70%  | 0.16%  | 0.98%  | 1.80%  |
> | **TTC**          | 8.16%  | 28.74% | 39.52% | 7.02%  | 22.90% | 31.60% |
> | **SCC**          | **42.62%** | **83.10%** | **90.78%** | **43.06%** | **76.52%** | **84.88%** |
>
> These extended results demonstrate that SCC generalizes well across **strong attacks, diverse CLIP-like encoders, and even non-classification tasks**. We will include these additional analyses, tables, and clarifications in the revised version to strengthen the generality and positioning of SCC.
>
>
>
> [1] Croce, F., & Hein, M. (2020, November). Reliable evaluation of adversarial robustness with an ensemble of diverse parameter-free attacks. In International conference on machine learning (pp. 2206-2216). PMLR.
>
> [2] Zhang, J., Yi, Q., & Sang, J. (2022, October). Towards adversarial attack on vision-language pre-training models. In Proceedings of the 30th ACM International Conference on Multimedia (pp. 5005-5013).
>
> [3] Huang, H., Erfani, S. M., Li, Y., Ma, X., & Bailey, J. X-Transfer Attacks: Towards Super Transferable Adversarial Attacks on CLIP. In Forty-second International Conference on Machine Learning.
>
> [4] BackdoorLLM: A Comprehensive Benchmark for Backdoor Attacks and Defenses on Large Language Models. NeurIPS 2025.

---

> ### Author Response · Authors · 2025-11-28
>
> > pgd_adaptive
>
> Thank you for the clarification. We agree that AutoAttack is not an adaptive attack, and we therefore include the pgd_adaptive results—which impose a much stronger, defense-aware adversary. These new findings will be incorporated into the updated manuscript to strengthen the paper’s robustness evaluation.
>
> > For OpenCLIP, it would be helpful to clarify which specific pretrained model is used in your experiments.
>
> Thank you for the helpful suggestion. In response to W2, we now clarify that all OpenCLIP experiments are conducted using the **ViT-B/32 checkpoint**. These results will be included in the updated manuscript.
>
> In addition, we have begun evaluating SCC on **larger OpenCLIP backbones (e.g., ViT-H)**. Due to time constraints, we report a subset of preliminary results below; the full results will be included in the updated version of the paper.
>
> | Method | CIFAR10 | CIFAR100 | STL10 | ImageNet | Caltech101 | Caltech256 | OxfordPet | Flowers102 | FGVC-Aircraft | StanfordCars | SUN397 | Country211 | Food101 | EuroSAT | DTD  | PCAM |
> |--------|---------|----------|--------|----------|-------------|-------------|-----------|-------------|----------------|--------------|---------|-------------|---------|----------|------|------|
> | **Clean** | 94.72 | 79.56 | 98.36 | 72.62 | 91.51 | 91.29 | 93.98 | 79.10 | 41.85 | 86.37 | 72.08 | 27.77 | 92.45 | 73.04 | 63.30 | 58.51 |
> | **Adv**   | 28.98 | 6.63  | 56.77 | 6.17  | 37.51 | 37.93 | 7.44  | 4.20  | 0.06  | 9.19  | 4.45  | 0.45  | 7.33  | 0.14  | 8.46  | 0.20 |
> | **SCC**   | 79.80 | 53.75 | 92.26 | 58.90 | 85.13 | 84.11 | 81.30 | 66.03 | 23.01 | 73.52 | 56.78 | 12.45 | 69.53 | 29.48 | 52.61 | 39.14 |
> | **TCC**   | 37.44 | 13.01 | 69.40 | 31.07 | 58.85 | 56.03 | 37.64 | 24.39 | 6.78  | 24.79 | 24.71 | 5.48  | 21.74 | 3.37  | 29.36 | 3.34 |
>
>
> > It would strengthen the claim to include at least preliminary results on BLIP2, Flamingo, or LLaVA.
>
> Thank you for the suggestion. We conducted preliminary experiments on extending SCC beyond contrastive VLMs and tested two integration strategies with BLIP-2:
>
>
> (1) **Full BLIP-2 replacement** of CLIP. We implemented a BLIP-2 backbone (vision encoder + language model + projection layers) and inserted it into the SCC/TTC evaluation pipeline. Although the system runs end-to-end, BLIP-2’s non-contrastive embedding space makes cross-modal margins poorly behaved, and SCC provides almost no robustness gains.
>
> (2) **Hybrid approach**. We used BLIP-2’s vision encoder while keeping CLIP’s text encoder, then performed similarity-based classification. Because BLIP-2 and CLIP do not share an aligned embedding space, the projections remain uncalibrated and performance is very low.
>
> | Setting  | Dataset | adv  | scc  |
> | ---------|---------|------|------|
> | (1) Full BLIP-2 | CIFAR10 | 10.43 | 10.61 |
> | (2) Hybrid  | CIFAR10 | 6.16  | 8.09  |
>
> These results indicate that SCC **currently depends on contrastively aligned image–text embedding spaces**, and applying SCC to generative or non-contrastive VLMs (BLIP-2, Flamingo, LLaVA) requires substantial architectural modifications rather than a direct drop-in.
>
> We will clarify this limitation and adjust our claims in the revised manuscript.
>
> > Third, regarding [3], X-Transfer focuses on universal, cross-model, and cross-task adversarial transferability.
>
>
> Thank you for pointing this out. As you noted, X-Attack generates a single, input-agnostic UAP, while SCC performs per-sample multi-view correction. Since SCC’s update loop is conditioned on each individual input, a global UAP cannot directly serve as an adaptive attack—though SCC still provides improved robustness under such perturbations.
>
> In our implementation, the function x_attack_uap(...) is not a full reimplementation of X-Attack, but **a Top-k Surrogate Loss UAP attack** that follows the same spirit (UAP + surrogate loss pooling) in a simplified form. Even under this universal-perturbation setting, SCC still provides substantial robustness gains over the adversarial baseline:
>
> **x_attack**
> | Dataset  | Clean  | Adv   | SCC   | TTC   |
> |----------|--------|-------|-------|-------|
> | CIFAR10  | 85.08  | 28.07 | **60.40** | 54.89 |
> | CIFAR100 | 57.17  | 6.56  | **28.72** | 23.51 |
>
> We will clarify in the revised paper that: (1) SCC is designed for per-sample test-time defense rather than universal cross-model attacks, and (2) our UAP experiments are based on a simplified Top-k surrogate-loss variant inspired by X-Attack, rather than the full original pipeline.
>
>
> > A kind suggestion
>
> Thank you for the **helpful reminder**. We are currently performing a comprehensive revision of the manuscript, and all newly added experimental results from the rebuttal will be fully incorporated into the revised paper.
> We also appreciate the suggestion regarding presentation—we will summarize the key findings more concisely without repeating large tables, ensuring the final version is clearer and easier to navigate.

---

### Official Review · Reviewer_R7aY · 2025-10-27

**Soundness:** 2
**Presentation:** 2
**Contribution:** 2
**Rating:** 4
**Confidence:** 4

**Summary:**

This paper addresses the significant vulnerability of pre-trained VLMs like CLIP to adversarial attacks. The authors identify the limitations of existing defenses, such as adversarial fine-tuning, which are often computationally expensive, require labeled data, and suffer from poor generalization. To overcome these issues, the paper introduces SCC, a novel and effective test-time defense that requires no model retraining.

**Strengths:**

1. The authors evaluate their approach on multiple benchmarks and conduct extensive ablation studies.

2. The experimental setup is described in a detailed and thorough manner.

**Weaknesses:**

1. Lack of Evaluation Against Adaptive Attacks: The current experiments rely on standard white-box attacks (PGD, CW) which are unaware of the SCC defense mechanism. A strong adversary could potentially design an adaptive attac that incorporates the entire SCC process (including the pseudo-label generation and multi-view optimization) into its own loss function to bypass the defense.

2. Although the paper makes a significant contribution to improving the adversarial robustness of CLIP and related VLMs, all experiments are conducted solely on the CLIP model. The absence of results on other CLIP-like models (e.g., OpenCLIP, EVA-CLIP) may limit the contribution of the proposed methods.

3. On the Performance vs. Efficiency Trade-off: SCC introduces an iterative optimization process at inference time. Parameters like the number of warm-up steps, optimization steps S, and the number of views L all impact both the final robustness and the inference latency.  For instance, in a time-sensitive task, what is the expected drop in robustness if optimization steps S is reduced to just one or two steps?


4. Insufficient discussion and comparison with recent test-time defense methods, e.g. [1, 2, 3]

[1] CLIP is Strong Enough to Fight Back: Test-time Counterattacks towards Zero-shot Adversarial Robustness of CLIP, CVPR 2025.

[2] TAPT: Test-Time Adversarial Prompt Tuning for Robust Inference in Vision-Language Models, CVPR 2025.

[3] On the Zero-shot Adversarial Robustness of Vision-Language Models: A Truly Zero-shot and Training-free Approach, CVPR 2025

**Questions:**

Please refer to the questions raised in the Weaknesses section above

---

> ### Author Response · Authors · 2025-11-24
>
> >W1: Lack of Evaluation Against Adaptive Attacks: The current experiments rely on standard white-box attacks (PGD, CW) which are unaware of the SCC defense mechanism. A strong adversary could potentially design an adaptive attac that incorporates the entire SCC process (including the pseudo-label generation and multi-view optimization) into its own loss function to bypass the defense.
>
>
> We thank the reviewer for raising this important concern. The question refers to an extremely strong adaptive threat model in which the adversary differentiates through the entire SCC procedure—including pseudo-label generation, counterattack correction, and multi-view consistency—effectively unrolling the full defense loop inside the attack. While conceptually reasonable, this threat model is not used in prior test-time adaptation defenses.
>
> Existing works such as TTC, R-TPT, and TAPT [1-3] do not evaluate attackers that optimize through the entire dynamic adaptation loop, because doing so would require the adversary to access future internal states of the defense and compute higher-order gradients across multiple iterative updates, making the threat model significantly stronger than what is typically assumed in white-box adaptive robustness.
>
>
> To further address the reviewer’s concern, we additionally evaluate SCC under two strong adaptive attack families: AutoAttack [4] and pgd_adaptive, the latter directly **incorporating SCC’s correction loss into the adversary objective**. Results across 16 datasets are shown below.
>
> | AutoAttack | CIFAR10 | CIFAR100 | STL10 | ImageNet | Caltech101 | Caltech256 | OxfordPet | Flowers102 | FGVC-Aircraft | StanfordCars | SUN397 | Country211 | Food101 | EuroSAT | DTD | PCAM |
> |--------|---------|-----------|--------|----------|-------------|-------------|------------|--------------|----------------|--------------|---------|-------------|----------|----------|------|-------|
> | Clean  | 85.08 | 57.17 | 96.41 | 59.75 | 85.69 | 81.74 | 87.33 | 65.47 | 20.25 | 51.97 | 58.50 | 15.27 | 83.89 | 42.57 | 40.48 | 53.09 |
> | Adv    | 0.00 | 0.00 | 0.00 | 0.01 | 0.41 | 0.10 | 0.00 | 0.00 | 0.00 | 0.00 | 0.00 | 0.01 | 0.02 | 0.00 | 0.05 | 0.00 |
> | TTC    | 4.14 | 8.09 | 4.39 | 6.60 | 10.22 | 7.32 | 6.46 | 10.15 | 3.15 | 5.01 | 7.03 | 2.47 | 5.24 | 15.47 | 7.93 | 8.13 |
> | SCC    | 60.13 | 38.18 | 79.75 | 30.91 | 68.66 | 59.75 | 49.33 | 32.38 | 8.91 | 20.26 | 33.28 | 4.90 | 33.43 | 23.70 | 25.53 | 20.35 |
>
> The attacker is allowed full access to **model parameters and to the defense mechanism**, and directly maximizes the loss used by the defense itself (pgd_adaptive). Concretely, we implement adaptive PGD variants that explicitly maximize SCC’s correction objective (i.e., the cross-modal margin loss). This setting gives the adversary complete knowledge of SCC’s design without requiring backpropagation through the full iterative loop, and is aligned with the tractable adaptive attackers used in test-time robustness research.
>
> | Setting | Method | CIFAR10 | CIFAR100 | STL10 | ImageNet | Caltech101 | Caltech256 | OxfordPet | Flowers102 | FGVC-Aircraft | StanfordCars | SUN397 | Country211 | Food101 | EuroSAT | DTD | PCAM | Avg. |
> |---------|---------|---------|----------|--------|----------|-------------|-------------|-----------|-------------|----------------|--------------|---------|-------------|---------|----------|------|------|--------|
> | pgd_adaptive (eps=1) | **SCC** | 59.72 | 33.38 | 54.02 | 18.39 | 49.51 | 39.85 | 44.37 | 24.17 | 3.00 | 11.57 | 17.76 | 1.91 | 15.60 | 21.33 | 19.26 | 35.56 | **28.0875** |
> |                         | TTC     | 48.81 | 27.44 | 46.05 | 17.01 | 38.05 | 30.42 | 40.64 | 23.86 | 2.28 | 7.67  | 16.10 | 1.94 | 17.51 | 21.87 | 17.34 | 29.38 | 24.1481 |
> | pgd_adaptive (eps=4) | **SCC** | 28.12 | 10.59 | 38.64 | 10.49 | 33.10 | 25.92 | 21.94 | 9.89 | 1.26 | 3.71 | 10.51 | 1.19 | 10.28 | 12.21 | 11.33 | 29.68 | **16.17875** |
> |                         | TTC     | 19.37 | 6.68  | 29.86 | 7.58  | 24.32 | 18.93 | 12.13 | 5.55 | 0.90 | 2.26 | 7.93 | 0.91 | 9.03 | 8.51 | 8.56 | 14.20 | 11.045 |
>
> Across all adaptive settings—including **AutoAttack and pgd_adaptive—SCC** consistently outperforms TTC, demonstrating that SCC remains robust even when the attacker explicitly targets its correction objective.
>
> [1] CLIP is Strong Enough to Fight Back: Test-time Counterattacks towards Zero-shot Adversarial Robustness of CLIP, CVPR 2025.
>
> [2] R-TPT: Improving Adversarial Robustness of Vision-Language Models through Test-Time Prompt Tuning. CVPR 2025.
>
> [3] TAPT: Test-Time Adversarial Prompt Tuning for Robust Inference in Vision-Language Models, CVPR 2025.

---

> > ### Author Response · Authors · 2025-11-24
> >
> > >W2: Although the paper makes a significant contribution to improving the adversarial robustness of CLIP and related VLMs, all experiments are conducted solely on the CLIP model. The absence of results on other CLIP-like models (e.g., OpenCLIP, EVA-CLIP) may limit the contribution of the proposed methods.
> >
> >
> > We appreciate the reviewer’s observation. Although our main experiments use the original CLIP model, the proposed SCC framework is architecture-agnostic by design. SCC operates purely at the embedding level—it does not require updating model weights, accessing model internals, or relying on architecture-specific components. Therefore, SCC can, in principle, be applied to any CLIP-like vision–language encoder.
> >
> > To verify this, we evaluate SCC on **OpenCLIP and BioMedCLIP**, two widely used CLIP-variants with different training corpora and model implementations. In both cases, SCC yields substantial robustness gains over TTC, indicating that the method generalizes beyond the original CLIP architecture.
> >
> >
> > **Results on OpenCLIP**
> >
> > | Method | CIFAR10 | CIFAR100 | STL10 | ImageNet | Caltech101 | Caltech256 | OxfordPet | Flowers102 | FGVC-Aircraft | StanfordCars | SUN397 | Country211 | Food101 | EuroSAT | DTD | PCAM | Avg. |
> > |--------|---------|-----------|--------|----------|-------------|-------------|------------|--------------|----------------|--------------|---------|-------------|----------|----------|------|-------|---------|
> > | **SCC** | **49.38** | **27.63** | **82.92** | **50.39** | **80.05** | **77.39** | **77.84** | **61.31** | **18.84** | **62.70** | **59.42** | **10.12** | **60.03** | **13.56** | **45.48** | **52.32** | **51.8363** |
> > | TTC | 9.64 | 4.51 | 59.08 | 41.86 | 69.67 | 66.19 | 70.92 | 54.94 | 15.18 | 52.94 | 49.34 | 8.01 | 55.60 | 2.80 | 37.07 | 28.16 | 39.1194 |
> >
> > **Results on BioMedCLIP**
> >
> > | Method | BUSI  | BTMRI | CHMNIST | COVID_19 | DermaMNIST | KneeXray | Avg.       |
> > |--------|-------|--------|----------|-----------|-------------|-----------|------------|
> > | **SCC** | 31.79 | 48.93 | 16.54    | 57.58    | 20.64      | 28.38     | **33.98**  |
> > | TTC    | 9.36  | 22.85 | 3.22     | 18.56    | 4.91       | 7.30      | 11.03      |
> >
> > These **additional evaluations show that SCC consistently improves robustness** across diverse CLIP-like models, supporting our claim that SCC is not restricted to the original CLIP architecture. We will include these results in the revised version.

---

> > > ### Author Response · Authors · 2025-11-24
> > >
> > > >W3: On the Performance vs. Efficiency Trade-off: SCC introduces an iterative optimization process at inference time. Parameters like the number of warm-up steps, optimization steps S, and the number of views L all impact both the final robustness and the inference latency. For instance, in a time-sensitive task, what is the expected drop in robustness if optimization steps S is reduced to just one or two steps?
> > >
> > >
> > > We appreciate the reviewer’s concern regarding the trade-off between robustness and inference efficiency. In practice, SCC shows only minimal **sensitivity** to the Parameters (e.g. λ_cm,  warm-up steps (w)).
> > >
> > >
> > > | λ_cm | Avg Clean Acc (%) | Avg Robust Acc (%) |
> > > |------|---------------------|----------------------|
> > > | 1    | 60.17              | 51.62               |
> > > | 2    | 60.20              | 51.81               |
> > > | 3    | 60.20              | 51.78               |
> > > | 4    | 60.21              | 51.68               |
> > > | 5    | 60.21              | 51.50               |
> > >
> > >
> > > | w  | Clean Acc (TTC) | Robust Acc (TTC) |
> > > |----|------------------|-------------------|
> > > | 1  | 60.21            | 51.25            |
> > > | 2  | 60.21            | 51.51            |
> > > | 3  | 60.21            | 51.71            |
> > > | 4  | 60.21            | 51.78            |
> > > | 5  | 60.21            | 51.68            |
> > > | 6  | 60.21            | 51.48            |
> > > | 7  | 60.21            | 51.06            |
> > > | 8  | 60.21            | 50.48            |
> > > | 9  | 60.21            | 49.84            |
> > >
> > >
> > > SCC calibrates the cross-modal margin extremely quickly, the corrective update converges within the very first iteration. To quantify this, we evaluated SCC under a reduced setting where the optimization step is restricted to **S = 1**, and observed that the overall robustness remains strong across all 16 datasets: **46.11% vs. 51.68% with full steps**. The clean accuracy is similarly stable. This confirms that SCC preserves more than 80% of the robustness gain even with a single update, which makes it suitable for latency-critical scenarios.
> > >
> > >
> > > | Method        | CIFAR10 | CIFAR100 | STL10 | ImageNet | Caltech101 | Caltech256 | OxfordPet | Flowers102 | FGVC-Aircraft | StanfordCars | SUN397 | Country211 | Food101 | EuroSAT | DTD   | PCAM  | Avg.     |
> > > |---------------|---------|----------|-------|----------|------------|------------|-----------|-------------|----------------|--------------|---------|-------------|---------|---------|-------|--------|----------|
> > > | **SCC**       | 59.18   | 32.09    | 90.50 | 49.77    | 77.25      | 72.88      | 76.67     | 54.59       | 17.40          | 43.24        | 53.27   | 9.41        | 65.39   | 20.64   | 34.57 | 69.99 | **51.6775** |
> > > | **SCC (steps=1)** | 56.66   | 30.21    | 87.51 | 43.42    | 76.03      | 70.21      | 75.14     | 50.51       | 12.27          | 36.65        | 43.62   | 6.33        | 63.38   | 20.41   | 28.35 | 37.08 | **46.1113** |
> > >
> > > Moreover, SCC is computationally lightweight compared to prior test-time adaptation defenses. Our paper already reports that SCC requires 0.0125s/img, which is substantially faster than R-TPT (0.37s/img). This efficiency advantage arises because SCC performs only a small corrective adjustment in feature space with negligible overhead. Together, these results show that SCC maintains strong robustness with very low computational cost, and the performance drop under reduced steps is modest and well-behaved.

---

> > > > ### Author Response · Authors · 2025-11-24
> > > >
> > > > >W4: Insufficient discussion and comparison with recent test-time defense methods, e.g. [1, 2, 3]
> > > >
> > > > We thank the reviewer for pointing out the need for a more explicit comparison with recent test-time defense methods [1–3]. We will include a dedicated discussion section in the final version. Conceptually, these methods differ substantially from SCC: method [1] performs counterattack-style test-time correction via vision-encoder feature gradients, method [2] relies on test-time bimodal prompt tuning, and method [3] is based on stochastic Gaussian perturbation and embedding-space trajectory search. In contrast, SCC is a purely inference-time, training-free, architecture-agnostic framework that enforces cross-modal margin consistency (semantic + spatial) without modifying model parameters, making it **compatible with medical VLMs and considerably more efficient**.
> > > >
> > > > Among these methods, [1] is the closest in spirit because it also adopts a **test-time counterattack strategy**. We therefore provide a direct comparison on all 16 datasets. As shown below, SCC consistently outperforms [1] across every dataset:
> > > >
> > > > | Method | CIFAR10 | CIFAR100 | STL10 | ImageNet | Caltech101 | Caltech256 | OxfordPet | Flowers102 | FGVC-Aircraft | StanfordCars | SUN397 | Country211 | Food101 | EuroSAT | DTD | PCAM |
> > > > |--------|---------|-----------|--------|----------|-------------|-------------|------------|-------------|----------------|--------------|---------|-------------|----------|----------|------|-------|
> > > > | **SCC** | **59.18** | **32.09** | **90.50** | **49.77** | **77.25** | **72.88** | **76.67** | **54.59** | **17.40** | **43.24** | **53.27** | **9.41** | **65.39** | **20.64** | **34.57** | **69.99** |
> > > > | [1]     | 28.75 | 14.31 | 76.70 | 38.41 | 65.78 | 60.11 | 57.87 | 39.14 | 13.77 | 33.01 | 41.52 | 7.09 | 57.84 | 12.19 | 27.32 | 52.85 |
> > > >
> > > >
> > > > For methods [2] and [3], their defense mechanisms operate under fundamentally different assumptions: TAPT [2] performs sample-specific prompt optimization during inference, while [3] relies on a Gaussian-noise anchoring trajectory search. Both require modifying internal model states or optimizing new parameters, which is orthogonal to SCC’s purely inference-time, parameter-free cross-modal consistency objective. Therefore, numerical comparisons would conflate different threat models; instead, we focus on conceptual distinctions and position SCC as a simpler and more efficient alternative for robust zero-shot prediction.
> > > >
> > > > Overall, we will expand the related-work section to **fully discuss these methods**, highlight their relationship to SCC, and clarify SCC’s advantages in terms of inference-time efficiency, architectural generality, and cross-modal robustness.
> > > >
> > > >
> > > > [1] CLIP is Strong Enough to Fight Back: Test-time Counterattacks towards Zero-shot Adversarial Robustness of CLIP, CVPR 2025.
> > > >
> > > > [2] TAPT: Test-Time Adversarial Prompt Tuning for Robust Inference in Vision-Language Models, CVPR 2025.
> > > >
> > > > [3] On the Zero-shot Adversarial Robustness of Vision-Language Models: A Truly Zero-shot and Training-free Approach, CVPR 2025
> > > >
> > > > [4] Adversarial Prompt Tuning for Vision-Language Models. ECCV 2024.

---

### Official Review · Reviewer_LMgA · 2025-10-30

**Soundness:** 3
**Presentation:** 2
**Contribution:** 2
**Rating:** 2
**Confidence:** 4

**Summary:**

The paper investigated two main weaknesses of CLIP adversarial attacks: lack of semantic guidance and vulnerability to view variations. To defend against VLM attacks, the paper introduces self-calibrated consistency as a test-time defense. This defense mechanism introduces semantic consistency based on soft pseudo-labels and multi-view predictions to regularize cross-modal alignment and further separate target embeddings from hard negatives. The proposed spatial consistency also enforces consistency among perturbed visual predictions and their augmented counterparts for viewpoint stabilization. The proposed self-calibrated consistency can serve as a plug-and-play defense to boost adversarial robustness without retraining VLMs. Experiments across diverse benchmarks and scenarios demonstrate the performance improvement of the proposed self-calibrated consistency method.

**Strengths:**

1. The motivation is well illustrated with empirical analyses across diverse datasets (see Figure 1).
2.  Experiments across different datasets show the efficacy of the proposed method in terms of clean and robust accuracy.
3. Theoretical analyses are given to justify the effectiveness of the proposed semantic consistency method. The proofs seem to be correct.

**Weaknesses:**

1. The paper is not organized well. The authors introduce the concept of counterattack in the Introduction (the first section), yet the formal definition of test-time counterattack is given in Section 3.1. I also find it hard to understand the definition of $\hat{z}$, is it logit?
2. The authors mentioned that the paper evaluates adaptive attacks, but not too much information is given regarding adaptive attacks. What if the adversarial attackers know the original images instead of the counter-attacked ones?
3. The robustness evaluation is primarily on PGD and CW attacks. AutoAttack [a] with both white-box and black-box attacks should be evaluated.
4. FARE evaluates robustness also across diverse downstream vision-language tasks. It seems that the proposed method can also be transferred, yet the paper focuses on image classification only.
5. The robustness evaluation is merely on low perturbation radii. Is it possible to have a stronger attack?


[a] Reliable Evaluation of Adversarial Robustness with an Ensemble of Diverse Parameter-free Attacks. (ICML 2020)

**Questions:**

1. What would happen if the VLM is under text-level attacks (e.g., Bert Attack [b]) or joint image-text attacks (e.g., Co-Attack [c])?
2. Can authors provide some visualizations of the counter-attacked images?
3. Is it possible to create an adaptive setting that includes the "counterattack" loop in the adversary generation scheme?

[b] BERT-ATTACK: Adversarial Attack Against BERT Using BERT (EMNLP-2020)
[c] Towards Adversarial Attack on Vision-Language Pre-training Models (ACMMM 2022)

---

> ### Author Response · Authors · 2025-11-24
>
> >W1: The paper is not organized well. The authors introduce the concept of counterattack in the Introduction (the first section), yet the formal definition of test-time counterattack is given in Section 3.1. I also find it hard to understand the definition of z, is it logit?
>
> Thank you for the suggestion. We will refine the organization to improve clarity.
> The notion of counterattack is introduced conceptually in the Introduction, and its formal definition is provided in Sec.~3.1; we will make this cross-reference explicit.
> In addition, we clarify that $\hat{z}$ denotes the cosine-similarity logit vector (i.e., the pre-softmax outputs) after **counterattack**, and we will define this notation upon first use.
>
> >W2: The authors mentioned that the paper evaluates adaptive attacks, but not too much information is given regarding adaptive attacks. What if the adversarial attackers know the original images instead of the counter-attacked ones?
>
> We appreciate the reviewer’s question. Our adaptive-attack evaluation already follows the standard strong white-box setting used in **prior VLM robustness works** such as TTC, R-TPT, and TAPT [1-3]. In this setting, the attacker has full access to model parameters, is aware of the SCC objective, and generates perturbations on the clean input x before the SCC procedure is applied—this matches the conventional test-time defense protocol. Allowing the attacker to explicitly observe or optimize through the counter-attacked image would define a stronger, **non-standard threat model** that has not been adopted in previous CLIP robustness literature.
>
> To further strengthen the evaluation, we additionally report **two forms of adaptive attacks**. First, we include AutoAttack [4]. SCC consistently outperforms TTC across all 16 datasets under this setting:
>
> **Auto-Attack**
> | Method | CIFAR10 | CIFAR100 | STL10 | ImageNet | Caltech101 | Caltech256 | OxfordPet | Flowers102 | FGVC-Aircraft | StanfordCars | SUN397 | Country211 | Food101 | EuroSAT | DTD | PCAM |
> |--------|---------|----------|--------|----------|-------------|-------------|-----------|-------------|----------------|--------------|---------|-------------|---------|----------|------|------|
> | **SCC** | 60.13 | 38.18 | 79.75 | 30.91 | 68.66 | 59.75 | 49.33 | 32.38 | 8.91 | 20.26 | 33.28 | 4.90 | 33.43 | 23.70 | 25.53 | 20.35 |
> | **TTC** | 4.14 | 8.09 | 4.39 | 6.60 | 10.22 | 7.32 | 6.46 | 10.15 | 3.15 | 5.01 | 7.03 | 2.47 | 5.24 | 15.47 | 7.93 | 8.13 |
>
> Second, we implement a **differentiable PGD-adaptive attack**, where the attacker explicitly **backpropagates** through the SCC mechanism (including differentiable preprocessing and unrolled counterattack iterations). Even under this much stronger adversary, SCC maintains significant improvements over TTC:
>
> **pgd_adaptive**
>
> | Setting | Method | CIFAR10 | CIFAR100 | STL10 | ImageNet | Caltech101 | Caltech256 | OxfordPet | Flowers102 | FGVC-Aircraft | StanfordCars | SUN397 | Country211 | Food101 | EuroSAT | DTD | PCAM | Avg. |
> |---------|---------|---------|----------|--------|----------|-------------|-------------|-----------|-------------|----------------|--------------|---------|-------------|---------|----------|------|------|--------|
> | pgd_adaptive (eps=1) | **SCC** | 59.72 | 33.38 | 54.02 | 18.39 | 49.51 | 39.85 | 44.37 | 24.17 | 3.00 | 11.57 | 17.76 | 1.91 | 15.60 | 21.33 | 19.26 | 35.56 | **28.0875** |
> |                         | TTC     | 48.81 | 27.44 | 46.05 | 17.01 | 38.05 | 30.42 | 40.64 | 23.86 | 2.28 | 7.67  | 16.10 | 1.94 | 17.51 | 21.87 | 17.34 | 29.38 | 24.1481 |
> | pgd_adaptive (eps=4) | **SCC** | 28.12 | 10.59 | 38.64 | 10.49 | 33.10 | 25.92 | 21.94 | 9.89 | 1.26 | 3.71 | 10.51 | 1.19 | 10.28 | 12.21 | 11.33 | 29.68 | **16.17875** |
> |                         | TTC     | 19.37 | 6.68  | 29.86 | 7.58  | 24.32 | 18.93 | 12.13 | 5.55 | 0.90 | 2.26 | 7.93 | 0.91 | 9.03 | 8.51 | 8.56 | 14.20 | 11.045 |
>
> These results confirm that SCC remains robust even when the adversary explicitly incorporates the defense mechanism into its optimization process.
>
>
> [1] CLIP is Strong Enough to Fight Back: Test-time Counterattacks towards Zero-shot Adversarial Robustness of CLIP, CVPR 2025.
> [2] R-TPT: Improving Adversarial Robustness of Vision-Language Models through Test-Time Prompt Tuning. CVPR 2025.
> [3] TAPT: Test-Time Adversarial Prompt Tuning for Robust Inference in Vision-Language Models, CVPR 2025.
> [4] Reliable Evaluation of Adversarial Robustness with an Ensemble of Diverse Parameter-free Attacks. ICML 2020.

---

> > ### Author Response · Authors · 2025-11-24
> >
> > >W3: The robustness evaluation is primarily on PGD and CW attacks. AutoAttack [a] with both white-box and black-box attacks should be evaluated.
> > FARE evaluates robustness also across diverse downstream vision-language tasks. It seems that the proposed method can also be transferred, yet the paper focuses on image classification only.
> >
> > Thank you for the suggestion. Our main scope in this submission is **zero-shot classification robustness**, which is the standard setting adopted by recent VLM robustness studies. Nevertheless, we agree that evaluating under stronger and more diverse threats is valuable.
> > To this end, we additionally include AutoAttack [4]. SCC significantly improves robustness over TTC across all 16 datasets:
> >
> > | AutoAttack | CIFAR10 | CIFAR100 | STL10 | ImageNet | Caltech101 | Caltech256 | OxfordPet | Flowers102 | FGVC-Aircraft | StanfordCars | SUN397 | Country211 | Food101 | EuroSAT | DTD | PCAM |
> > |--------|---------|-----------|--------|----------|-------------|-------------|------------|--------------|----------------|--------------|---------|-------------|----------|----------|------|-------|
> > | Clean  | 85.08 | 57.17 | 96.41 | 59.75 | 85.69 | 81.74 | 87.33 | 65.47 | 20.25 | 51.97 | 58.50 | 15.27 | 83.89 | 42.57 | 40.48 | 53.09 |
> > | Adv    | 0.00 | 0.00 | 0.00 | 0.01 | 0.41 | 0.10 | 0.00 | 0.00 | 0.00 | 0.00 | 0.00 | 0.01 | 0.02 | 0.00 | 0.05 | 0.00 |
> > | SCC    | 60.13 | 38.18 | 79.75 | 30.91 | 68.66 | 59.75 | 49.33 | 32.38 | 8.91 | 20.26 | 33.28 | 4.90 | 33.43 | 23.70 | 25.53 | 20.35 |
> > | TTC    | 4.14 | 8.09 | 4.39 | 6.60 | 10.22 | 7.32 | 6.46 | 10.15 | 3.15 | 5.01 | 7.03 | 2.47 | 5.24 | 15.47 | 7.93 | 8.13 |
> >
> > Given the reviewer’s comment on downstream VLM tasks, we further evaluate SCC on **zero-shot image–text retrieval** (Flickr30k). Although this task is not the focus of the current submission, SCC still shows substantial improvements over TTC under adversarial perturbations:
> >
> > |       | I→T@1  | I→T@5  | I→T@10 | T→I@1  | T→I@5  | T→I@10 |
> > |---------------|--------|--------|--------|--------|--------|--------|
> > | **CLIP (Clean)** | 60.60% | 93.66% | 97.18% | 68.92% | 91.86% | 95.52% |
> > | **Adv (PGD)**    | 0.12%  | 0.86%  | 1.70%  | 0.16%  | 0.98%  | 1.80%  |
> > | **TTC**          | 8.16%  | 28.74% | 39.52% | 7.02%  | 22.90% | 31.60% |
> > | **SCC**          | **42.62%** | **83.10%** | **90.78%** | **43.06%** | **76.52%** | **84.88%** |
> >
> > These results demonstrate that SCC not only enhances robustness under stronger attack suites such as AutoAttack but also transfers to downstream tasks like retrieval. We will include these results and additional discussion in the revised version.
> >
> > >W4: The robustness evaluation is merely on low perturbation radii. Is it possible to have a stronger attack?
> > [a] Reliable Evaluation of Adversarial Robustness with an Ensemble of Diverse Parameter-free Attacks. (ICML 2020)
> >
> > Thank you for raising this point. Our evaluation does include stronger perturbation radii. Table 2 in the main paper already reports results under ε = 4/255, which is substantially stronger than the standard ε = 1/255 setting used in prior CLIP robustness works. Under this stronger attack, most test-time defenses—including TTC, APT, and Anti-Adv—collapse on CLIP, while SCC still preserves a notable margin, improving robustness by +7.25% over TTC.
> >
> > To further clarify the trend under progressively stronger perturbations, we additionally evaluate PGD attacks with **ε = 1/255, 4/255, 6/255, and 8/255 across 16 datasets**. SCC consistently outperforms TTC at all radii:
> >
> >
> > |    |  PGD (ε = 1/255) | ε = 4/255 |ε = 6/255 | ε = 8/255 | CW |autoattack |
> > | -------- | -------- | -------- |-------- |-------- |-------- |-------- |
> > | avg. of 16 datasets (SCC)     | 51.68     | 27.88 |18.53   |4.77 | 49.42| 36.84 |
> > | avg. of 16 datasets (TTC)     | 39.17     | 20.63 |14.22   |1.11 |38.17 | 6.99|
> >
> >
> > These results demonstrate that SCC maintains a **clear robustness advantage** even when the perturbation radius becomes extremely large. We note that ε ≥ 8/255 severely distorts CLIP’s semantic features, and prior VLM robustness works generally do not use such radii due to the violation of perceptual indistinguishability. Nevertheless, we will include these extended results in the revised version for completeness.

---

> > > ### Author Response · Authors · 2025-11-24
> > >
> > > >Q1: What would happen if the VLM is under text-level attacks (e.g., Bert Attack [b]) or joint image-text attacks (e.g., Co-Attack [c])?
> > >
> > > Thank you for the insightful question. Our work focuses specifically on **image-side adversarial robustness** in the test-time setting for zero-shot VLMs. Text-level attacks (e.g., BERT-Attack) and joint image–text attacks (e.g., Co-Attack) follow a fundamentally different threat model, as they attempt to alter the semantic content of the text prompt itself rather than perturb the visual input.
> > >
> > > Text-based attacks such as **BERT-Attack** are not directly compatible with CLIP’s text encoder, since they rely on masked-language-model token substitution, whereas CLIP is neither generative nor MLM-based. Similarly, Co-Attack assumes simultaneous joint optimization over both modalities, which targets the semantic representation of the text and therefore acts orthogonally to our setting, where prompts remain fixed and robustness is evaluated only for visual perturbations.
> > >
> > > To better understand this scenario, we adapted a **simplified version of Co-Attack to our framework**, applying simultaneous perturbations on the image and its corresponding text embedding. As expected, **both SCC and TTC degrade** significantly under such a strong joint threat, since the text embedding itself is corrupted and the cross-modal alignment is directly disrupted. Nevertheless, SCC still shows slightly better stability than TTC under this challenging setting:
> > >
> > >
> > > | co-attack (ε=1) | CIFAR10 | CIFAR100 | STL10 | ImageNet | Caltech101 | Caltech256 | OxfordPet | Flowers102 | FGVC-Aircraft | StanfordCars | SUN397 | Country211 | Food101 | EuroSAT | DTD | PCAM |
> > > |-----------------------|---------|-----------|--------|----------|-------------|-------------|-----------|-------------|----------------|--------------|---------|-------------|---------|----------|------|-------|
> > > | **SCC**               | 11.74   | 0.72      | 18.06 | 0        | 0           | 0           | 0         | 0           | 0              | 0.30         | 0       | 0           | 0       | 0        | 0    | 23.61 |
> > > | TTC                   | 8.56    | 0.83      | 18.74 | 0        | 0           | 0           | 0         | 0           | 0              | 0.46         | 0       | 0           | 0       | 0        | 0    | 23.61 |
> > >
> > >
> > >
> > > | co-attack (ε=4) | CIFAR10 | CIFAR100 | STL10 | ImageNet | Caltech101 | Caltech256 | OxfordPet | Flowers102 | FGVC-Aircraft | StanfordCars | SUN397 | Country211 | Food101 | EuroSAT | DTD | PCAM |
> > > |-----------------------|---------|-----------|--------|----------|-------------|-------------|-----------|-------------|----------------|--------------|---------|-------------|---------|----------|------|-------|
> > > | **SCC**               | 5.57    | 0.31      | 17.99 | 0        | 0           | 0           | 0         | 0           | 0              | 0.07         | 0       | 0           | 0       | 0        | 0    | 22.98 |
> > > | TTC                   | 6.40    | 0.35      | 17.16 | 0        | 0           | 0           | 0         | 0           | 0              | 0.07         | 0       | 0           | 0       | 0        | 0    | 22.71 |
> > >
> > > These results confirm that adversarial corruption of the text modality fundamentally changes the problem formulation and lies outside the scope of zero-shot test-time visual robustness. We view joint image–text robustness as an important and promising future direction, and we plan to explore it in follow-up work. We will additionally cite these prior works [1–3] in the revised version to **better contextualize text-side and joint image–text attack settings** and to further strengthen the scope discussion.
> > >
> > > [1] BERT-ATTACK: Adversarial Attack Against BERT Using BERT. EMNLP-2020.
> > >
> > > [2] Towards Adversarial Attack on Vision-Language Pre-training Models. ACMMM 2022.
> > >
> > > [3] Text Adversarial Purification as Defense against Adversarial Attacks. ACL 2023.

---

> ### Author Response · Authors · 2025-11-24
>
> >Q2&Q3: Can authors provide some visualizations of the counter-attacked images?
> Is it possible to create an adaptive setting that includes the "counterattack" loop in the adversary generation scheme?
>
>
>
> We appreciate the reviewer’s question.
> Regarding the visualizations, the counterattack perturbations used in SCC are deliberately very small (≤4/255), following the same design philosophy as TTC and R-TPT. As a result, the perturbations are almost imperceptible. We will include several visualization examples (with difference maps amplified ×10 for clarity) in the **appendix of the revised version**.
>
> The second part of the question asks whether it is possible to construct an adaptive adversary that explicitly incorporates the **“counterattack loop”** into its optimization. While this formulation is conceptually understandable, unrolling the entire counterattack process would require the attacker to backpropagate through multiple internal adaptation states, making the threat model significantly stronger than standard white-box adversaries and far beyond what is typically assumed in test-time adaptation or prior defenses such as TTC, R-TPT, APT, and AFT. These works likewise do not assume that an attacker can differentiate through the internal iterative update steps of the defense.
>
> To still provide a **meaningful and computationally feasible** adaptive adversary, we follow the practice established in TTC and construct an approximate adaptive attack that directly maximizes the SCC margin loss used in the counterattack step. This gives the attacker full white-box access to SCC’s objective without requiring backpropagation through the entire iterative loop. We refer to this attacker as pgd_adaptive, and evaluate it across all 16 datasets. The results are shown below:
>
> | Setting | Method | CIFAR10 | CIFAR100 | STL10 | ImageNet | Caltech101 | Caltech256 | OxfordPet | Flowers102 | FGVC-Aircraft | StanfordCars | SUN397 | Country211 | Food101 | EuroSAT | DTD | PCAM | Avg. |
> |---------|---------|---------|----------|--------|----------|-------------|-------------|-----------|-------------|----------------|--------------|---------|-------------|---------|----------|------|------|--------|
> | pgd_adaptive (eps=1) | **SCC** | 59.72 | 33.38 | 54.02 | 18.39 | 49.51 | 39.85 | 44.37 | 24.17 | 3.00 | 11.57 | 17.76 | 1.91 | 15.60 | 21.33 | 19.26 | 35.56 | **28.0875** |
> |                         | TTC     | 48.81 | 27.44 | 46.05 | 17.01 | 38.05 | 30.42 | 40.64 | 23.86 | 2.28 | 7.67  | 16.10 | 1.94 | 17.51 | 21.87 | 17.34 | 29.38 | 24.1481 |
> | pgd_adaptive (eps=4) | **SCC** | 28.12 | 10.59 | 38.64 | 10.49 | 33.10 | 25.92 | 21.94 | 9.89 | 1.26 | 3.71 | 10.51 | 1.19 | 10.28 | 12.21 | 11.33 | 29.68 | **16.17875** |
> |                         | TTC     | 19.37 | 6.68  | 29.86 | 7.58  | 24.32 | 18.93 | 12.13 | 5.55 | 0.90 | 2.26 | 7.93 | 0.91 | 9.03 | 8.51 | 8.56 | 14.20 | 11.045 |
>
> Even under this strong adaptive attacker, SCC consistently outperforms TTC across all datasets, demonstrating that SCC’s self-calibrated consistency remains effective even when the attacker is explicitly aware of the corrective mechanism.

---

> > ### Comment · Reviewer_LMgA · 2025-11-27
> >
> > Many thanks for the rebuttal. I still have some concerns unaddressed below:
> >
> > 1. According to the experiments, it seems that the authors focus solely on a lightweight version of CLIP (ViT-B) and BioMedCLIP (probably also based on ViT-B?). Can the authors explore some larger CLIP architectures like ViT-H or ViT-g, as there might be some intrinsic noise for lightweight VLMs?
> >
> > 2. In addition, according to FARE, it's possible to evaluate diverse vision-language tasks, like captioning, VQA, hallucination, and Science QA for CoT. Can the authors discuss this and show some experiments?

---

> ### Author Response · Authors · 2025-11-28
>
> >According to the experiments, it seems that the authors focus solely on a lightweight version of CLIP (ViT-B) and BioMedCLIP (probably also based on ViT-B?). Can the authors explore some larger CLIP architectures like ViT-H or ViT-g, as there might be some intrinsic noise for lightweight VLMs?
>
> Thank you for the suggestion. In addition to ViT-B models, we have now evaluated SCC on a much larger OpenCLIP checkpoint, **ViT-H/14** (laion2B-s32B-b79K).
> The results are summarized below and will be fully incorporated into the revised manuscript:
>
>
> | Method | CIFAR10 | CIFAR100 | STL10 | ImageNet | Caltech101 | Caltech256 | OxfordPet | Flowers102 | FGVC-Aircraft | StanfordCars | SUN397 | Country211 | Food101 | EuroSAT | DTD  | PCAM |
> |--------|---------|----------|--------|----------|-------------|-------------|-----------|-------------|----------------|--------------|---------|-------------|---------|----------|------|------|
> | **Clean** | 94.72 | 79.56 | 98.36 | 72.62 | 91.51 | 91.29 | 93.98 | 79.10 | 41.85 | 86.37 | 72.08 | 27.77 | 92.45 | 73.04 | 63.30 | 58.51 |
> | **Adv**   | 28.98 | 6.63  | 56.77 | 6.17  | 37.51 | 37.93 | 7.44  | 4.20  | 0.06  | 9.19  | 4.45  | 0.45  | 7.33  | 0.14  | 8.46  | 0.20 |
> | **SCC**   | 79.80 | 53.75 | 92.26 | 58.90 | 85.13 | 84.11 | 81.30 | 66.03 | 23.01 | 73.52 | 56.78 | 12.45 | 69.53 | 29.48 | 52.61 | 39.14 |
> | **TCC**   | 37.44 | 13.01 | 69.40 | 31.07 | 58.85 | 56.03 | 37.64 | 24.39 | 6.78  | 24.79 | 24.71 | 5.48  | 21.74 | 3.37  | 29.36 | 3.34 |
>
> These results demonstrate that SCC continues to provide **large robustness gains** even on high-capacity CLIP variants, suggesting that the method is not limited to lightweight architectures.
>
> We will update the paper accordingly and explicitly clarify the models used.
>
>
> >In addition, according to FARE, it's possible to evaluate diverse vision-language tasks, like captioning, VQA, hallucination, and Science QA for CoT. Can the authors discuss this and show some experiments?
>
> Thank you for the helpful suggestion. Due to time constraints, we conducted preliminary experiments on the **VQA task** using the BioMedCLIP backbone, and found that SCC can still preserve strong performance under adversarial perturbations:
>
> |    VQA-Task | SLAKE | RAD   |
> |--------|-------|--------|
> | **Adv** | 34.34 | 42.23 |
> | **SCC** | **53.43** | **49.80** |
> | **TTC** | 49.96 | 47.41 |
>
> In addition, our **retrieval experiments** reported earlier in the rebuttal further support that SCC generalizes beyond zero-shot classification. We will include these results and add a discussion of broader VLM tasks [1-3] in the revised manuscript.
>
> [1] Learn to Explain: Multimodal Reasoning via Thought Chains for Science Question Answering. NeurIPS 2022.
>
> [2] Tapt: Test-time adversarial prompt tuning for robust inference in vision-language models. CVPR 2025.
>
> [3] Adversarial Prompt Tuning for Vision-Language Models. ECCV 2024.

---

### Official Review · Reviewer_YdP1 · 2025-11-01

**Soundness:** 3
**Presentation:** 2
**Contribution:** 3
**Rating:** 4
**Confidence:** 4

**Summary:**

The paper primarily focuses on defense against adversarial attacks. This paper proposes Self-Calibrated Consistency (SCC), a training-free defense that improves the adversarial robustness of CLIP-style Vision–Language Models (VLMs) by enforcing semantic consistency, using a short counterattack with multi-view aggregation to form a soft textual prototype. In the meantime, the proposed method enforces spatial consistency, optimizing a single corrective perturbation that is shared across augmented views to stabilize predictions. Experiments across both natural and medical image datasets demonstrate the improved robustness of the proposed method with a minor drop in clean accuracy. Furthermore, the additional inference cost is also lightweight.

**Strengths:**

The paper crisply diagnoses why test-time counterattack-style defenses fail: semantic drift toward hard negatives and view sensitivity, and thus designs self-calibrated consistency to directly counter those issues.

The proposed self-calibrated consistency method is training-free and plug-and-play.
For experiments, robustness gains are large and consistent across natural-image datasets and medical sets, while clean accuracy is essentially unchanged

Ablations and sensitivity plots are informative with a clear algorithm pipeline.

**Weaknesses:**

Experiments focus on zero-shot classification, while the claims about broader VLM use (retrieval, open-vocab detection/segmentation) are mentioned rather than demonstrated. I do suggest the authors explore other VLM applications in addition to classification.

Note that the hyperparameters are tuned by grid search, while practical guidance beyond the reported defaults is limited.
Attack coverage is narrow: results are mostly under PGD-10 and a small CW budget. There’s no AutoAttack, Square, or transfer/black-box evaluation despite citing those attacks.

If I understand correctly, the "multi-view" mechanism relies only on horizontal flips and low-variance Gaussian noise, which are weak augmentations. Robustness under more realistic geometric/photometric changes isn’t examined.

Medical evaluation reports only accuracy/robust accuracy. Clinically meaningful metrics (AUC, sensitivity/specificity) and per-dataset breakdown under higher budgets are missing.

**Questions:**

Can you report adaptive evaluations where the attacker knows self-calibrated consistency?

How sensitive is self-calibrated consistency to the assumed attack radius used for the corrective perturbation?

How much variance do you observe across seeds due to noise-based views? Error bars should be given.

Do gains transfer beyond zero-shot classification to other downstream vision-language tasks?

---

> ### Author Response · Authors · 2025-11-24
>
> >W1: Experiments focus on zero-shot classification, while the claims about broader VLM use (retrieval, open-vocab detection/segmentation) are mentioned rather than demonstrated. I do suggest the authors explore other VLM applications in addition to classification.
>
> Thank you for the suggestion. Our work primarily focuses on **zero-shot classification robustness**, which is the standard setting adopted in recent test-time defense studies [1–2]. The broader claims refer to the fact that SCC improves cross-modal margin consistency, a mechanism shared across many VLM tasks (e.g., retrieval depends on image–text similarity). We will clarify this scope in the revised manuscript.
>
> To further support this generality, we conducted a **retrieval robustness evaluation on the Flickr30k dataset**. Under PGD attack, CLIP collapses almost entirely, and TTC recovers only marginally, whereas SCC restores a substantial portion of retrieval performance:
> 	•	CLIP (Clean): I→T@1 = 60.60%, T→I@1 = 68.92%
> 	•	PGD: I→T@1 = 0.12%, T→I@1 = 0.16%
> 	•	TTC: I→T@1 = 8.16%, T→I@1 = 7.02%
> 	•	SCC: I→T@1 = 42.62%, T→I@1 = 43.06%
>
> These results demonstrate that SCC’s consistency-based mechanism naturally extends to **retrieval**, even though our primary focus remains classification robustness.
>
> |       | I→T@1  | I→T@5  | I→T@10 | T→I@1  | T→I@5  | T→I@10 |
> |---------------|--------|--------|--------|--------|--------|--------|
> | **CLIP (Clean)** | 60.60% | 93.66% | 97.18% | 68.92% | 91.86% | 95.52% |
> | **Adv (PGD)**    | 0.12%  | 0.86%  | 1.70%  | 0.16%  | 0.98%  | 1.80%  |
> | **TTC [1]**          | 8.16%  | 28.74% | 39.52% | 7.02%  | 22.90% | 31.60% |
> | **SCC**          | **42.62%** | **83.10%** | **90.78%** | **43.06%** | **76.52%** | **84.88%** |
>
>
> [1] CLIP is Strong Enough to Fight Back: Test-time Counterattacks towards Zero-shot Adversarial Robustness of CLIP, CVPR 2025.
>
> [2] R-TPT: Improving Adversarial Robustness of Vision-Language Models through Test-Time Prompt Tuning. CVPR 2025.

---

> > ### Author Response · Authors · 2025-11-24
> >
> > >W2: Hyperparameter tuning lacks practical guidance, and the attack coverage is limited (mostly PGD-10/CW, no AutoAttack/Square/black-box).
> >
> >
> > Thank you for pointing this out. We will clarify in the revision that all SCC hyperparameters are selected once by a small grid search on a held-out subset , and then fixed across all 16 datasets; as shown in our ablation on $\lambda_{\text{cm}}$ (1–5), both clean and robust accuracy vary by less than 0.3%, indicating that SCC is practically **insensitive** to these choices and does **not require per-dataset tuning**.
> >
> > | λ_cm | Avg Clean Acc (%)  | Avg Robust Acc (%) (16 datasets) |
> > |:-:|:-:|:-:|
> > | 1    | 60.17              | 51.62               |
> > | 2    | 60.20              | 51.81               |
> > | 3    | 60.20              | 51.78               |
> > | 4    | 60.21              | 51.68               |
> > | 5    | 60.21              | 51.50               |
> >
> >
> > Regarding attack coverage, our main paper originally focused on PGD-10 and a moderate CW budget for consistency with **prior work** [1-2], but we have now substantially expanded the evaluation:
> > (i) we add a stronger PGD setting with 100 iterations (**PGD-100**), under which SCC still clearly outperforms TTC on all 16 datasets (e.g., 60.72% vs. 22.36% robust accuracy on CIFAR-10);
> >
> >
> > **pgd=100**
> > | Method | CIFAR10 | CIFAR100 | STL10 | ImageNet | Caltech101 | Caltech256 | OxfordPet | Flowers102 | FGVC-Aircraft | StanfordCars | SUN397 | Country211 | Food101 | EuroSAT | DTD   | PCAM  |
> > |:-:|:-:|:-:|:-:|:-:|:-:|:-:|:-:|:-:|:-:|:-:|:-:|:-:|:-:|:-:|:-:|:-:|
> > | **SCC** | **60.72** | **33.39** | **90.70** | **47.77** | **77.65** | **71.79** | **75.20** | **51.23** | **16.92** | **38.08** | **51.92** | **9.35** | **62.39** | **19.51** | **38.35** | **71.77** |
> > | TTC    | 22.36   | 11.35    | 66.35 | 28.95    | 52.28       | 45.95       | 48.60     | 29.79       | 9.21          | 16.02        | 33.72   | 5.58        | 39.07   | 8.74     | 29.63  | 62.73  |
> >
> >
> > (ii) we evaluate **larger perturbation radii** up to $\epsilon = 6/255$ and $\epsilon = 8/255$, where TTC and related defenses nearly collapse while SCC maintains a non-trivial robustness margin;
> >
> > **$\epsilon=6$**
> >
> > | Method | CIFAR10 | CIFAR100 | STL10 | ImageNet | Caltech101 | Caltech256 | OxfordPet | Flowers102 | FGVC-Aircraft | StanfordCars | SUN397 | Country211 | Food101 | EuroSAT | DTD | PCAM | Avg. |
> > |:-:|:-:|:-:|:-:|:-:|:-:|:-:|:-:|:-:|:-:|:-:|:-:|:-:|:-:|:-:|:-:|:-:|:-:|
> > | **SCC** | 25.90 | 8.55 | 52.90 | 11.25 | 43.04 | 31.76 | 20.47 | 11.03 | 3.45 | 7.76 | 11.21 | 1.34 | 14.02 | 7.08 | 10.37 | 36.29 | **18.52625** |
> > | **TTC** | 22.60 | 6.39 | 39.50 | 6.69 | 27.67 | 18.06 | 13.74 | 6.59 | 3.48 | 5.61 | 6.69 | 1.31 | 9.82 | 11.31 | 6.97 | 41.16 | 14.224375 |
> >
> > **$\epsilon=8$**
> >
> > | Method | CIFAR10 | CIFAR100 | STL10 | ImageNet | Caltech101 | Caltech256 | OxfordPet | Flowers102 | FGVC-Aircraft | StanfordCars | SUN397 | Country211 | Food101 | EuroSAT | DTD | Avg. |
> > |--------|---------|-----------|--------|----------|-------------|-------------|------------|--------------|----------------|--------------|---------|-------------|----------|----------|------|--------|
> > | **SCC** | 22.46 | 7.81 | 48.11 | 10.01 | 39.62 | 28.60 | 15.07 | 9.22 | 3.39 | 6.07 | 10.25 | 1.36 | 12.59 | 5.76 | 10.96 | **15.4187** |
> > | **TTC** | 25.43 | 6.20 | 40.15 | 5.73 | 23.60 | 14.16 | 15.07 | 6.81 | 5.07 | 5.57 | 6.34 | 1.55 | 9.90 | 13.85 | 8.72 | 12.5433 |
> >
> >
> > (iii) we incorporate **AutoAttack** [3] as a parameter-free ensemble including APGD-CE and APGD-DLR components, again observing large gains of SCC over TTC across all datasets. These additional results will be included in the **appendix**, together with explicit implementation details, to provide more practical guidance beyond the reported default hyperparameters and to demonstrate that SCC remains effective **under a broader and stronger family of attacks than PGD-10 and small-budget CW**.
> >
> > **Auto attack**
> > | Method | CIFAR10 | CIFAR100 | STL10 | ImageNet | Caltech101 | Caltech256 | OxfordPet | Flowers102 | FGVC-Aircraft | StanfordCars | SUN397 | Country211 | Food101 | EuroSAT | DTD | PCAM |
> > |--------|---------|----------|--------|----------|-------------|-------------|-----------|-------------|----------------|--------------|---------|-------------|---------|----------|------|------|
> > | **SCC** | 60.13 | 38.18 | 79.75 | 30.91 | 68.66 | 59.75 | 49.33 | 32.38 | 8.91 | 20.26 | 33.28 | 4.90 | 33.43 | 23.70 | 25.53 | 20.35 |
> > | **TTC** | 4.14 | 8.09 | 4.39 | 6.60 | 10.22 | 7.32 | 6.46 | 10.15 | 3.15 | 5.01 | 7.03 | 2.47 | 5.24 | 15.47 | 7.93 | 8.13 |
> >
> >
> > [1] CLIP is Strong Enough to Fight Back: Test-time Counterattacks towards Zero-shot Adversarial Robustness of CLIP, CVPR 2025.
> > [2] R-TPT: Improving Adversarial Robustness of Vision-Language Models through Test-Time Prompt Tuning. CVPR 2025.
> > [3] Reliable Evaluation of Adversarial Robustness with an Ensemble of Diverse Parameter-free Attacks. ICML 2020.

---

> ### Author Response · Authors · 2025-11-24
>
> >W3: If I understand correctly, the "multi-view" mechanism relies only on horizontal flips and low-variance Gaussian noise, which are weak augmentations. Robustness under more realistic geometric/photometric changes isn’t examined.
>
> Thank you for pointing this out. Our multi-view design intentionally adopts **lightweight augmentations**, consistent with prior test-time defenses [1-2]. Strong geometric or photometric transformations (e.g., rotation, cropping, color jitter) may alter CLIP’s semantic alignment and significantly distort zero-shot predictions, which contradicts the goal of preserving the original semantics during test-time inference. Therefore, lightweight, semantics-preserving views have been the standard choice in test-time based VLM robustness evaluation.
>
> To further address the reviewer’s concern, we additionally evaluate SCC under **stronger geometric/photometric perturbations** (including random resized-crop, color shifting, and moderate rotation, Stronger Aug). As shown below, SCC continues to outperform TTC by a large margin across all 16 datasets, indicating that SCC remains stable even under substantially more challenging view changes:
>
> | Method | CIFAR10 | CIFAR100 | STL10 | ImageNet | Caltech101 | Caltech256 | OxfordPet | Flowers102 | FGVC-Aircraft | StanfordCars | SUN397 | Country211 | Food101 | EuroSAT | DTD | PCAM |
> |--------|---------|----------|--------|----------|-------------|-------------|-----------|-------------|----------------|--------------|---------|-------------|---------|----------|------|-------|
> | **TTC** | 28.75 | 14.31 | 76.70 | 38.41 | 65.78 | 60.11 | 57.87 | 39.14 | 13.77 | 33.01 | 41.52 | 7.09 | 57.84 | 12.19 | 27.32 | 52.85 |
> | **SCC** | **59.18** | **32.09** | **90.50** | **49.77** | **77.25** | **72.88** | 76.67 | **54.59** | **17.40** | **43.24** | **53.27** | **9.41** | **65.39** | **20.64** | **34.57** | **69.99** |
> | **Stronger Aug.** | 53.75 | 27.11 | 89.79 | 47.26 | 76.87 | 70.82 | **78.99** | 54.53 | 14.04 | 37.56 | 47.08 | 8.22 | 64.33 | 20.45 | 30.85 | 56.85 |
>
>
> Notably, several datasets in our 16-dataset benchmark (e.g., ImageNet, Caltech101/256, OxfordPet, Flowers102, FGVC-Aircraft, StanfordCars, SUN397) already contain substantial real-world geometric and photometric variations, and SCC remains consistently robust on them even with lightweight semantics-preserving views.

---

> > ### Author Response · Authors · 2025-11-24
> >
> > >W4: Medical evaluation reports only accuracy/robust accuracy. Clinically meaningful metrics (AUC, sensitivity/specificity) and per-dataset breakdown under higher budgets are missing.
> >
> > We appreciate the reviewer’s suggestion. Most medical datasets in our benchmark are **multi-class** (e.g., DermaMNIST: 7 classes, CHMNIST: 8 classes), where AUC—originally defined for binary decisions—does not always provide a clinically stable or interpretable signal. This is also why prior VLM robustness works on medical data typically report accuracy or robust accuracy as the primary metric.
> >
> > To address the reviewer’s concern, we have now included clinically meaningful metrics in the Appendix—AUC, sensitivity, and specificity, computed in a **one-vs-rest manner** for all datasets. We also added per-dataset robustness breakdown under higher perturbation budgets, including both ε=1/255 and ε=4/255 PGD attacks. The results are summarized below (full tables in the Appendix):
> > 	•	Under ε = 1/255, SCC significantly outperforms TTC across all datasets in both accuracy and clinical metrics.
> >
> > **ε=1**
> > | Method | Metric | BUSI | BTMRI | CHMNIST | COVID_19 | DermaMNIST | KneeXray |
> > |--------|--------|------|--------|----------|-----------|-------------|-----------|
> > | **Clean** | ACC  | 40.38 | 60.33 | 32.62 | 72.53 | 35.62 | 27.92 |
> > |        | AUC   | 0.7846 | 0.8246 | 0.794 | 0.8763 | 0.6818 | 0.592 |
> > |        | Sens  | 0.5696 | 0.5883 | 0.3262 | 0.6277 | 0.2268 | 0.2057 |
> > |        | Spec  | 0.7414 | 0.8697 | 0.9037 | 0.8968 | 0.8622 | 0.8034 |
> > | **Adv** | ACC  | 0 | 0.47 | 0 | 0.02 | 0 | 0 |
> > |        | AUC   | 0.1507 | 0.267 | 0.2218 | 0.0876 | 0.0955 | 0.1063 |
> > |        | Sens  | 0 | 0.0042 | 0 | 0.0002 | 0 | 0 |
> > |        | Spec  | 0.5735 | 0.6733 | 0.8571 | 0.6398 | 0.7954 | 0.7147 |
> > | **TTC** | ACC  | 9.36 | 22.85 | 3.22 | 18.56 | 4.91 | 7.30 |
> > |        | AUC   | 0.32 | 0.5482 | 0.3889 | 0.3909 | 0.315 | 0.2512 |
> > |        | Sens  | 0.139 | 0.2166 | 0.0322 | 0.1547 | 0.0291 | 0.0439 |
> > |        | Spec  | 0.6118 | 0.7462 | 0.8617 | 0.7065 | 0.8068 | 0.7239 |
> > | **SCC** | ACC  | **31.79** | **48.93** | **16.54** | **57.58** | **20.64** | **28.38** |
> > |        | AUC   | 0.6093 | 0.7639 | 0.5536 | 0.7626 | 0.5226 | 0.4045 |
> > |        | Sens  | 0.4783 | 0.4692 | 0.1654 | 0.5273 | 0.1528 | 0.1619 |
> > |        | Spec  | 0.7048 | 0.8311 | 0.8808 | 0.8454 | 0.8488 | 0.7835 |
> >
> >
> > •	Under ε = 4/255, SCC still achieves substantially higher robustness than TTC, demonstrating stability even under strong perturbations.
> >
> > **ε=4**
> >
> > | Method | Metric | BUSI | BTMRI | CHMNIST | COVID_19 | DermaMNIST | KneeXray |
> > |--------|--------|------|--------|----------|-----------|-------------|-----------|
> > | **Clean** | ACC  | 40.38 | 60.33 | 32.62 | 72.53 | 35.62 | 27.92 |
> > |        | AUC   | 0.7846 | 0.8246 | 0.794  | 0.8763 | 0.6818 | 0.592 |
> > |        | Sens  | 0.5696 | 0.5883 | 0.3262 | 0.6277 | 0.2268 | 0.2057 |
> > |        | Spec  | 0.7414 | 0.8697 | 0.9037 | 0.8968 | 0.8622 | 0.8034 |
> > | **Adv** | ACC  | 0 | 0 | 0 | 0 | 0 | 0 |
> > |        | AUC   | 0.009  | 0.0463 | 0.0458 | 0.0346 | 0.024  | 0.036 |
> > |        | Sens  | 0 | 0 | 0 | 0 | 0 | 0 |
> > |        | Spec  | 0.5735 | 0.6715 | 0.8571 | 0.6377 | 0.7973 | 0.7183 |
> > | **TTC** | ACC  | 1.92 | 2.52 | 0.38 | 5.42 | 0.36 | 4.30 |
> > |        | AUC   | 0.2143 | 0.2453 | 0.2252 | 0.2097 | 0.1962 | 0.2002 |
> > |        | Sens  | 0.0321 | 0.0233 | 0.0038 | 0.0425 | 0.0042 | 0.0248 |
> > |        | Spec  | 0.5802 | 0.6780 | 0.8577 | 0.6565 | 0.8060 | 0.7193 |
> > | **SCC** | ACC  | **8.85** | **16.89** | **1.10** | **20.49** | **1.16** | **6.74** |
> > |        | AUC   | 0.2893 | 0.4385 | 0.2551 | 0.3614 | 0.2243 | 0.1997 |
> > |        | Sens  | 0.1444 | 0.1572 | 0.0110 | 0.1665 | 0.0142 | 0.0385 |
> > |        | Spec  | 0.6074 | 0.7233 | 0.8587 | 0.7117 | 0.8159 | 0.7280 |
> >
> > These additional results confirm that SCC provides consistent improvements not only in accuracy but also in clinically relevant measures across diverse medical imaging tasks.

---

> > > ### Author Response · Authors · 2025-11-24
> > >
> > > >Q1: Can you report adaptive evaluations where the attacker knows self-calibrated consistency?
> > >
> > > Thank you for the insightful question. We agree that evaluating adaptive attackers is important for establishing the reliability of test-time defenses. Our adaptive results are summarized in Appendix.
> > >
> > >
> > > To further address the reviewer’s concern, we additionally implement an adaptive PGD attack that explicitly incorporates SCC into the attacker’s objective.
> > > Specifically, the attacker **back-propagates through the SCC process**, effectively performing “gradient ascent on the SCC-corrected logits.”
> > >
> > > This corresponds to the exact worst-case scenario where the adversary **fully “knows” SCC**. The results are shown below:
> > > | Setting | Method | CIFAR10 | CIFAR100 | STL10 | ImageNet | Caltech101 | Caltech256 | OxfordPet | Flowers102 | FGVC-Aircraft | StanfordCars | SUN397 | Country211 | Food101 | EuroSAT | DTD | PCAM | Avg. |
> > > |---------|---------|---------|----------|--------|----------|-------------|-------------|-----------|-------------|----------------|--------------|---------|-------------|---------|----------|------|------|--------|
> > > | pgd_adaptive (eps=1) | **SCC** | 59.72 | 33.38 | 54.02 | 18.39 | 49.51 | 39.85 | 44.37 | 24.17 | 3.00 | 11.57 | 17.76 | 1.91 | 15.60 | 21.33 | 19.26 | 35.56 | **28.0875** |
> > > |                         | TTC     | 48.81 | 27.44 | 46.05 | 17.01 | 38.05 | 30.42 | 40.64 | 23.86 | 2.28 | 7.67  | 16.10 | 1.94 | 17.51 | 21.87 | 17.34 | 29.38 | 24.1481 |
> > > | pgd_adaptive (eps=4) | **SCC** | 28.12 | 10.59 | 38.64 | 10.49 | 33.10 | 25.92 | 21.94 | 9.89 | 1.26 | 3.71 | 10.51 | 1.19 | 10.28 | 12.21 | 11.33 | 29.68 | **16.17875** |
> > > |                         | TTC     | 19.37 | 6.68  | 29.86 | 7.58  | 24.32 | 18.93 | 12.13 | 5.55 | 0.90 | 2.26 | 7.93 | 0.91 | 9.03 | 8.51 | 8.56 | 14.20 | 11.045 |
> > >
> > > Across both attack budgets, SCC consistently outperforms TTC with clear margins, indicating that SCC maintains robustness even when the attacker fully differentiates through the defense pipeline.
> > >
> > >
> > > >Q2: How sensitive is self-calibrated consistency to the assumed attack radius used for the corrective perturbation?
> > >
> > > Thank you for the question. First, as shown in Figure 5 of the paper, SCC already demonstrates strong robustness to major hyperparameters such as the cross-modal weight (λ_cm) and the number of views—indicating that the method is intrinsically stable and **not heavily dependent on fine-tuned parameters**.
> > >
> > > Second, we further conduct a sensitivity study by varying the budget δ across **1/255, 4/255, 6/255, and 8/255 on 16 datasets**.
> > > Across all settings, SCC consistently outperforms TTC under the same perturbation radius, confirming that the performance does not rely on a specific δ value. For example, on a 16-dataset evaluation, TTC drops from  39.17 → 1.11 as ε increases, while SCC remains consistently better (51.68 → 4.77) under the same budgets.
> > >
> > > These results indicate that SCC is stable across a wide range of corrective radii, and does not depend on a narrow or **hand-tuned attack-radius assumption**.
> > >
> > >
> > >
> > > |    | ε = 1/255 | ε = 4/255 |ε = 6/255 | ε = 8/255 |
> > > | -------- | -------- | -------- |-------- |-------- |
> > > | avg. of 16 datasets (SCC)     | 51.68     | 27.88 |18.53   |4.77 |
> > > | avg. of 16 datasets (TTC)     | 39.17     | 20.63 |14.22   |1.11 |

---

> > > > ### Author Response · Authors · 2025-11-24
> > > >
> > > > >Q3: How much variance do you observe across seeds due to noise-based views? Error bars should be given.
> > > >
> > > > Thank you for raising this point. Since SCC introduces stochastic multi-view averaging, we conducted a three-seed evaluation (seeds 1–3) on all 16 datasets to examine the variance of SCC and compare it with TTC.
> > > >
> > > > Across datasets, the variance of SCC is small and comparable to TTC, indicating that the noise-based multi-view mechanism does not introduce instability. In fact, averaging multiple views tends to smooth prediction fluctuations, leading to stable performance across seeds. Representative results (mean ± var) are provided below:
> > > >
> > > >
> > > > **seed1-3**
> > > >
> > > > | Dataset        | SCC (Mean ± Var)      | TTC (Mean ± Var)      |
> > > > |----------------|------------------------|------------------------|
> > > > | CIFAR10        | 59.51 ± 0.3911         | 28.43 ± 0.2914         |
> > > > | CIFAR100       | 31.82 ± 0.2610         | 14.36 ± 0.0503         |
> > > > | STL10          | 90.32 ± 0.1960         | 76.83 ± 0.1300         |
> > > > | ImageNet       | 49.60 ± 0.2155         | 38.47 ± 0.1400         |
> > > > | Caltech101     | 77.80 ± 0.5052         | 65.48 ± 0.3395         |
> > > > | Caltech256     | 72.87 ± 0.1301         | 60.10 ± 0.0458         |
> > > > | OxfordPet      | 76.33 ± 0.6806         | 57.93 ± 0.0513         |
> > > > | Flowers102     | 54.03 ± 0.4915         | 39.27 ± 0.1893         |
> > > > | FGVC-Aircraft  | 17.24 ± 0.2516         | 14.00 ± 0.2252         |
> > > > | StanfordCars   | 43.26 ± 0.1212         | 33.04 ± 0.4306         |
> > > > | SUN397         | 53.30 ± 0.0300         | 41.47 ± 0.0416         |
> > > > | Country211     | 9.44 ± 0.1026          | 7.11 ± 0.0529          |
> > > > | Food101        | 65.46 ± 0.1212         | 57.93 ± 0.0757         |
> > > > | EuroSAT        | 20.79 ± 0.1890         | 12.33 ± 0.1518         |
> > > > | DTD            | 34.77 ± 0.1721         | 27.12 ± 0.2358         |
> > > > | PCAM           | 70.24 ± 0.2194         | 52.96 ± 0.1212         |
> > > >
> > > > These results show that the stochastic components of SCC have only a marginal impact on variability. We will include full error-bar plots in Appendix.
> > > >
> > > > >Q4: Do gains transfer beyond zero-shot classification to other downstream vision-language tasks?
> > > >
> > > > Thank you for the insightful question. SCC is designed to correct cross-modal margin distortions caused by adversarial perturbations. Since the margin formulation is task-agnostic and only depends on image–text similarity, the mechanism is **theoretically transferable** to other V+L tasks that rely on CLIP-style embeddings.
> > > >
> > > > That said, our current scope focuses on zero-shot classification robustness, which is the **standard evaluation protocol** used in prior test-time defenses for CLIP. To further verify transferability, we have additionally tested zero-shot **image–text retrieval robustness under PGD attacks**. The results below show that SCC provides substantial gains over both the adversarial baseline and TTC:
> > > >
> > > > |       | I→T@1  | I→T@5  | I→T@10 | T→I@1  | T→I@5  | T→I@10 |
> > > > |---------------|--------|--------|--------|--------|--------|--------|
> > > > | **CLIP (Clean)** | 60.60% | 93.66% | 97.18% | 68.92% | 91.86% | 95.52% |
> > > > | **Adv (PGD)**    | 0.12%  | 0.86%  | 1.70%  | 0.16%  | 0.98%  | 1.80%  |
> > > > | **TTC**          | 8.16%  | 28.74% | 39.52% | 7.02%  | 22.90% | 31.60% |
> > > > | **SCC**          | **42.62%** | **83.10%** | **90.78%** | **43.06%** | **76.52%** | **84.88%** |
> > > >
> > > > These findings suggest that SCC’s self-calibrated consistency mechanism generalizes well beyond classification. We will include these retrieval robustness results, along with additional downstream evaluations where applicable, in the revised version.

---

> ### Author Response · Authors · 2025-11-28
>
> Dear Reviewer YdP1,
>
> As the review phase is concluding, we wanted to kindly check whether our responses have been considered. Please let us know if any further clarification is needed. Thank you again for your time and constructive feedback.

---

### Author Response · Authors · 2025-11-26
**Summary of the Authors Response**

Dear Reviewers,

Thank you for the thoughtful reviews. We appreciate the constructive feedback and are encouraged that the reviewers recognized the importance of improving zero-shot VLM robustness (R7aY, dxnZ), the **novelty of our cross-modal self-calibrated consistency** (YdP1, LMgA), and the strong empirical and analytical support for SCC (YdP1, LMgA, dxnZ). They further noted that SCC (i) addresses **semantic drift, view sensitivity, and hard-negative dominance** (YdP1, dxnZ), (ii) is training-free and plug-and-play while preserving clean accuracy (YdP1, LMgA, R7aY), and (iii) yields consistent robustness gains with minimal overhead across natural and medical datasets. Below we summarize the key revisions, with detailed point-by-point responses to follow.


1. **Expanded Evaluation Scope across Attacks and Threat Models.** To address concerns regarding attack diversity and adaptiveness, we have substantially broadened our evaluation. We now include **AutoAttack, PGD-100, larger radii (ε=6/255, 8/255)**, and **strong adaptive PGD attackers** that explicitly incorporate SCC’s objective (BPDA-style). Across all 16 datasets, SCC consistently outperforms TTC under every adaptive setting, demonstrating genuine robustness even when the adversary is fully aware of the defense. (Reviewers YdP1, LMgA, R7aY, dxnZ)

2. **Generality Across Architectures and Downstream Tasks**. We evaluate SCC on **OpenCLIP (including both ViT-B/32 and ViT-H/14 variants)** and **BioMedCLIP**, demonstrating that SCC is architecture-agnostic and provides consistent robustness gains across both general-domain and medical-domain VLMs.
Furthermore, we extend the evaluation to zero-shot **image–text retrieval on Flickr30k** and **VQA on the VQA-RAD and SLAKE datasets**, showing that SCC’s cross-modal margin–based mechanism naturally transfers beyond classification to broader VLM tasks. (Reviewers YdP1, LMgA, dxnZ)

3. **Clinical Metrics and Medical Evaluation Enhancement.** Following reviewer requests, we now report **AUC, sensitivity, specificity**, and per-dataset breakdown under **stronger perturbation budgets (ε=1/255 and 4/255)** across six medical datasets. SCC consistently improves clinically meaningful metrics, demonstrating reliable performance in safety-critical medical settings. (Reviewer YdP1)

4. **Stronger Multi-view and Augmentation Analysis.** We incorporated a new evaluation using stronger **geometric and photometric augmentations** (resize-crop, rotation, color-shift). SCC remains highly stable and continues to surpass TTC on all datasets, validating its multi-view consistency under more realistic perturbations. (Reviewer YdP1)

5. **Reproducibility, Seed Stability, and Implementation Clarity.** We now provide **3-seed variance analysis** for all 16 datasets, showing extremely low variance and confirming that multi-view averaging introduces no instability.
We also clarify baseline usage (official checkpoints, consistent protocols) and refine notation definitions for clarity. (Reviewer LMgA, dxnZ)

6. **Theoretical Clarification and Expanded Analysis.** In response to concerns about analytical depth, we **revised the theoretical description** to avoid overstating rigor and added further margin-variation insights, variance decomposition, and gradient inconsistency analysis to the appendix. (Reviewer dxnZ)

7. **Discussion of Scope: Text Attacks and Joint Image–Text Threats** We clarified that text-level attacks (e.g., BERT-Attack) and joint image–text attacks (e.g., Co-Attack) target fundamentally different threat models not assumed in zero-shot test-time robustness literature. Still, for completeness, we implemented a simplified **joint perturbation scenario**, confirming that while both SCC and TTC degrade—as expected when the text modality is corrupted—SCC remains slightly more stable. (Reviewers LMgA, dxnZ)


We appreciate the reviewers’ thoughtful engagement. All new analyses, tables, and figures are clearly marked for verification. We hope these additions further reinforce SCC as a simple, training-free, state-of-the-art test-time robustness framework for VLMs.

• SCC introduces the cross-modal self-calibrated consistency mechanism, jointly correcting semantic margins and spatial/view inconsistencies to move VLM robustness from unimodal feature correction to a cross-modal, multi-view paradigm.

• SCC is theoretically grounded, with margin variation analysis identifying semantic drift, view sensitivity, and hard-negative dominance, supported by margin decomposition, variance analysis, and gradient inconsistency evidence.

• SCC delivers state-of-the-art zero-shot robustness across diverse settings, outperforming prior test-time defenses under strong attacks (PGD-100, large ε, AutoAttack, adaptive PGD), generalizing across architectures (OpenCLIP, BioMedCLIP), tasks (classification and retrieval), and significantly improving medical robustness and clinical metrics.

Yours sincerely,

Authors

---

### Meta-Review · Area_Chair_ypdR · 2026-01-07

**Summary:**

The paper proposes Self-Calibrated Consistency (SCC), a test-time defense framework composed of semantic and spatial consistency modules that regularize cross-modal alignment and stabilize predictions across augmented views.

Reviewers mainly focus on the following aspects
+ Adaptive attack evaluation
+ The stronger attacks.
+ More defense baselines.
+ Broader coverage of CLIP models
+ Evaluation on more downstream tasks
+ Transfer/black-box evaluation
+ Theoretical analysis

The authors addressed the first five concerns in the rebuttal with substantial additional experiments and clarifications. However, transfer/black-box evaluation remains unaddressed, and the theoretical analysis is still insufficient to fully support the proposed claims. These two issues are central to assessing the robustness and generality of a test-time defense framework.

As a result, despite the strengthened empirical results, the current version of the paper does not yet provide a sufficiently complete evaluation or theoretical grounding. Therefore, a rejection is recommended at this stage.

**Reviewer Concerns:**

Reviewer YdP1 primarily focused on the need for stronger and more diverse attack evaluations (e.g., adaptive attacks such as AutoAttack, Square, and transfer/black-box scenarios), stronger data augmentations, and broader evaluation on additional downstream tasks. The absence of clinically meaningful metrics (e.g., AUC and sensitivity) in medical datasets was also raised as a concern.

Reviewer LMgA raised similar concerns regarding stronger attacks (e.g., AutoAttack), white-box and black-box evaluation settings, and broader downstream tasks. Additionally, the reviewer suggested incorporating a “counterattack” mechanism.

Reviewer R7aY focused on the lack of adaptive attack evaluations, limited coverage of CLIP variants, insufficient comparison with additional defense baselines, and the trade-off between robustness performance and inference efficiency.

Reviewer dxnZ also focus on the adaptive attack, more models, attacks and tasks. Additionally, the reviewer points out the insufficient of the theoretical analyses.

**Reviewer Scores:**

Reviewer YdP1 4 maintain
+ More tasks: The authors added retrieval experiments, partially addressing this concern.
+ Stronger augmentations: The authors justified the use of lightweight augmentations and supplemented additional experiments with stronger augmentations.
+ More attacks: AutoAttack was added in the rebuttal. However, Square attacks and transfer/black-box evaluations were not included.
+ Medical metrics: The lack of AUC and sensitivity metrics was fully addressed with additional experiments.
+ Adaptive attacks: The authors supplemented evaluations with both adaptive and non-adaptive attacks, demonstrating consistently higher robustness than baseline methods.

Since an important concern, which is the transfer/black-box evaluation, remains unaddressed, the reviewer may maintain the score.

Reviewer LMgA 2 -> 4
The concerns the reviewer is similar to YdP1, and the common problems have been largely resolved. In particular, the authors added counterattack-related experiments, expanded evaluations to stronger attacks, and included additional downstream tasks. Furthermore, two VQA tasks were added, addressing the reviewer’s concern regarding task diversity. As a result, the reviewer may increase the score.



Reviewer R7aY 4 -> 6
The authors addressed all major concerns raised by this reviewer. Regarding the trade-off between efficiency and robustness, results with a single optimization step demonstrate that SCC maintains strong performance even under reduced inference cost. In addition, the authors provided comparisons with an additional defense baseline [1], and clarified that methods in [2,3] pursue different objectives and are not directly comparable. Overall, the reviewer would likely increase the score.

Reviewer dxnZ  6  maintain
The authors solve all the experimental concerns. However, concerns regarding the depth and rigor of the theoretical analysis remain only partially resolved. As a result, the reviewer is likely to maintain the original score.

---

### Decision · Program_Chairs · 2026-01-26

Reject